# Improved Dimension Dependence for Bandit Convex Optimization with Gradient Variation

**Hang Yu** [1][2]  **Yu-Hu Yan** [1][2]  **Peng Zhao** [1][2]

## Abstract

Gradient-variation online learning has drawn increasing attention due to its deep connections to game theory and optimization. It has been studied extensively in the full-information setting, but is underexplored with bandit feedback. In this work, we focus on gradient variation in Bandit Convex Optimization (BCO) with two-point feedback. By proposing a refined analysis of the *non-consecutive* gradient variation, a fundamental quantity in gradient variation with bandit feedback, we improve the dimension dependence for both convex and strongly convex functions compared with the best known results (Chiang et al., 2013). Our improved analysis of the non-consecutive gradient variation also implies other favorable problem-dependent guarantees, such as gradient-variance and small-loss regret bounds. Beyond the two-point setup, we demonstrate the versatility of our technique by achieving the *first* gradient-variation bound for one-point bandit linear optimization over hyper-rectangular domains. Finally, we validate the effectiveness of our results in more challenging tasks such as dynamic and universal regret minimization, establishing the *first* gradient-variation dynamic and universal regret bounds for two-point BCO.

## 1. Introduction

Online Convex Optimization (OCO) is a powerful and fundamental framework for modeling the interaction between a learner and the environment over time (Hazan, 2016; Orabona, 2019). In round $t \in [T]$, the learner selects $\mathbf{x}_t \in \mathcal{X} \subseteq \mathbb{R}^d$, while the environment simultaneously chooses a convex function $f_t : \mathcal{X} \to \mathbb{R}$. Then the learner suffers the loss $f_t(\mathbf{x}_t)$ and receives gradient feedback about the online function, aiming to optimize the game-theoretical performance measure known as regret (Cesa-Bianchi & Lugosi, 2006), which is defined as

$$\mathrm{REG}_T^{(\mathrm{OCO})} \triangleq \sum_{t=1}^{T} f_t(\mathbf{x}_t) - \min_{\mathbf{x} \in \mathcal{X}} \sum_{t=1}^{T} f_t(\mathbf{x}).$$

For OCO, the minimax optimal regret results are $\mathcal{O}(\sqrt{T})$ for convex and $\mathcal{O}(\log T)$ for strongly convex functions (Hazan, 2016). Beyond worst-case minimax optimality, the literature considers enhancing the adaptivity of the learner by adapting the regret to problem-dependent hardness. Among various problem-dependent quantities, *gradient variation* (Chiang et al., 2012; Yang et al., 2014) is of particular interest, defined as the cumulative variation of gradients across consecutive functions:

$$V_T \triangleq \sum_{t=2}^{T} \sup_{\mathbf{x} \in \mathcal{X}} \|\nabla f_t(\mathbf{x}) - \nabla f_{t-1}(\mathbf{x})\|_2^2. \qquad (1.1)$$

By adapting to the gradient variation, the aforementioned minimax regret guarantees can be improved to $\mathcal{O}(\sqrt{V_T})$ for convex functions and $\mathcal{O}(\log V_T)$ for strongly convex functions. Gradient-variation online learning has received renewed attention in recent years, particularly since its introduction into dynamic regret minimization (Zhao et al., 2020; 2024), which has led to dedicated algorithmic structures and new technical tools and has further inspired a growing body of work in various settings (Qiu et al., 2023; Tsai et al., 2023; Yan et al., 2023; Tarzanagh et al., 2024; Xie et al., 2024; Yan et al., 2024; Zhao et al., 2026). This line of research also reveals that gradient-variation adaptivity has profound connections to bridging adversarial and stochastic optimization (Sachs et al., 2022; Chen et al., 2024), enabling fast rates in games (Rakhlin & Sridharan, 2013b; Syrgkanis et al., 2015), and facilitating acceleration in smooth offline optimization (Cutkosky, 2019; Kavis et al., 2019; Yan et al., 2025; Zhao et al., 2025b), among others.

While gradient-variation online learning has been studied extensively in the full-information setting, it is still underexplored in Bandit Convex Optimization (BCO), where the

---

[1]State Key Laboratory for Novel Software Technology, Nanjing University, China [2]School of Artificial Intelligence, Nanjing University, China. Correspondence to: Peng Zhao <zhaop@lamda.nju.edu.cn>.

*Proceedings of the 43$^{rd}$ International Conference on Machine Learning*, Seoul, South Korea. PMLR 306, 2026. Copyright 2026 by the author(s).

*Table 1.* Comparison of problem-dependent regret bounds for two-point BCO. Here, we consider the *nondegenerate* setup for clarity, where we assume $V_T, W_T, F_T \geq \Omega(d)$. $V_T, W_T$, and $F_T$ denote the gradient variation (1.1), gradient variance (3.4), and small loss (3.5), respectively. The $\widetilde{\mathcal{O}}(\cdot)$ notation omits logarithmic factors in the dimension $d$ and the time horizon $T$. We use '—' to denote results that match but do not improve upon state-of-the-art bounds.

| | **Linear** | **Convex** | $\lambda$**-Strongly Convex** |
|---|---|---|---|
| **Chiang et al. (2013)** | $\mathcal{O}\left(d^{\frac{3}{2}}\sqrt{V_T}\right)$ | $\widetilde{\mathcal{O}}\left(d^2\sqrt{V_T}\right)$ | $\mathcal{O}\left(\frac{d^2}{\lambda}\log V_T\right)$ |
| **Ours** [Gradient Variation $V_T$] | — | $\widetilde{\mathcal{O}}\left(d^{\frac{3}{2}}\sqrt{V_T}\right)$ [Theorem 1] | $\mathcal{O}\left(\frac{d}{\lambda}\log V_T\right)$ [Theorem 2] |
| **Ours** [Gradient Variance $W_T$] | $\mathcal{O}\left(\sqrt{dW_T}\right)$ [Theorem 3] | $\mathcal{O}\left(d\sqrt{W_T}\right)$ [Theorem 3] | $\mathcal{O}\left(\frac{d}{\lambda}\log W_T\right)$ [Theorem 3] |
| **Ours** [Small Loss $F_T$] | $\mathcal{O}\left(\sqrt{dF_T}\right)$ [Theorem 4] | $\mathcal{O}\left(\sqrt{dF_T}\right)$ [Theorem 4] | $\mathcal{O}\left(\frac{d}{\lambda}\log F_T\right)$ [Theorem 4] |

learner only has access to the function values. Based on the number of function values queried, BCO can be classified into one-point, two-point, and multi-point settings. In the one-point setup, achieving gradient-variation regret (specialized as squared path-length regret in multi-armed bandits) remains open (Wei & Luo, 2018). By contrast, when it comes to the *two-point* setup, the gradient-variation regret bounds become attainable (Chiang et al., 2013). Specifically, two-point BCO allows the learner to query two points $\mathbf{x}_t, \mathbf{x}_t' \in \mathcal{X}$ at round $t \in [T]$, and to observe the corresponding loss values, $f_t(\mathbf{x}_t)$ and $f_t(\mathbf{x}_t')$, revealed by an oblivious environment. Chiang et al. (2013) initiated the study of gradient variation in two-point BCO and provided the first bounds of $\mathcal{O}(\sqrt{d^3V_T})$, $\widetilde{\mathcal{O}}(d^2\sqrt{V_T})$, and $\mathcal{O}(\frac{d^2}{\lambda}\log(dV_T))$ for linear, convex, and $\lambda$-strongly convex functions, where $d$ is the dimension and $\widetilde{\mathcal{O}}(\cdot)$ omits the logarithmic factors in $T$ and $d$. While their results enjoy the optimal dependence on $V_T$, they incur a large dimension dependence, as the $\Omega(\sqrt{dT})$ convexity lower bound (Duchi et al., 2015) indicates that a tighter dimension dependence is possible for gradient-variation bounds in two-point BCO.

Mitigating the dimension dependence is a fundamental challenge in BCO (Agarwal et al., 2010; Fokkema et al., 2024) and zeroth-order stochastic optimization (Duchi et al., 2015; Nesterov & Spokoiny, 2017; Wang et al., 2018), and there has been a lot of progress on this front. The difficulty stems from the inherent information bottleneck in bandit feedback, where reconstructing a $d$-dimensional gradient from scalar function values necessitates a sampling complexity that scales unfavorably with dimension $d$ (Lattimore, 2025).

In bandit gradient-variation online learning, reducing the dimension dependence poses additional challenges. To see this, we provide an intuition. In OCO, where the learner has access to the full gradient information in *all* directions, e.g., $\nabla f_t$ and $\nabla f_{t-1}$, the gradient-variation regret is straightforward to achieve by using the well-known optimistic online learning technique (Chiang et al., 2012). However, with bandit feedback, the learner can sample *only one* direction at each round. For example, at the $t$-th round, the learner samples a random direction $i_t \in [d]$, constructs a gradient

estimator, and obtains an estimate of $\nabla_{i_t} f_t$, where $\nabla_i f$ denotes the gradient of $f$ in the $i$-th direction. Therefore, it is hard to analyze $\nabla_{i_t} f_t - \nabla_{i_{t-1}} f_{t-1}$ directly because the two directions between consecutive rounds are very likely to be different. To this end, in bandit optimization, an essential quantity is a *non-consecutive* version of the gradient variation (Chiang et al., 2013; Wei & Luo, 2018), which conceptually depends on the following term:

$$\sum_{t=1}^{T}(\nabla_{i_t} f_t - \nabla_{i_t} f_{\alpha_t})^2, \qquad (1.2)$$

where $\alpha_t$ is the largest integer such that $0 \leq \alpha_t < t$ and $i_{\alpha_t} = i_t$. Since the learner can only sample one direction at each round, the non-consecutive sampling gap, i.e., $t - \alpha_t$, will inevitably scale with the dimension $d$, leading to an additional dimension dependence compared with regret bounds in the full-information setting.

In this work, we tighten the dimension dependence of the gradient-variation regret bounds in two-point BCO by unraveling the inherent correlation structure in the *non-consecutive gradient variation*. By carefully decoupling these dependencies, we achieve $\widetilde{\mathcal{O}}(d^{\frac{3}{2}}\sqrt{V_T})$ for convex functions and $\mathcal{O}(\frac{d}{\lambda}\log V_T)$ for $\lambda$-strongly convex functions, thereby improving the best known results by factors of nearly $\sqrt{d}$ and $d$, respectively.

Our analysis for non-consecutive gradient-variation also implies regret scaling with other favorable problem-dependent quantities, such as gradient variance $W_T$ and small loss $F_T$, thereby offering multiple perspectives to depict the problem-dependent hardness. Among the implied results, in particular, we achieve $\mathcal{O}\left(\sqrt{dF_T} + d\right)$ for convex functions and $\mathcal{O}\left(\sqrt{dW_T} + d\right)$ for linear functions, which are both optimal up to an *additive* $\mathcal{O}(d)$ term. Table 1 summarizes our complete results.

Beyond the two-point setup, we generalize our techniques to one-point Bandit Linear Optimization (BLO). Briefly, we introduce a novel gradient estimator with an associated algorithm and establish the *first* gradient-variation regret bound for one-point BLO, in a special case where the domain is a

hyper-rectangle, highlighting the versatility of our approach.

Finally, we showcase the effectiveness of our methods in more challenging environments: *(i) dynamic regret* (Zhang et al., 2018), where the learner competes against oblivious time-varying comparators; *(ii) universal regret* (van Erven & Koolen, 2016), where the learner has no prior knowledge of the function's curvature but aims to match the guarantees of curvature-aware methods. To conclude, we establish the *first* gradient-variation dynamic and universal regret bounds for two-point BCO.

**Contributions.** Our contributions are summarized below:

- For two-point BCO with gradient variation, we obtain $\widetilde{\mathcal{O}}(d^{\frac{3}{2}}\sqrt{V_T})$ and $\mathcal{O}(\frac{d}{\lambda}\log V_T)$ for convex and $\lambda$-strongly convex functions, thereby improving the previously best known results by factors of almost $\sqrt{d}$ and $d$, respectively.

- We achieve the *first* gradient-variance and small-loss regret bounds for two-point BCO; among them, $\mathcal{O}(\sqrt{dF_T} + d)$ and $\mathcal{O}(\sqrt{dW_T} + d)$ for convex and linear functions, respectively, are *the first* problem-dependent guarantees that can recover the minimax optimal $\mathcal{O}(\sqrt{dT})$ regret bound.

- We derive the *first* gradient-variation regret bound in the one-point BLO setting over hyper-rectangular domains.

- We establish the *first* gradient-variation dynamic and universal regret bounds in two-point BCO.

**Organization.** The rest of the paper is organized as follows: In Section 2, we introduce the preliminaries. In Section 3, we present our main results for two-point BCO. In Section 4, we extend our methods to one-point BLO. In Section 5, we generalize our methods to more challenging environments, including dynamic regret and universal regret. Finally, in Section 6, we conclude the paper. Due to page limits, all proofs are deferred to appendices.

## 2. Preliminaries

In this section, we introduce the notation and assumptions, and briefly review the method of Chiang et al. (2013).

### 2.1. Notations and Assumptions

**Notation.** For any $N \in \mathbb{N}$, we define $[N]$ as $\{1, \ldots, N\}$. We represent the $i$-th coordinate of a bold vector $\mathbf{v}$ (or $\boldsymbol{v}$) by the corresponding regular-font symbol $v_i$, i.e., $\mathbf{v}$ (or $\boldsymbol{v}$) = $(v_1, \ldots, v_d)^\top$. We use $\nabla_i f$ to denote the partial derivative of $f$ with respect to the $i$-th coordinate. We use $\|\cdot\|$ for $\|\cdot\|_2$ by default. We write $a \lesssim b$, or $a = \mathcal{O}(b)$, if there exists a constant $C < \infty$ such that $a \leq Cb$. We use $\mathcal{O}(\cdot)$ to highlight the dependence on $d$, $T$, and problem-dependent quantities, while $\widetilde{\mathcal{O}}(\cdot)$ omits logarithmic factors in $d$ and $T$. Throughout the paper, we treat the $\log \log T$ factor as a constant and omit it following Luo & Schapire (2015).

**Assumption 1** (Boundedness)**.** The feasible domain $\mathcal{X} \subseteq \mathbb{R}^d$ is compact, convex, and satisfies $r\mathbb{B} \subseteq \mathcal{X} \subseteq R\mathbb{B}$, where $\mathbb{B} = \{\mathbf{x} \in \mathbb{R}^d \mid \|\mathbf{x}\| \leq 1\}$ is a unit ball.

**Assumption 2** (Lipschitzness)**.** For any $\mathbf{x}, \mathbf{x}' \in \mathcal{X}$ and all $t \in [T]$, $|f_t(\mathbf{x}) - f_t(\mathbf{x}')| \leq G\|\mathbf{x} - \mathbf{x}'\|$.

**Assumption 3** (Smoothness)**.** For any $\mathbf{x}, \mathbf{x}' \in \mathcal{X}$ and all $t \in [T]$, $\|\nabla f_t(\mathbf{x}) - \nabla f_t(\mathbf{x}')\| \leq L\|\mathbf{x} - \mathbf{x}'\|$.

Assumptions 1 and 2 are standard for BCO (Flaxman et al., 2005; Agarwal et al., 2010; Lattimore, 2025). Assumption 3 is essential for first-order methods to achieve problem-dependent regret (Srebro et al., 2010; Chiang et al., 2012).

### 2.2. A Review of Chiang et al. (2013)

For full-information feedback, a standard technique for gradient-variation regret is Optimistic Online Gradient Descent (OOGD) (Chiang et al., 2012). At round $t$, the learner uses an optimism term $M_t$, which serves as a predictive hint of the upcoming gradient $\nabla f_t(\mathbf{x}_t)$. Based on this optimism, OOGD proceeds with the following two-step updates:

$$\mathbf{x}_t = \Pi_{\mathcal{X}}[\widehat{\mathbf{x}}_t - \eta_t M_t], \quad \widehat{\mathbf{x}}_{t+1} = \Pi_{\mathcal{X}}[\widehat{\mathbf{x}}_t - \eta_t \nabla f_t(\mathbf{x}_t)]$$

where $\eta_t > 0$ is a time-varying step size, $\widehat{\mathbf{x}}_t$ and $\widehat{\mathbf{x}}_{t+1}$ are internal decisions, and $\Pi_{\mathcal{X}}[\mathbf{x}] \triangleq \arg\min_{\mathbf{y} \in \mathcal{X}} \|\mathbf{x} - \mathbf{y}\|$ is the Euclidean projection onto the feasible domain $\mathcal{X}$. The resulting regret depends on the cumulative prediction error $\sum_{t=1}^{T} \|\nabla f_t(\mathbf{x}_t) - M_t\|^2$, which characterizes the accuracy of the prediction $M_t$. A straightforward instantiation of $M_t$ is to set it as the preceding gradient $\nabla f_{t-1}(\mathbf{x}_{t-1})$. Such a predictive choice is sufficient to attain the optimal gradient-variation regret (Chiang et al., 2012).

For two-point BCO, where the learner only has access to function values rather than gradients, we define the corresponding cumulative prediction error as:

$$\bar{V}_T \triangleq \sum_{t=1}^{T} \|\mathbf{g}_t - \widetilde{\mathbf{g}}_t\|^2, \tag{2.1}$$

where $\mathbf{g}_t$ is the gradient estimator at round $t$ and $\widetilde{\mathbf{g}}_t$ denotes the optimism constructed from historical information up to round $t - 1$. A direct choice in bandits would use the estimator $\mathbf{g}_t = c_t \mathbf{u}_t$ and set the optimism $\widetilde{\mathbf{g}}_t$ to the preceding estimator $\mathbf{g}_{t-1}$. Here, $\mathbf{u}_t$ denotes a random vector drawn from a specified distribution, and $c_t$ is an estimation constant chosen to ensure $\mathbb{E}[\mathbf{g}_t] \approx \nabla f_t(\mathbf{x}_t)$. For example, in two-point BCO (Agarwal et al., 2010), $\mathbf{u}_t$ is uniformly sampled from the unit sphere, and $c_t = \frac{d}{2\delta}(f_t(\mathbf{x}_t + \delta \mathbf{u}_t) - f_t(\mathbf{x}_t - \delta \mathbf{u}_t))$, where $\delta > 0$ is a small exploration parameter. However, in this case, the gap $\|\mathbf{g}_t - \widetilde{\mathbf{g}}_t\|$ becomes unmanageable, as the randomness of $\mathbf{u}_t$ and $\mathbf{u}_{t-1}$ causes severely misaligned consecutive estimators with high probability.

To bridge this gap, inspired by the gradient estimator in Hazan & Kale (2009a; 2011), Chiang et al. (2013) introduced a novel gradient estimator and an optimism term that effectively address direction misalignment. Specifically, at $t \in [T]$, the gradient estimator $\mathbf{g}_t$ and the optimism $\widetilde{\mathbf{g}}_t$ are constructed as follows:

$$\begin{aligned}
\mathbf{g}_t &= d\left(v_t - \widetilde{g}_{t,i_t}\right)\mathbf{e}_{i_t} + \widetilde{\mathbf{g}}_t, \\
\widetilde{\mathbf{g}}_{t+1} &= \left(v_t - \widetilde{g}_{t,i_t}\right)\mathbf{e}_{i_t} + \widetilde{\mathbf{g}}_t,
\end{aligned} \tag{2.2}$$

where $i_t$ is drawn uniformly from $[d]$, $\{\mathbf{e}_1, \ldots, \mathbf{e}_d\}$ is the standard basis of $\mathbb{R}^d$, and $v_t \triangleq \frac{1}{2\delta}(f_t(\mathbf{w}_t + \delta\mathbf{e}_{i_t}) - f_t(\mathbf{w}_t - \delta\mathbf{e}_{i_t}))$ serves as an estimate of the directional derivative of $f_t$ at $\mathbf{w}_t$ along $\mathbf{e}_{i_t}$. Here, $\mathbf{w}_t$ is the center around which the query points $\mathbf{x}_t$ and $\mathbf{x}'_t$ are sampled as $\mathbf{x}_t = \mathbf{w}_t + \delta\mathbf{e}_{i_t}$ and $\mathbf{x}'_t = \mathbf{w}_t - \delta\mathbf{e}_{i_t}$, where $\delta > 0$ is a small exploration parameter. By concentrating the difference on a single coordinate, $\mathbf{g}_t - \widetilde{\mathbf{g}}_t = d(v_t - \widetilde{g}_{t,i_t})\mathbf{e}_{i_t}$, Eq. (2.2) yields a manageable difference between $\mathbf{g}_t$ and $\widetilde{\mathbf{g}}_t$, which further leads to a controllable $\bar{V}_T$. Leveraging this construction, Chiang et al. (2013) integrated the estimator and optimism in Eq. (2.2) into OOGD. We restate their method in Algorithm 1.

Despite this innovative design, Chiang et al. (2013) underestimated the non-consecutive nature of the gradient estimators. Specifically, by choosing the estimators from Eq. (2.2), $\bar{V}_T$ (2.1) exhibits the following structure:

$$\bar{V}_T = d^2 \sum_{t=1}^{T} (v_t - v_{\alpha_t})^2, \tag{2.3}$$

where $\alpha_t$ is the largest integer such that $0 \le \alpha_t < t$ and $i_{\alpha_t} = i_t$. Intuitively, the $(v_t - v_{\alpha_t})$ term measures the gap between two directional derivative estimates from two iterations in which the sampled direction is the same, leading to a natural non-consecutive structure. Essentially, this term is closely tied to the dimension dependence of the regret bound, as it captures the distance accumulated before the algorithm re-samples the same direction, given that it draws one direction out of $d$ at each round. In the next section, we present the analysis of Chiang et al. (2013) for the essential quantity $\bar{V}_T$, its limitations, and our improved analysis.

## 3. Our Method

In this section, we improve the analysis of the *non-consecutive* structure defined in Eq. (2.3). For the sake of emphasis, we refer to $\bar{V}_T$ (2.3) as the non-consecutive gradient variation in the remainder of the paper. In Section 3.1, we present an improved analysis of $\mathbb{E}[\bar{V}_T]$, which enables us to establish enhanced regret bounds for general convex functions. In Section 3.2, we provide a tight characterization of the maximal term within the expected non-consecutive gradient variation, i.e., $\max_{t\in[T]} \mathbb{E}[\|\mathbf{g}_t - \widetilde{\mathbf{g}}_t\|^2]$. This characterization, coupled with a stabilized step-size schedule, yields

---

**Algorithm 1** Algorithm of Chiang et al. (2013)

**Input:** Step sizes $\{\eta_t\}_{t=1}^{T}$.
1: Let $\mathbf{w}_1 = \widehat{\mathbf{w}}_1 = \mathbf{0}$ and $\widetilde{\mathbf{g}}_1 = \mathbf{0}$. Set exploration parameter $\delta = \frac{1}{2d^2 LTR}$ and shrinkage parameter $\xi = \frac{\delta}{r}$.
2: **for** $t = 1, 2, \ldots, T$ **do**
3:     Choose $i_t$ uniformly from $[d]$.
4:     Submit two query points $\mathbf{x}_t = \mathbf{w}_t + \delta\mathbf{e}_{i_t}, \mathbf{x}'_t = \mathbf{w}_t - \delta\mathbf{e}_{i_t}$, and observe $f_t(\mathbf{x}_t)$ and $f_t(\mathbf{x}'_t)$.
5:     Compute the gradient estimator $\mathbf{g}_t$ and the optimism $\widetilde{\mathbf{g}}_{t+1}$ as in Eq. (2.2)
6:     Update the iterate as follows:

$$\begin{aligned}
\widehat{\mathbf{w}}_{t+1} &= \Pi_{(1-\xi)\mathcal{X}}\left[\widehat{\mathbf{w}}_t - \eta_t\mathbf{g}_t\right], \\
\mathbf{w}_{t+1} &= \Pi_{(1-\xi)\mathcal{X}}\left[\widehat{\mathbf{w}}_{t+1} - \eta_{t+1}\widetilde{\mathbf{g}}_{t+1}\right]
\end{aligned}$$

7: **end for**

---

an improved regret bound in the strongly convex setting. Finally, in Section 3.3, we show that the non-consecutive gradient variation yields other problem-dependent guarantees, such as gradient-variance and small-loss bounds.

### 3.1. Improvement on Convex Case

In this part, we focus on bandit gradient-variation regret for convex functions and improve upon the result of Chiang et al. (2013) by a factor of nearly $\sqrt{d}$, thereby closing the regret gap between the convex and linear settings in the $\widetilde{\mathcal{O}}(\cdot)$-notation, i.e., regardless of the logarithmic factors in the dimension $d$.

We first restate the decomposition of the non-consecutive gradient variation $\bar{V}_T$ (2.3) from Chiang et al. (2013) in Lemma 1, with the proof deferred to Appendix C.1.

**Lemma 1** (Decomposition of $\bar{V}_T$ in Chiang et al. (2013))**.** *Under Assumptions 1-3, for convex functions, Algorithm 1 satisfies the following guarantee:*

$$\bar{V}_T \lesssim 2d^2 \sum_{t=1}^{T} \sum_{i=1}^{d} \rho_{t,i} \left(\nabla_i f_t(\mathbf{w}_{t-1}) - \nabla_i f_{t-1}(\mathbf{w}_{t-1})\right)^2$$

$$+ 2d^2 \sum_{t=1}^{T} \sum_{i=1}^{d} \rho_{t,i} \left(\nabla_i f_t(\mathbf{w}_t) - \nabla_i f_t(\mathbf{w}_{t-1})\right)^2. \tag{3.1}$$

*Here, $\rho_{t,i}$ is the non-consecutive sampling gap defined as $\rho_{t,i} \triangleq \tau_2 - \tau_1$, where $\tau_1 \triangleq \max\{0 \le \tau < t \mid i_\tau = i\}$ and $\tau_2 \triangleq \min\{t \le \tau \le T \mid i_\tau = i\}$, with the convention that $\tau_2 = T + 1$ if no such $\tau$ exists.*

Intuitively, $\rho_{t,i}$ quantifies the duration between the most recent sampling of coordinate $i$ before $t$ and its next sampling at or after $t$. In Eq. (3.1), the first term captures gradient variation along the sampled directions and is therefore connected to $V_T$ (1.1), while the second term essentially mea-

sures the algorithmic stability between $\mathbf{w}_{t-1}$ and $\mathbf{w}_t$, which will be cancelled by negative terms in the regret analysis.

Next, we focus on the first term in Eq. (3.1), due to its close connection to $V_T$. To highlight the difference between our analysis and that of Chiang et al. (2013), we first revisit their analysis in the linear case, then explain why the same strategy suffers an additional $d$ factor in the convex case, and finally show how we overcome this issue.

In the linear setting, as analyzed by Chiang et al. (2013), the gradient difference $\nabla_i f_t(\mathbf{x}) - \nabla_i f_{t-1}(\mathbf{x})$ is constant in $\mathbf{x}$ and is independent of the non-consecutive gap $\rho_{t,i}$. Taking expectation gives

$$\mathbb{E}\left[\sum_{i=1}^d \rho_{t,i}(\nabla_i f_t(\mathbf{w}_{t-1}) - \nabla_i f_{t-1}(\mathbf{w}_{t-1}))^2\right] \quad (3.2)$$
$$\leq \max_{i\in[d]} \mathbb{E}[\rho_{t,i}] \sup_{\mathbf{x}\in\mathcal{X}} \|\nabla f_t(\mathbf{x}) - \nabla f_{t-1}(\mathbf{x})\|^2,$$

where the inequality follows because linear functions have constant gradients. Following Lemma 5 of Chiang et al. (2013), we have $\mathbb{E}[\rho_{t,i}] \leq 2d$ for all $i \in [d]$, yielding an $\mathcal{O}(d)$ dimension dependence.

Furthermore, when the same analysis of Chiang et al. (2013) is applied to convex functions, this independence no longer holds, as both $\rho_{t,i}$ and the gradient difference share the randomness of the direction sampling sequence $\{i_s\}_{s=1}^t$. This interdependence complicates the analysis and leads to a coarse upper bound below:

$$\text{Eq. (3.2)} \leq \mathbb{E}\left[\sum_{i=1}^d \rho_{t,i}\right] \sup_{\mathbf{x}\in\mathcal{X}} \|\nabla f_t(\mathbf{x}) - \nabla f_{t-1}(\mathbf{x})\|^2,$$

where the inequality is due to $w_i^2 \leq \|\mathbf{w}\|^2$ for any vector $\mathbf{w} \in \mathbb{R}^d$. Since each $\mathbb{E}[\rho_{t,i}]$ is of order $\mathcal{O}(d)$, this analysis introduces an additional $d$ factor.

To address this challenge, we decouple the dependence between the sampling gap $\rho_{t,i}$ and the gradient difference by taking the supremum over all sampling gaps:

$$\text{Eq. (3.2)} = \mathbb{E}\left[\sum_{i=1}^d \rho_{t,i}(\nabla_i f_t(\mathbf{w}_{t-1}) - \nabla_i f_{t-1}(\mathbf{w}_{t-1}))^2\right]$$
$$\leq \mathbb{E}\left[\max_{i\in[d]} \rho_{t,i} \sum_{i=1}^d (\nabla_i f_t(\mathbf{w}_{t-1}) - \nabla_i f_{t-1}(\mathbf{w}_{t-1}))^2\right]$$
$$\leq \mathbb{E}\left[\max_{i\in[d]} \rho_{t,i}\right] \sup_{\mathbf{x}\in\mathcal{X}} \|\nabla f_t(\mathbf{x}) - \nabla f_{t-1}(\mathbf{x})\|^2.$$

It remains to control the largest non-consecutive sampling gap $\max_{i\in[d]} \rho_{t,i}$. Our technical contribution is to establish the connection between $\max_{i\in[d]} \rho_{t,i}$ and the Coupon Collector's Problem (CCP). By doing this, we achieve a tighter $\mathbb{E}[\max_{i\in[d]} \rho_{t,i}] = \mathcal{O}(d\log d)$, which introduces only an

extra $\log d$ factor compared with the analysis in the linear setting. The classical CCP studies the waiting time needed to collect all $d$ coupon types when, at each draw, one coupon is sampled uniformly at random from the $d$ types. A standard property of CCP is that this expected waiting time is $\mathcal{O}(d\log d)$ (Motwani & Raghavan, 1995). In our setting, each coordinate sampling $i_t$ can be viewed as drawing one coupon from $[d]$. Specifically, for a fixed time $t$, we decompose the maximal gap as

$$\max_{i\in[d]} \rho_{t,i} \leq \max_{i\in[d]}(t - \tau_1) + \max_{i\in[d]}(\tau_2 - t + 1), \quad (3.3)$$

where we recall that $\tau_1 \triangleq \max\{0 \leq \tau < t \mid i_\tau = i\}$ and $\tau_2 \triangleq \min\{t \leq \tau \leq T \mid i_\tau = i\}$. The first term is the number of steps needed to see all coordinates when searching backward from time $t$, and the second term is the number of steps needed to see all coordinates when searching forward from time $t$. Since the coordinates are sampled independently and uniformly from $[d]$, the expectation of each term is dominated by a CCP waiting time. Applying these upper bounds to both terms in Eq. (3.3), we obtain the desired expected upper bound on $\max_{i\in[d]} \rho_{t,i}$. The detailed reduction is provided in Appendix B.

The aforementioned insight allows us to derive a refined analysis for the non-consecutive gradient variation $\bar{V}_T$, as formalized in Lemma 2. The proof is in Appendix C.2.

**Lemma 2.** *Under Assumptions 1-3, for convex functions, Algorithm 1 satisfies the following guarantee:*

$$\mathbb{E}\left[\bar{V}_T\right] \leq 16d^3 L^2 \log(dT) \cdot \mathbb{E}\left[\sum_{t=1}^T \|\mathbf{w}_t - \mathbf{w}_{t-1}\|^2\right]$$
$$+ 8d^3 V_T \log d + \mathcal{O}(1).$$

By leveraging Lemma 2, we achieve a tighter dimension dependence for convex functions in Theorem 1 below, with the proof deferred to Appendix C.3.

**Theorem 1.** *Under Assumptions 1-3, for convex functions, choosing $\eta_t = \frac{R}{\sqrt{1152d^3 R^4 L^2 \log(dT) + \bar{V}_{t-1}}}$, Algorithm 1 satisfies the following guarantee:*

$$\mathbb{E}[\text{REG}_T] \triangleq \mathbb{E}\left[\sum_{t=1}^T \frac{1}{2}(f_t(\mathbf{x}_t) + f_t(\mathbf{x}'_t)) - \min_{\mathbf{x}\in\mathcal{X}} \sum_{t=1}^T f_t(\mathbf{x})\right]$$
$$\leq \widetilde{\mathcal{O}}\left(\sqrt{\min\{d^3 V_T, dT + d^3\}}\right),$$

*where $V_T$ and $\bar{V}_t$ are defined in Eq. (1.1) and Eq. (2.1).*

Up to logarithmic factors, Theorem 1 effectively closes the regret gap between the convex and linear settings. Furthermore, while the dimension dependence in our gradient variation bound is not as tight as that in the minimax-optimal $\mathcal{O}(\sqrt{dT})$, our result performs better in *benign* environments, e.g., when $V_T = o(T/d^2)$. Meanwhile, our result offers an $\widetilde{\mathcal{O}}(\sqrt{dT + d^3})$ worst-case safeguard, matching optimal regret up to an additive $\widetilde{\mathcal{O}}(d^{3/2})$ term.

## 3.2. Improvement on Strongly Convex Case

In this part, we focus on bandit gradient-variation regret for strongly convex functions. Our solution consists of two key components: a more stable step-size schedule and a tight characterization of the *maximal expected variation*, $\max_{t \in [T]} \mathbb{E}[\|\mathbf{g}_t - \widetilde{\mathbf{g}}_t\|^2]$. To contextualize our improvements, we begin with a brief review of the problem-dependent learning rate by Chiang et al. (2013).

Specifically, Chiang et al. (2013) chose a problem-dependent learning rate schedule as $\eta_t \approx \frac{1}{\lambda \bar{V}_{t-1}}$, where $\bar{V}_{t-1}$ is defined in Eq. (2.1). This learning rate is not stable enough, as the randomness of the gradient estimator can perturb it when the function value varies dramatically, leading to large regret. Moreover, the stochasticity in the step size makes the analysis challenging due to the correlation between the step size and the gradient estimator.

To tackle this issue, we adopt a more stable and deterministic learning rate schedule (Chen et al., 2023):

$$\eta_t = \frac{1}{\lambda t}.$$

Building upon this deterministic step size, we propose a tight analysis for the following maximal expected variation. Below, we establish Lemma 3, with the proof in Appendix A.

**Lemma 3.** *Under Assumptions 1-3, for convex functions, Algorithm 1 satisfies, for any $t \in [T]$,*

$$\mathbb{E}\left[\|\mathbf{g}_t - \widetilde{\mathbf{g}}_t\|^2\right] \leq 8dG^2 + \mathcal{O}\left(\frac{1}{d^2 T^2}\right).$$

By combining the deterministic step size and the tight analysis for the maximal expected variation, we achieve an improved regret guarantee in the strongly convex setting in Theorem 2, with the proof deferred to Appendix C.4.

**Theorem 2.** *Under Assumptions 1-3, for $\lambda$-strongly convex functions, choosing $\eta_t = \frac{1}{\lambda t}$, Algorithm 1 enjoys*

$$\mathbb{E}[\text{REG}_T] \leq \mathcal{O}\left(\frac{d}{\lambda} \log(dV_T)\right).$$

Compared to $\mathcal{O}\left(\frac{d^2}{\lambda} \log(dV_T)\right)$ of Chiang et al. (2013, Theorem 16), Theorem 2 tightens the dimensional dependence from $d^2$ to $d$. As a byproduct, our result also tightens the *worst-case* bound for strongly convex functions, improving the $\mathcal{O}\left(\frac{d^2}{\lambda} \log T\right)$ of Agarwal et al. (2010) by a factor of $d$.

**Corollary 1.** *With the same assumptions and step size as in Theorem 2, Algorithm 1 enjoys $\mathbb{E}[\text{REG}_T] \leq \mathcal{O}(\frac{d}{\lambda} \log T)$.*

Notably, without smoothness, the same regret guarantee can be achieved by a simple algorithm coupled with a dedicated concentration-based analysis (Shamir, 2017). We defer the formal details and analysis to Appendix C.5.

## 3.3. Implications to Small Loss and Gradient Variance

In this part, we demonstrate that with careful analysis, the non-consecutive gradient variation naturally yields gradient-variance regret (Hazan & Kale, 2009a; 2011) and small-loss regret (Srebro et al., 2010; Orabona et al., 2012).

To start with, we present an additional smoothness assumption for small-loss bounds.

**Assumption 4** (Appendix A of Yan et al. (2024)). Under the condition of $\|\nabla f_t(\mathbf{x})\| \leq G$ for any $\mathbf{x} \in \mathcal{X}$ and $t \in [T]$, all online functions are $L$-smooth: $\|\nabla f_t(\mathbf{x}) - \nabla f_t(\mathbf{y})\| \leq L\|\mathbf{x} - \mathbf{y}\|$ for any $t \in [T]$ and $\mathbf{x}, \mathbf{y} \in \mathcal{X}^+$, where $\mathcal{X}^+ \triangleq \{\mathbf{x} + \mathbf{b} \mid \mathbf{x} \in \mathcal{X}, \mathbf{b} \in G/L \cdot \mathbb{B}\}$ is a superset of $\mathcal{X}$.

Without loss of generality, we assume $L \geq 1$ in Assumption 4, since any $L'$-smooth function with $L' \leq L$ is also $L$-smooth. Then, we define the gradient variance $W_T$ as

$$W_T \triangleq \sup_{\boldsymbol{\xi}_1, \ldots, \boldsymbol{\xi}_T \in \mathcal{X}} \left\{ \sum_{t=1}^{T} \|\nabla f_t(\boldsymbol{\xi}_t) - \boldsymbol{\mu}_T\|^2 \right\}, \quad (3.4)$$

where $\boldsymbol{\mu}_T \triangleq \frac{1}{T} \sum_{t=1}^{T} \nabla f_t(\boldsymbol{\xi}_t)$ is the gradient mean. We define the small loss $F_T$ as

$$F_T \triangleq \min_{\mathbf{x} \in \mathcal{X}} \sum_{t=1}^{T} f_t(\mathbf{x}) - \sum_{t=1}^{T} \min_{\mathbf{x} \in \mathcal{X}^+} f_t(\mathbf{x}). \quad (3.5)$$

We clarify that the small-loss definition here generalizes the standard one defined over non-negative functions (Srebro et al., 2010). Thus, it requires smoothness on a superset of the original domain $\mathcal{X}$, as shown in Assumption 4.

Due to space limitations, we only focus on how to obtain $F_T$ bounds from the non-consecutive gradient variation $\bar{V}_T$ here. Specifically, we decompose $\bar{V}_T$ as follows:

$$\mathbb{E}[\bar{V}_T] \lesssim d^2 \mathbb{E}\left[\sum_{t=1}^{T} (\nabla_{i_t} f_t(\mathbf{w}_t) - \nabla_{i_t} f_{\alpha_t}(\mathbf{w}_{\alpha_t}))^2\right]$$
$$\leq 2d^2 \mathbb{E}\left[\sum_{t=1}^{T} \left(\nabla_{i_t} f_t(\mathbf{w}_t)^2 + \nabla_{i_t} f_{\alpha_t}(\mathbf{w}_{\alpha_t})^2\right)\right].$$

The primary challenge lies in evaluating the expectation over $\nabla_{i_t} f_{\alpha_t}(\mathbf{w}_{\alpha_t})^2$, which arises from the non-consecutive structure and the interdependence between $\alpha_t$ and $i_t$. A simplistic way to handle the coupling between $i_t$ and $\alpha_t$ is to coarsely upper-bound the $i_t$-th entry using $w_{i_t}^2 \leq \|\mathbf{w}\|^2$. While this eliminates the need to take expectation over $i_t$, it results in a loose $\mathcal{O}(d\sqrt{F_T})$ bound. To address this, we provide a refined analysis leveraging the law of total expectation to establish Lemma 4. The proof is in Appendix C.6.

**Lemma 4.** *Under Assumptions 1, 2, 4, for convex functions, Algorithm 1 enjoys*

$$\mathbb{E}[\bar{V}_T] \lesssim 16dL\mathbb{E}\left[\sum_{t=1}^{T} f_t(\mathbf{w}_t) - \sum_{t=1}^{T} \min_{\mathbf{x} \in \mathcal{X}^+} f_t(\mathbf{x})\right]. \quad (3.6)$$

Note that the right-hand side of Eq. (3.6) can be converted to the small-loss $F_T$ using standard techniques (Srebro et al., 2010; Orabona et al., 2012).

The analysis for gradient variance follows an analogous approach and is thus omitted here for brevity. To conclude, by leveraging a careful analysis of non-consecutivity, $\bar{V}_T$ also yields gradient-variance and small-loss bounds. We present the corresponding bounds for linear, convex, and strongly convex functions in Theorems 3 and 4. The proofs are deferred to Appendices C.7 and C.8.

**Theorem 3.** *Under Assumptions 1-3, denote by $\bar{V}_t$ the non-consecutive gradient variation defined in Eq. (2.1).*

- *Algorithm 1 with step size $\eta_t = R/\sqrt{d^2 + \bar{V}_{t-1}}$ enjoys $\mathcal{O}\left(\sqrt{dW_T} + d\right)$ regret for linear functions and $\mathcal{O}\left(d\sqrt{W_T + d}\right)$ regret for convex functions.*
- *Algorithm 1 with step size $\eta_t = 1/(\lambda t)$ enjoys $\mathcal{O}\left(\frac{d}{\lambda}\log(dW_T)\right)$ regret for $\lambda$-strongly convex functions.*

**Theorem 4.** *Under Assumptions 1, 2, 4, denote by $\bar{V}_t$ the non-consecutive gradient variation defined in Eq. (2.1).*

- *Algorithm 1 with step size $\eta_t = R/\sqrt{d^2 + \bar{V}_{t-1}}$ enjoys $\mathcal{O}\left(\sqrt{dF_T} + d\right)$ regret for convex and linear functions.*
- *Algorithm 1 with step size $\eta_t = 1/(\lambda t)$ enjoys $\mathcal{O}\left(\frac{d}{\lambda}\log(dF_T)\right)$ regret for $\lambda$-strongly convex functions.*

Up to an *additive* $\mathcal{O}(d)$ term, our gradient-variance bound is optimal for linear functions, while our small-loss result achieves optimality for convex functions. Notably, these two results recover the minimax optimal $\mathcal{O}(\sqrt{dT})$ regret when $T \geq d$. For strongly convex functions, the dimension dependencies of our problem-dependent bounds align with the best known results.

## 4. One-Point Bandit Linear Optimization

Beyond the two-point setup, we adapt our approach, which combines the estimator-and-optimism construction in Eq. (2.2) with a refined analysis of non-consecutive sampling gaps, to the *one-point* Bandit Linear Optimization (BLO) setting. Specifically, in one-point BLO, at each round $t \in [T]$, the learner can query only a single point $\mathbf{x}_t \in \mathcal{X}$ and observes the function value $f_t(\mathbf{x}_t) \triangleq \langle \boldsymbol{\ell}_t, \mathbf{x}_t \rangle$, which is revealed by an oblivious environment.

For one-point BLO, there are partial results that combine optimistic online learning and variance-reduced gradient estimators to derive *gradient-variance* regret (Hazan & Kale, 2009b; 2011). Specifically, their methods update a sequence $\{\mathbf{w}_t\}_{t=1}^T$ and query $\mathbf{x}_t = \mathbf{w}_t + \varepsilon_t \lambda_{t,i_t}^{-\frac{1}{2}} \mathbf{u}_{t,i_t}$. Here, $i_t$ is drawn uniformly from $[d]$, and $\varepsilon_t$ is sampled uniformly from $\{-1, +1\}$. $\lambda_{t,i_t}$ and $\mathbf{u}_{t,i_t}$ denote the $i_t$-th eigenvalue and eigenvector of the Hessian $\nabla^2 \mathcal{R}(\mathbf{w}_t)$ for a barrier function $\mathcal{R}(\cdot)$. The gradient estimator then takes the form of:

$$\mathbf{g}_t = d\langle \boldsymbol{\ell}_t - \widetilde{\mathbf{g}}_t, \mathbf{x}_t \rangle \varepsilon_t \lambda_{t,i_t}^{\frac{1}{2}} \mathbf{u}_{t,i_t} + \widetilde{\mathbf{g}}_t, \qquad (4.1)$$

---

**Algorithm 2** Gradient-Variation One-Point BLO

**Input:** Step size $\eta > 0$
1: **Initialization:** $\mathbf{w}_1 = \mathbf{0}$, $\widetilde{\mathbf{g}}_1 = \mathbf{0}$, $G_0 = \mathbf{0}$ and buffer vectors $\mathbf{r}^{(+1)} = \mathbf{r}^{(-1)} = \mathbf{0} \in \mathbb{R}^d$
2: **for** round $t \in [T]$ **do**
3:   Compute $\widetilde{\mathbf{g}}_t$ as in (4.2)
4:   Choose $i_t$ uniformly from $[d]$ and $\varepsilon_t$ uniformly from $\{-1, +1\}$ and fetch $z_t = r_{i_t}^{(\varepsilon_t)}$
5:   Play action $\mathbf{x}_t = \mathbf{w}_t + \varepsilon_t \lambda_{t,i_t}^{-1/2} \mathbf{e}_{i_t}$
6:   Observe $v_t = \langle \boldsymbol{\ell}_t, \mathbf{x}_t \rangle$ and compute $\mathbf{g}_t$ as in (4.2)
7:   Update buffer vector $r_{i_t}^{(\varepsilon_t)} = v_t$
8:   Calculate $G_t = G_{t-1} + \mathbf{g}_t$ and update

$$\mathbf{w}_{t+1} = \arg\min_{\mathbf{w} \in \mathcal{X}} \left\{ \eta \langle G_t + \widetilde{\mathbf{g}}_{t+1}, \mathbf{w} \rangle + \mathcal{R}(\mathbf{w}) \right\}$$

9: **end for**

---

where $\widetilde{\mathbf{g}}_t$ is an unbiased estimator of the mean loss vector, constructed by maintaining a separate reservoir of historical observations for each coordinate. While their method is effective for obtaining gradient-variance regret bounds for one-point BLO, extending it to gradient-variation regret remains a highly non-trivial open challenge.

Inspired by the construction in Eq. (2.2), we design a novel gradient estimator and establish the *first* gradient-variation regret bound for one-point BLO over hyper-rectangular domains, which is formally defined below.

**Assumption 5.** The domain $\mathcal{X} \subset \mathbb{R}^d$ is a hyper-rectangle of the form $\mathcal{X} = \prod_{i=1}^d [a_i, b_i]$, where $a_i < b_i$ for all $i \in [d]$.

The novel gradient estimator is constructed as follows:

$$\widetilde{\mathbf{g}}_t = \frac{1}{2} \sum_{i=1}^d \lambda_{t,i}^{\frac{1}{2}} \left( r_i^{(+1)} - r_i^{(-1)} \right) \mathbf{e}_i,$$
$$\mathbf{g}_t = d(\langle \boldsymbol{\ell}_t, \mathbf{x}_t \rangle - z_t)\varepsilon_t \lambda_{t,i_t}^{\frac{1}{2}} \mathbf{e}_{i_t} + \widetilde{\mathbf{g}}_t, \qquad (4.2)$$

where $i_t, \varepsilon_t$ share the same definition as in Eq. (4.1). Here, $\mathbf{r}^{(\pm 1)} \in \mathbb{R}^d$ are two auxiliary buffers storing historical function observations associated with the two perturbation signs. Specifically, let $\alpha_t$ be the most recent round before $t$ at which the same coordinate-sign pair was sampled, i.e., the largest integer satisfying $0 \leq \alpha_t < t$, $i_{\alpha_t} = i_t$, and $\varepsilon_{\alpha_t} = \varepsilon_t$. Then, $z_t \triangleq r_{i_t}^{(\varepsilon_t)} = \langle \boldsymbol{\ell}_{\alpha_t}, \mathbf{x}_{\alpha_t} \rangle$ is the stored function value from the most recent occurrence of $(i_t, \varepsilon_t)$. Finally, $\lambda_{t,i}$ denotes the $i$-th eigenvalue of the Hessian $\nabla^2 \mathcal{R}(\mathbf{w}_t)$, where $\mathcal{R}(\cdot)$ is the log-barrier function defined as $\mathcal{R}(\mathbf{w}) = -\sum_{i=1}^d (\log(w_i - a_i) + \log(b_i - w_i))$.

Using Eq. (4.2), we present Algorithm 2 with its regret guarantee in Theorem 5. The proof is deferred to Appendix D.

**Theorem 5.** *Under Assumptions 1, 2, 3, 5, choosing $\eta = \frac{1}{16RGd^2\sqrt{V_T \ln(2dT)}}$, with $V_T \triangleq \sum_{t=2}^T \|\boldsymbol{\ell}_t - \boldsymbol{\ell}_{t-1}\|^2$,*

*Algorithm 2* satisfies:

$$\mathbb{E}[\text{REG}_T] \leq \mathcal{O}\Big(d^{\frac{7}{2}}\sqrt{V_T \log^2 T \log(dT)}\Big).$$

To the best of our knowledge, this is the *first* gradient-variation regret bound for one-point BLO, albeit with the assumption of a hyper-rectangular feasible domain.

**Remark 1.** We adopt Assumption 5 mainly for technical reasons. Specifically, since the Hessian of $\mathcal{R}(\cdot)$ remains diagonal for hyper-rectangular domains, its eigenvectors coincide with the standard basis vectors, which enables the optimistic term $\widetilde{\mathbf{g}}_t$ to store historical data component-wise, making our analysis in Section 3.1 implementable.

**Remark 2.** Note that Algorithm 2 is not directly implementable due to a causality issue: the estimator $\widetilde{\mathbf{g}}_t$ depends on $\mathbf{w}_t$, which is in turn determined by $\widetilde{\mathbf{g}}_t$. However, we can determine $\widetilde{\mathbf{g}}_t$ and $\mathbf{w}_t$ by solving $d$ independent equations involving strictly monotonic functions. Consequently, $\mathbf{w}_t$ can be efficiently approximated via binary search to a precision of $1/T$ within $\mathcal{O}(\log T)$ iterations, incurring only a negligible additive $\mathcal{O}(1)$ term in the final regret. We refer readers to Appendix D.1 for a practical algorithm.

**Remark 3.** Compared with the gradient-variance estimator of Hazan & Kale (2009b; 2011) that maintains reservoir-based historical observations to estimate the running mean of the loss vectors, our objective is different: gradient-variation regret concerns the cumulative differences between loss vectors, rather than their deviations from a running mean. Accordingly, Algorithm 2 stores the most recent function value for each coordinate-sign pair and uses it as a reference in our estimator (4.2). This design yields the decomposition $\|\mathbf{g}_t - \widetilde{\mathbf{g}}_t\|^2 \lesssim \|\boldsymbol{\ell}_t - \boldsymbol{\ell}_{\alpha_t}\|^2 + \|\mathbf{x}_t - \mathbf{x}_{\alpha_t}\|^2$. The summation of the first term above gives the non-consecutive gradient variation, which can be further converted into gradient variation $V_T$, while the second term is handled through the stability analysis of the optimistic methods.

Finally, we stress that establishing gradient-variation regret for one-point BLO is pretty challenging, as it reduces to the open problem of obtaining squared path-length in multi-armed bandits (Wei & Luo, 2018). As a first step, we demonstrate the versatility of our approach on hyper-rectangular domains, deferring the general case to future research.

## 5. Extensions

In this section, we demonstrate the effectiveness of our methods in more challenging tasks, including dynamic and universal regret minimization.

### 5.1. Dynamic Regret Minimization

In this part, we extend our technique to the *dynamic regret* setting (Zinkevich, 2003) in two-point BCO. Specifically,

dynamic regret compares the learner's performance with a sequence of time-varying comparators $\{\mathbf{v}_t\}_{t=1}^T$ and is defined as follows:

$$\text{D-REG}_T \triangleq \frac{1}{2}\sum_{t=1}^T (f_t(\mathbf{x}_{t,1})+f_t(\mathbf{x}_{t,2}))-\sum_{t=1}^T f_t(\mathbf{v}_t). \quad (5.1)$$

Ideally, dynamic regret guarantees scale with the path length $P_T \triangleq \sum_{t=2}^T \|\mathbf{v}_t - \mathbf{v}_{t-1}\|$, which captures the environmental non-stationarity and is unknown to the learner. In the full-information setting, Zhang et al. (2018) obtained the minimax-optimal dynamic regret bound $\mathcal{O}(\sqrt{T(1+P_T)})$, which was later strengthened for smooth functions to the gradient-variation dynamic regret bound $\mathcal{O}(\sqrt{(1+V_T+P_T)(1+P_T)})$ (Zhao et al., 2024). For two-point BCO, Zhao et al. (2021) first established a dynamic regret bound of $\mathcal{O}(d\sqrt{T(1+P_T)})$, which was later sharpened to $\mathcal{O}(\sqrt{dT(1+P_T)})$ by He et al. (2025). However, dynamic gradient-variation regret remains unexplored in two-point BCO, where bandit feedback makes it difficult to exploit gradient variation while simultaneously tracking a time-varying comparator sequence.

Following Zhao et al. (2020), we adopt an online ensemble framework with a two-layer meta-base architecture to accommodate environmental non-stationarity, where multiple base learners explore the environment and a meta learner tracks the best base learner. To handle bandit feedback, we implement the base learners by running Algorithm 1 with different configurations on carefully designed surrogate functions, while the meta learner uses Optimistic-Hedge (Rakhlin & Sridharan, 2013a) to aggregate them.

The key technical step is to use Lemma 2 to reduce the non-consecutive gradient variation $\bar{V}_T$ (2.1) induced by bandit feedback to the standard gradient variation $V_T$ (1.1), while allowing the ensemble analysis to adapt to the unknown path length $P_T$ in non-stationary environments. This leads to *the first* gradient-variation dynamic regret guarantee with unknown path length in the two-point BCO setting. We state the resulting guarantee informally below and defer the complete version (Algorithm 5), the formal theorem, and the proof to Appendix E.1.

**Theorem 6** (Informal). *Under Assumptions 1-3, for convex functions and oblivious comparators $\{\mathbf{v}_t\}_{t=1}^T$, Algorithm 5 in Appendix E.1 achieves the dynamic regret bound*

$$\mathbb{E}[\text{D-REG}_T] \leq \widetilde{\mathcal{O}}\big(\sqrt{d^3(1+P_T+V_T)(1+P_T)}\big),$$

*without prior knowledge of the path length $P_T$.*

Compared with the $\mathcal{O}\big(\sqrt{(1+P_T+V_T)(1+P_T)}\big)$ bound (Zhao et al., 2020) under full-information feedback, Theorem 6 achieves the same dependence on $P_T$ and $V_T$ in expectation, up to logarithmic factors. On the other hand, compared with the worst-case optimal $\mathcal{O}(\sqrt{dT(1+P_T)})$

bound (He et al., 2025), our result incurs an additional factor of $d$ in the dimension dependence. Whether this $\mathcal{O}(d)$ gap can be eliminated for gradient-variation dynamic regret remains an open problem.

### 5.2. Universal Regret Minimization

In this part, we study *universal regret* (Zhao et al., 2025a) for two-point BCO, where the goal is to match the optimal performance for various function types and curvatures *without* prior knowledge of these properties. Specifically, for $\mathcal{F}_{\text{lin}}$ (for linear functions), $\mathcal{F}_{\text{cvx}}$ (for convex functions), and $\mathcal{F}_{\text{sc}}^{\lambda}$ (for $\lambda$-strongly convex functions), a universal online learning algorithm $\mathcal{A}$ aims to attain the following universal regret guarantee:

$$\text{REG}_T(\mathcal{A}, \{f_t\}_{t=1}^T) \lesssim \begin{cases} \text{REG}_T(\mathcal{A}_{\text{sc}}, \mathcal{F}_{\text{sc}}^{\lambda}), \\ \text{REG}_T(\mathcal{A}_{\text{cvx}}, \mathcal{F}_{\text{cvx}}), \\ \text{REG}_T(\mathcal{A}_{\text{lin}}, \mathcal{F}_{\text{lin}}), \end{cases} \quad (5.2)$$

where $\mathcal{A}_{\text{sc}}, \mathcal{A}_{\text{cvx}}, \mathcal{A}_{\text{lin}}$ are the (optimal) algorithms designed for $\mathcal{F}_{\text{sc}}^{\lambda}, \mathcal{F}_{\text{cvx}}$, and $\mathcal{F}_{\text{lin}}$, respectively. That is, $\mathcal{A}$ is supposed to have regret guarantees comparable to those of the optimal algorithms designed for the corresponding function classes. In OCO, universal regret has been extensively studied (van Erven & Koolen, 2016; Wang et al., 2019; Zhang et al., 2022; Yan et al., 2023), and Yan et al. (2024) achieved the optimal universal gradient-variation regret. However, universal gradient-variation regret remains unexplored in two-point BCO, where bandit feedback further complicates the adaptation to unknown curvature.

For adaptation to unknown curvature, we follow Yan et al. (2024) and instantiate an online ensemble algorithm with Algorithm 1 as base learners on carefully designed surrogate functions, while using Optimistic-Adapt-ML-Prod (Wei et al., 2016) as the meta learner on linearized losses; see Algorithm 6 for the complete algorithm.

To analyze this ensemble framework under bandit feedback, we further provide a new Bregman-divergence-based decomposition of the non-consecutive gradient variation $\bar{V}_T$ (2.1) in Lemma 5, with the proof deferred to Appendix E.2.

**Lemma 5.** *Under Assumptions 1, 2, 4, for convex functions, Algorithm 1 satisfies the following guarantee*

$$\mathbb{E}\left[\bar{V}_T\right] \leq 48d^3 L \log(dT) \cdot \mathbb{E}\left[\sum_{t=1}^T \mathcal{D}_{f_t}(\mathbf{w}^{\star}, \mathbf{w}_t)\right] + 12d^3 V_T \log d + \mathcal{O}(1),$$

*where $\mathcal{D}_{\psi}(\mathbf{x}, \mathbf{y}) \triangleq \psi(\mathbf{x}) - \psi(\mathbf{y}) - \langle \nabla\psi(\mathbf{y}), \mathbf{x} - \mathbf{y}\rangle$ denotes the Bregman divergence induced by a strictly convex and differentiable function $\psi : \mathcal{X} \to \mathbb{R}$.*

Compared with $\|\mathbf{w}_t - \mathbf{w}_{t-1}\|^2$ in Lemma 2, $\mathcal{D}_{f_t}(\mathbf{w}^{\star}, \mathbf{w}_t)$ in Lemma 5 is more tractable for universal regret in the general convex case, because the corresponding negative term naturally appears in the linearization of the regret: $f_t(\mathbf{w}_t) - f_t(\mathbf{w}^{\star}) = \langle \nabla f_t(\mathbf{w}_t), \mathbf{w}_t - \mathbf{w}^{\star}\rangle - \mathcal{D}_{f_t}(\mathbf{w}^{\star}, \mathbf{w}_t)$.

Using the online ensemble algorithm and the new decomposition of the non-consecutive gradient variation in Lemma 5, we establish *the first* gradient-variation universal regret for linear, convex, and strongly convex functions in two-point BCO. We give an informal theorem below and defer the formal version (Theorem 10) and its proof to Appendix E.2.

**Theorem 7** (Informal). *Under Assumptions 1, 2, 4, Algorithm 6 in Appendix E.2 achieves the following gradient-variation universal regret:*

$$\mathbb{E}[\text{REG}_T] \leq \begin{cases} \mathcal{O}\left(\frac{d}{\lambda}\log(dV_T)\right), & (\lambda\text{-strongly convex}), \\ \widetilde{\mathcal{O}}\left(\sqrt{d^3 V_T} + d^3\right), & (\text{convex}), \\ \mathcal{O}\left(\sqrt{d^3 V_T}\right), & (\text{linear}). \end{cases}$$

Compared with the curvature-aware bounds, Theorem 7 matches: *(i)* the $\mathcal{O}\left(\sqrt{d^3 V_T}\right)$ bound in the linear setting (Chiang et al., 2013); *(ii)* the $\mathcal{O}\left(\frac{d}{\lambda}\log(dV_T)\right)$ regret in the strongly convex case in Theorem 2; and *(iii)* the $\widetilde{\mathcal{O}}(\sqrt{d^3 V_T})$ regret in the general convex case in Theorem 1, up to an additive $\widetilde{\mathcal{O}}(d^3)$ term. As a byproduct, we also derive the *first* worst-case universal regret bounds for linear, convex, and strongly convex losses in two-point BCO, with the formal statement deferred to Theorem 11.

## 6. Conclusion

In this work, we investigate gradient-variation regret in two-point BCO. By providing a refined analysis of the non-consecutive structure, we achieve $\widetilde{\mathcal{O}}(d^{\frac{3}{2}}\sqrt{V_T})$ and $\mathcal{O}(\frac{d}{\lambda}\log(dV_T))$ for convex and $\lambda$-strongly convex functions, improving the best known results by factors of almost $\sqrt{d}$ and $d$, respectively. We also establish the first gradient-variance and small-loss bounds for two-point BCO, including the first problem-dependent guarantees that recover the minimax optimal $\mathcal{O}(\sqrt{dT})$ regret. Furthermore, we extend our techniques to one-point BLO, achieving the first gradient-variation regret bound $\widetilde{\mathcal{O}}(d^{\frac{7}{2}}\sqrt{V_T})$ for hyper-rectangular domains. We finally validate the effectiveness of our results in more challenging tasks such as dynamic and universal regret minimization.

Several important directions remain open. First, extending our guarantees to adaptive environments is an important direction. Second, it remains unknown whether optimal worst-case regret bounds can be attained for exp-concave functions, another important class in online learning. Finally, our results may also facilitate acceleration for smooth zeroth-order optimization (Nesterov & Spokoiny, 2017).

## Acknowledgements

This research was supported by NSFC (U23A20382, 62576164), the Fundamental and Interdisciplinary Disciplines Breakthrough Plan of the Ministry of Education of China (No. JYB2025XDXM118), and the "111 Center" (No. B26023).

## Impact Statement

This paper presents work whose goal is to advance the field of machine learning. There are many potential societal consequences of our work, none of which we feel must be specifically highlighted here.

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

# A. Properties of Gradient Estimator

In this section, we analyze the properties of the gradient estimator in Eq. (2.2).

Recall that $\alpha_t$ is the largest integer satisfying $0 \leq \alpha_t < t$ and $i_{\alpha_t} = i_t$, where $i_t$ is the direction chosen at round $t$. In addition, recall that $v_t$ is defined as $v_t \triangleq \frac{1}{2\delta}(f_t(\mathbf{w}_t + \delta\mathbf{e}_{i_t}) - f_t(\mathbf{w}_t - \delta\mathbf{e}_{i_t}))$. To ensure $\alpha_t$ is well-defined for the first occurrence of each direction, we introduce a dummy round $0$ in which every direction $i \in [d]$ is considered sampled. We set $f_0 \equiv 0$, so that $v_0 = 0$ and $\nabla f_0(\mathbf{w}_0) = \mathbf{0}$ for any dummy point $\mathbf{w}_0$.

**Lemma 6.** *Under Assumptions 1-3, choosing the exploration parameter $\delta = \frac{1}{2d^2 LTR}$, the gradient estimator and the optimism defined in Eq. (2.2) satisfy the following properties*

*(i)* $\widetilde{g}_{t,i_t} = \widetilde{g}_{\alpha_t+1,i_t} = v_{\alpha_t}$

*(ii)* $|v_t| \leq G$

*(iii)* $\mathbb{E}[v_t^2] \leq \dfrac{2G^2}{d} + \mathcal{O}\left(\frac{1}{d^4 T^2}\right)$

*(iv)* $\mathbb{E}[v_{\alpha_t}^2] \leq \dfrac{2G^2}{d} + \mathcal{O}\left(\frac{1}{d^4 T^2}\right)$

*(v)* $\|\widetilde{\mathbf{g}}_t\|^2 \leq dG^2$

*(vi)* $\|\mathbf{g}_t\|^2 \leq 10d^2 G^2$

*(vii)* $\mathbb{E}\left[\|\mathbf{g}_t\|^2\right] \leq 15dG^2 + \mathcal{O}\left(\frac{1}{d^2 T^2}\right)$

*(viii)* $\|\mathbf{g}_t - \widetilde{\mathbf{g}}_t\|^2 \leq d^2(\nabla_{i_t} f_t(\mathbf{w}_t) - \nabla_{i_t} f_{\alpha_t}(\mathbf{w}_{\alpha_t}))^2 + \mathcal{O}\left(\frac{1}{T}\right)$

*(ix)* $\|\mathbf{g}_t - \widetilde{\mathbf{g}}_t\|^2 \leq 4d^2 G^2$.

*(x)* $\mathbb{E}\left[\|\mathbf{g}_t - \widetilde{\mathbf{g}}_t\|^2\right] \leq 8dG^2 + \mathcal{O}\left(\frac{1}{d^2 T^2}\right)$ *(Lemma 3 in Section 3.2)*

*(xi)* $\|\mathbb{E}_t[\mathbf{g}_t] - \nabla f_t(\mathbf{w}_t)\| \leq \dfrac{\sqrt{d}L\delta}{2} = \mathcal{O}\left(\frac{1}{d^{3/2}T}\right)$

**Proof** To prove property *(i)*, we first provide the intuition behind the construction of $\widetilde{\mathbf{g}}_t$. At each round $t$, only the $i_t$-th coordinate of the optimism $\widetilde{\mathbf{g}}_{t+1}$ is updated using the new directional derivative estimate $v_t$, while all other coordinates remain unchanged. Consequently, for any coordinate $i$, $\widetilde{g}_{t,i}$ always stores the most recent directional derivative estimate $v_s$ from the last round $s < t$ where $i$ was selected. Specifically, for the coordinate $i_t$ chosen at round $t$, its latest update occurred at round $\alpha_t$, which implies $\widetilde{g}_{t,i_t} = v_{\alpha_t}$. Furthermore, since the $i_t$-th direction is not sampled between rounds $\alpha_t + 1$ and $t - 1$, this coordinate stays unchanged, ensuring $\widetilde{g}_{t,i_t} = \widetilde{g}_{\alpha_t+1,i_t}$.

For property *(ii)*, by $G$-Lipschitzness, we have

$$|v_t| = \left|\frac{1}{2\delta}(f_t(\mathbf{w}_t + \delta\mathbf{e}_{i_t}) - f_t(\mathbf{w}_t - \delta\mathbf{e}_{i_t}))\right| \leq G.$$

For property *(iii)*, by $L$-smoothness, for any $i \in [d]$, $t \in [T]$, and $\mathbf{x} \in (1 - \xi)\mathcal{X}$, where $\xi = \delta/r$, we have

$$\begin{aligned}
(f_t(\mathbf{x} + \delta\mathbf{e}_i) - f_t(\mathbf{x})) + (f_t(\mathbf{x}) - f_t(\mathbf{x} - \delta\mathbf{e}_i)) &\leq 2\langle\nabla f_t(\mathbf{x}), \delta\mathbf{e}_i\rangle + L\|\delta\mathbf{e}_i\|^2, \\
(f_t(\mathbf{x} + \delta\mathbf{e}_i) - f_t(\mathbf{x})) + (f_t(\mathbf{x}) - f_t(\mathbf{x} - \delta\mathbf{e}_i)) &\geq 2\langle\nabla f_t(\mathbf{x}), \delta\mathbf{e}_i\rangle - L\|\delta\mathbf{e}_i\|^2,
\end{aligned} \tag{A.1}$$

Therefore, the central-difference bias satisfies

$$\left|\frac{f_t(\mathbf{x} + \delta\mathbf{e}_i) - f_t(\mathbf{x} - \delta\mathbf{e}_i)}{2\delta} - \nabla_i f_t(\mathbf{x})\right| \leq \frac{L\delta}{2}. \tag{A.2}$$

By Eq. (A.1) and the inequality $(a + b)^2 \leq 2a^2 + 2b^2$, we have

$$\mathbb{E}\left[v_t^2\right] = \mathbb{E}\left[\left(\frac{1}{2\delta}(f_t(\mathbf{w}_t + \delta\mathbf{e}_{i_t}) - f_t(\mathbf{w}_t - \delta\mathbf{e}_{i_t}))\right)^2\right] \leq 2\mathbb{E}\left[\langle\nabla f_t(\mathbf{w}_t), \mathbf{e}_{i_t}\rangle^2\right] + \frac{L^2}{2}\delta^2 \leq \frac{2G^2}{d} + \mathcal{O}\left(\frac{1}{d^4 T^2}\right).$$

For property *(iv)*, by Eq. (A.2), we have

$$\mathbb{E}\left[v_{\alpha_t}^2\right] = \mathbb{E}\left[\left(\frac{1}{2\delta}(f_{\alpha_t}(\mathbf{w}_{\alpha_t} + \delta\mathbf{e}_{i_t}) - f_{\alpha_t}(\mathbf{w}_{\alpha_t} - \delta\mathbf{e}_{i_t}))\right)^2\right] \leq 2\mathbb{E}\left[\langle\nabla f_{\alpha_t}(\mathbf{w}_{\alpha_t}), \mathbf{e}_{i_t}\rangle^2\right] + \frac{L^2}{2}\delta^2.$$

Denote $I_t = (i_1, \ldots, i_t)$ for any $t \in [T]$. Taking the expectation over $I_t$ and choosing $\mathbf{c} = \mathbf{0}$ in Lemma 7, we have

$$\mathbb{E}\left[\langle\nabla f_{\alpha_t}(\mathbf{w}_{\alpha_t}), \mathbf{e}_{i_t}\rangle^2\right] = \sum_{s=1}^{t-1}\frac{1}{d^2}\left(1 - \frac{1}{d}\right)^{t-s-1}\mathbb{E}_{I_{s-1}}\left[\|\nabla f_s(\mathbf{w}_s)\|^2\right] \leq \frac{G^2}{d^2}\sum_{s=1}^{t-1}\left(1 - \frac{1}{d}\right)^{t-s-1}$$

$$= \frac{G^2}{d^2} \sum_{k=0}^{t-2} \left(1 - \frac{1}{d}\right)^k \leq \frac{G^2}{d^2} \cdot \frac{1 - \left(1 - \frac{1}{d}\right)^{t-1}}{1 - \left(1 - \frac{1}{d}\right)} \leq \frac{G^2}{d} \left[1 - \left(1 - \frac{1}{d}\right)^{t-1}\right] \leq \frac{G^2}{d},$$

where the first inequality is by the Lipschitz assumption.

For property *(v)*, since for each $i \in [d]$, there exists $0 \leq s_i \leq t - 1$ such that $\widetilde{g}_{t,i} = v_{s_i}$. Thus, we have

$$\|\widetilde{\mathbf{g}}_t\|^2 = \sum_{i=1}^d \widetilde{g}_{t,i}^2 = \sum_{i=1}^d v_{s_i}^2 \leq dG^2,$$

where the last inequality holds due to property *(ii)*.

For property *(vi)*, we have

$$\|\mathbf{g}_t\|^2 = \|d(v_t - v_{\alpha_t})\mathbf{e}_{i_t} + \widetilde{\mathbf{g}}_t\|^2 \leq 2d^2(v_t - v_{\alpha_t})^2 + 2\|\widetilde{\mathbf{g}}_t\|^2 \leq 10d^2G^2,$$

where the last inequality holds due to property *(ii)* and property *(v)*.

For property *(vii)*, we have

$$\mathbb{E}\left[\|\mathbf{g}_t\|^2\right] = \mathbb{E}\left[\|d(v_t - v_{\alpha_t})\mathbf{e}_{i_t} + \widetilde{\mathbf{g}}_t\|^2\right] \leq 3d^2\mathbb{E}\left[v_t^2\right] + 3d^2\mathbb{E}\left[v_{\alpha_t}^2\right] + 3\mathbb{E}\left[\|\widetilde{\mathbf{g}}_t\|^2\right] \leq 15dG^2 + \mathcal{O}\left(\frac{1}{d^2T^2}\right),$$

where the last inequality holds due to property *(iii)*, property *(iv)*, and property *(v)*.

For property *(viii)*, we have

$$\|\mathbf{g}_t - \widetilde{\mathbf{g}}_t\|^2 = \|d(v_t - v_{\alpha_t})\mathbf{e}_{i_t}\|^2 = d^2(v_t - v_{\alpha_t})^2.$$

Define $\varepsilon = |v_t - \langle \nabla f_t(\mathbf{w}_t), \mathbf{e}_{i_t} \rangle| + |v_{\alpha_t} - \langle \nabla f_{\alpha_t}(\mathbf{w}_{\alpha_t}), \mathbf{e}_{i_t} \rangle|$. By Eq. (A.2), we have $\varepsilon \leq L\delta$. Thus, we have

$$(v_t - v_{\alpha_t})^2 \leq \left(|\nabla_{i_t} f_t(\mathbf{w}_t) - \nabla_{i_t} f_{\alpha_t}(\mathbf{w}_{\alpha_t})| + \varepsilon\right)^2$$

$$\leq \left(\nabla_{i_t} f_t(\mathbf{w}_t) - \nabla_{i_t} f_{\alpha_t}(\mathbf{w}_{\alpha_t})\right)^2 + 4\varepsilon G + \varepsilon^2 \leq \left(\nabla_{i_t} f_t(\mathbf{w}_t) - \nabla_{i_t} f_{\alpha_t}(\mathbf{w}_{\alpha_t})\right)^2 + \mathcal{O}\left(\frac{1}{d^2T}\right).$$

For property *(ix)*, we have

$$\|\mathbf{g}_t - \widetilde{\mathbf{g}}_t\|^2 = \|d(v_t - v_{\alpha_t})\mathbf{e}_{i_t}\|^2 \leq 4d^2G^2,$$

where the last inequality holds due to property *(ii)*.

For property *(x)*, we have

$$\mathbb{E}\left[\|\mathbf{g}_t - \widetilde{\mathbf{g}}_t\|^2\right] = \mathbb{E}\left[\|d(v_t - v_{\alpha_t})\mathbf{e}_{i_t}\|^2\right] \leq 2d^2\mathbb{E}\left[v_t^2\right] + 2d^2\mathbb{E}\left[v_{\alpha_t}^2\right] \leq 8dG^2 + \mathcal{O}\left(\frac{1}{d^2T^2}\right),$$

where the last inequality holds due to property *(iii)* and property *(iv)*.

For property *(xi)*, by Eq. (A.2), we have for $i \in [d]$

$$\left|\frac{1}{2\delta}\left(f_t(\mathbf{w}_t + \delta\mathbf{e}_i) - f_t(\mathbf{w}_t - \delta\mathbf{e}_i)\right) - \nabla_i f_t(\mathbf{w}_t)\right| \leq L\delta/2.$$

Since $\mathbf{w}_t$ and $\widetilde{\mathbf{g}}_t$ are fixed under $\mathbb{E}_t[\cdot]$, we have

$$\mathbb{E}_t[\mathbf{g}_t] = \mathbb{E}_t\left[d(v_t - \widetilde{g}_{t,i_t})\mathbf{e}_{i_t} + \widetilde{\mathbf{g}}_t\right] = d\mathbb{E}_t\left[v_t\mathbf{e}_{i_t}\right],$$

where the first step uses $d\mathbb{E}_t[\widetilde{g}_{t,i_t}\mathbf{e}_{i_t}] = \widetilde{\mathbf{g}}_t$ because $i_t$ is uniformly sampled from $[d]$. Thus, we have

$$\|\mathbb{E}_t[\mathbf{g}_t] - \nabla f_t(\mathbf{w}_t)\| \leq \sqrt{d} \max_{i \in [d]} \left|\frac{1}{2\delta}\left(f_t(\mathbf{w}_t + \delta\mathbf{e}_i) - f_t(\mathbf{w}_t - \delta\mathbf{e}_i)\right) - \nabla_i f_t(\mathbf{w}_t)\right| \leq \frac{\sqrt{d}L\delta}{2} = \mathcal{O}\left(\frac{1}{d^{3/2}T}\right).$$

∎

Finally, we prove a useful technical lemma that was used throughout the proof.

**Lemma 7.** *With the same assumptions and parameters as in [Lemma 6](), for any $t \in [T]$, $i \in [d]$ and constant vector $\mathbf{c} \in \mathbb{R}^d$, [Algorithm 1]() ensures that*

$$\mathbb{E}\left[\langle \nabla f_{\alpha_t}(\mathbf{w}_{\alpha_t}) - \mathbf{c}, \mathbf{e}_{i_t}\rangle^2\right] \leq \sum_{s=1}^{t-1} \frac{1}{d^2}\left(1-\frac{1}{d}\right)^{t-s-1} \mathbb{E}_{I_{s-1}}\left[\|\nabla f_s(\mathbf{w}_s) - \mathbf{c}\|^2\right] + \frac{1}{d}\left(1-\frac{1}{d}\right)^{t-1}\|\mathbf{c}\|^2,$$

*where $I_{s-1} \triangleq (i_1, \ldots, i_{s-1})$ is the history of previously selected coordinates.*

**Proof** Denoting by $\mathbb{I}(\cdot)$ the indicator function, we have

$$\mathbb{E}\left[\langle \nabla f_{\alpha_t}(\mathbf{w}_{\alpha_t}) - \mathbf{c}, \mathbf{e}_{i_t}\rangle^2\right] = \mathbb{E}_{I_t}\left[\sum_{s=0}^{t-1}\sum_{i=1}^{d} \mathbb{I}(i_t = i, \alpha_t = s)\langle \nabla f_s(\mathbf{w}_s) - \mathbf{c}, \mathbf{e}_i\rangle^2\right]$$

$$= \sum_{s=0}^{t-1}\sum_{i=1}^{d}\mathbb{E}_{I_t}\left[\mathbb{I}(i_t = i, \alpha_t = s)\langle \nabla f_s(\mathbf{w}_s) - \mathbf{c}, \mathbf{e}_i\rangle^2\right]$$

$$= \sum_{s=0}^{t-1}\sum_{i=1}^{d}\mathbb{E}_{I_{s-1}}\left[\mathbb{E}_{i_s,\ldots,i_t}\left[\mathbb{I}(i_t = i, \alpha_t = s)\langle \nabla f_s(\mathbf{w}_s) - \mathbf{c}, \mathbf{e}_i\rangle^2 \mid I_{s-1}\right]\right]$$

$$= \sum_{s=0}^{t-1}\sum_{i=1}^{d}\mathbb{P}(i_t = i, \alpha_t = s)\mathbb{E}_{I_{s-1}}\left[\langle \nabla f_s(\mathbf{w}_s) - \mathbf{c}, \mathbf{e}_i\rangle^2\right]$$

$$= \sum_{s=1}^{t-1}\sum_{i=1}^{d}\left[\frac{1}{d^2}\left(1-\frac{1}{d}\right)^{t-s-1}\right]\mathbb{E}_{I_{s-1}}\left[\langle \nabla f_s(\mathbf{w}_s) - \mathbf{c}, \mathbf{e}_i\rangle^2\right] + \frac{1}{d}\left(1-\frac{1}{d}\right)^{t-1}\|\nabla f_0(\mathbf{w}_0) - \mathbf{c}\|^2$$

$$= \sum_{s=1}^{t-1}\frac{1}{d^2}\left(1-\frac{1}{d}\right)^{t-s-1}\mathbb{E}_{I_{s-1}}\left[\sum_{i=1}^{d}\langle \nabla f_s(\mathbf{w}_s) - \mathbf{c}, \mathbf{e}_i\rangle^2\right] + \frac{1}{d}\left(1-\frac{1}{d}\right)^{t-1}\|\mathbf{c}\|^2,$$

where the first step uses the law of total expectation to sum over all possible realizations of the random coordinate $i_t$ and the last update time $\alpha_t$, the third step uses the tower property $\mathbb{E}[\cdot] = \mathbb{E}_{I_{s-1}}[\mathbb{E}[\cdot \mid I_{s-1}]]$, the fourth step follows because $\mathbb{I}(i_t = i, \alpha_t = s)$ only depends on $i_s, \ldots, i_t$ and, given $s$, $\mathbf{w}_s$ only depends on $I_{s-1}$, the fifth step substitutes the joint probability $\mathbb{P}(i_t = i, \alpha_t = s) = \frac{1}{d^2}(1-\frac{1}{d})^{t-s-1}$ for $s > 0$, which is derived from the independent uniform sampling of coordinates, and the last step uses $\nabla f_0(\mathbf{w}_0) \triangleq \mathbf{0}$. ∎

## B. Analysis of Non-Consecutive Sampling Gap

In this section, we analyze the stochastic behavior of the non-consecutive sampling gap $\rho_{t,i}$. At each time step $t \in [T]$, a sample $i_t$ is drawn independently and uniformly from $[d]$. Then, the gap $\rho_{t,i}$ is defined as:

$$\rho_{t,i} \triangleq \tau_2 - \tau_1, \text{ where } \tau_1 \triangleq \max\{0 \leq \tau < t \mid i_\tau = i\}, \text{ and } \tau_2 \triangleq \min\{t \leq \tau \leq T \mid i_\tau = i\}. \tag{B.1}$$

For completeness, we set $\tau_2 = T + 1$ if coordinate $i$ is never sampled at or after time $t$ and $\tau_1 = 0$ if coordinate $i$ is never sampled before time $t$. Intuitively, $\rho_{t,i}$ quantifies the duration between the most recent sampling of coordinate $i$ before time $t$ and its next sampling at or after time $t$.

We now establish the connection between $\rho_{t,i}$, the geometric distribution, and the Coupon Collector's Problem (CCP).

**Relation to Geometric Distribution.** To start, we analyze $\rho_{t,i} + 1$ by decomposing it into two components, $t - \tau_1$ and $\tau_2 - t + 1$, which represent the waiting times of two independent sampling processes:

- **Backward Search:** Starting from time $t$, we search backward through the sequence $X_{t-1}, X_{t-2}, \ldots$ until coordinate $i$ is first encountered at index $\tau_1$. The search stops exactly at this hit, making $t - \tau_1$ the search duration.

- **Forward Search:** Similarly, we search forward from time $t$ through $X_t, X_{t+1}, \ldots$ until the next occurrence of $i$ at index $\tau_2$. The forward duration is $\tau_2 - t + 1$.

Since each $i_\tau$ is sampled uniformly from $[d]$, the waiting times $t - \tau_1$ and $\tau_2 - t + 1$ are independent truncated geometric random variables with success probability $p = 1/d$. Specifically, $t - \tau_1$ is truncated at $t$, and $\tau_2 - t + 1$ is truncated at $T - t + 2$, with the last value corresponding to no future occurrence. Consequently, the analysis of $\rho_{t,i}$ reduces to studying the sum of two independent geometric random variables subject to boundary constraints.

**Relation to Coupon Collector's Problem.** While the geometric distribution characterizes the gap for a fixed coordinate $i$, the term $\max_{i \in [d]} \rho_{t,i}$ is naturally captured by the Coupon Collector's Problem. In the standard CCP, a collector seeks to obtain all $d$ distinct types of coupons, where each draw yields type $i \in [d]$ uniformly at random with probability $1/d$. The primary object of interest is the *stopping time*, defined as the total number of draws required to complete the collection.

In our context, $\max_{i \in [d]}(t - \tau_1)$ corresponds to the time required to observe every coordinate at least once when searching backward from $t$. Similarly, $\max_{i \in [d]}(\tau_2 - t + 1)$ represents the time required to encounter every coordinate when searching forward. Thus, both terms behave as the waiting time in a standard CCP with $d$ bins truncated by the available time horizon.

The following lemma summarizes the key properties of these distributions:

**Lemma 8.** *Consider a process where in each round $t = 1, 2, \ldots$, a number $X_t$ is sampled uniformly and independently from $[n]$.*

- *For any specific $i \in [n]$, let $T_i = \min\{t \mid X_t = i\}$ be the first time $i$ is sampled. Then $\mathbb{E}[T_i] = n$.*

- *Let $T_{all} = \max_{i \in [n]} T_i$ be the time when the last remaining number in $[n]$ is sampled (i.e., the time to collect all numbers). Then $\mathbb{E}[T_{all}] \leq 2n \log n$ (for $n \geq 3$).*

By Lemma 8, setting $n = d$, we have directly the following two properties:

$$\mathbb{E}[\rho_{t,i}] \leq 2\mathbb{E}[T_i] - 1 \leq 2d. \tag{B.2}$$

$$\mathbb{E}\left[\max_{i \in [d]} \rho_{t,i}\right] \leq 2\mathbb{E}\left[\max_{i \in [d]} T_i\right] - 1 \leq 4d \log d. \tag{B.3}$$

Finally, we provide the proof of Lemma 8.

**Proof** To start with, we prove $\mathbb{E}[T_i] = n$ using the properties of the geometric distribution.

Fix a specific target number $i \in [n]$. In any round $t$, the probability of sampling $i$ is $p = \frac{1}{n}$. Since the samples $X_t$ are drawn independently, the waiting time $T_i$ for the first occurrence of $i$ follows a geometric distribution with success probability $p$. The expectation of a geometric random variable with parameter $p$ is $1/p$. Therefore:

$$\mathbb{E}[T_i] = \frac{1}{p} = \frac{1}{1/n} = n.$$

Next, we prove the second property, which is equivalent to the classical *Coupon Collector's Problem*. We decompose the total waiting time $T_{all}$ into phases. Let $S_k$ denote the number of *additional* rounds required to collect the $k$-th distinct number, given that $k - 1$ distinct numbers have already been collected. The total time is the sum of the times in each phase:

$$T_{all} = \sum_{k=1}^{n} S_k.$$

Consider the phase where $k - 1$ distinct numbers have already been found. There are $n - (k - 1)$ "unseen" numbers remaining in the pool of size $n$. The probability $p_k$ of drawing a *new* number in any single trial during this phase is:

$$p_k = \frac{n - (k - 1)}{n}.$$

The number of trials $S_k$ in this phase follows a geometric distribution with parameter $p_k$. Thus, its expected value is:

$$\mathbb{E}[S_k] = \frac{1}{p_k} = \frac{n}{n - k + 1}.$$

By the linearity of expectation, the total expected time is:

$$\mathbb{E}[T_{\text{all}}] = \sum_{k=1}^{n} \mathbb{E}[S_k] = \sum_{k=1}^{n} \frac{n}{n-k+1} = n \sum_{j=1}^{n} \frac{1}{j}, \tag{B.4}$$

where the last step follows from the change of index $j = n - k + 1$. The sum $H_n = \sum_{j=1}^{n} \frac{1}{j}$ is the $n$-th Harmonic number, which satisfies the bound $H_n \leq \log n + 1$ for all $n \geq 1$. Substituting this into Eq. (B.4):

$$\mathbb{E}[T_{\text{all}}] \leq n(\log n + 1) = n \log n + n \leq 2n \log n,$$

which completes the proof. ∎

# C. Omitted Proofs in Section 3

In this section, we provide the detailed proofs for the results presented in Section 3.

## C.1. Proof of Lemma 1

**Proof** By property *(viii)* of Lemma 6, we have

$$\bar{V}_T = \sum_{t=1}^{T} \|\mathbf{g}_t - \widetilde{\mathbf{g}}_t\|^2 \leq d^2 \sum_{t=1}^{T} (\nabla_{i_t} f_t(\mathbf{w}_t) - \nabla_{i_t} f_{\alpha_t}(\mathbf{w}_{\alpha_t}))^2 + \mathcal{O}(1). \tag{C.1}$$

Then, we bound $(\nabla_{i_t} f_t(\mathbf{w}_t) - \nabla_{i_t} f_{\alpha_t}(\mathbf{w}_{\alpha_t}))^2$ as follows:

$$(\nabla_{i_t} f_t(\mathbf{w}_t) - \nabla_{i_t} f_{\alpha_t}(\mathbf{w}_{\alpha_t}))^2 = \left( \sum_{s=\alpha_t+1}^{t} \nabla_{i_t} f_s(\mathbf{w}_s) - \nabla_{i_t} f_{s-1}(\mathbf{w}_{s-1}) \right)^2$$

$$\leq (t - \alpha_t) \sum_{s=\alpha_t+1}^{t} (\nabla_{i_t} f_s(\mathbf{w}_s) - \nabla_{i_t} f_{s-1}(\mathbf{w}_{s-1}))^2 \leq \sum_{s=\alpha_t+1}^{t} \rho_{s,i_t} (\nabla_{i_t} f_s(\mathbf{w}_s) - \nabla_{i_t} f_{s-1}(\mathbf{w}_{s-1}))^2$$

$$\leq 2 \sum_{s=\alpha_t+1}^{t} \rho_{s,i_t} (\nabla_{i_t} f_s(\mathbf{w}_s) - \nabla_{i_t} f_s(\mathbf{w}_{s-1}))^2 + 2 \sum_{s=\alpha_t+1}^{t} \rho_{s,i_t} (\nabla_{i_t} f_s(\mathbf{w}_{s-1}) - \nabla_{i_t} f_{s-1}(\mathbf{w}_{s-1}))^2,$$

where the second step is by the Cauchy-Schwarz inequality and the third step uses $t - \alpha_t \leq \rho_{s,i_t}$ for $\alpha_t < s \leq t$ according to the definition of the non-consecutive sampling gap $\rho_{s,i}$ in Eq. (B.1).

Summing over $t = 1, \ldots, T$, we have

$$\sum_{t=1}^{T} (\nabla_{i_t} f_t(\mathbf{w}_t) - \nabla_{i_t} f_{\alpha_t}(\mathbf{w}_{\alpha_t}))^2$$

$$\leq 2 \sum_{t=1}^{T} \sum_{s=\alpha_t+1}^{t} \rho_{s,i_t} \left( (\nabla_{i_t} f_s(\mathbf{w}_s) - \nabla_{i_t} f_s(\mathbf{w}_{s-1}))^2 + (\nabla_{i_t} f_s(\mathbf{w}_{s-1}) - \nabla_{i_t} f_{s-1}(\mathbf{w}_{s-1}))^2 \right)$$

$$= 2 \sum_{i=1}^{d} \sum_{\substack{t:i_t=i \\ 1 \leq t \leq T}} \sum_{s=\alpha_t+1}^{t} \rho_{s,i} \left( (\nabla_i f_s(\mathbf{w}_s) - \nabla_i f_s(\mathbf{w}_{s-1}))^2 + (\nabla_i f_s(\mathbf{w}_{s-1}) - \nabla_i f_{s-1}(\mathbf{w}_{s-1}))^2 \right)$$

$$\leq 2 \sum_{i=1}^{d} \sum_{s=1}^{T} \rho_{s,i} (\nabla_i f_s(\mathbf{w}_{s-1}) - \nabla_i f_{s-1}(\mathbf{w}_{s-1}))^2 + 2 \sum_{i=1}^{d} \sum_{s=1}^{T} \rho_{s,i} (\nabla_i f_s(\mathbf{w}_s) - \nabla_i f_s(\mathbf{w}_{s-1}))^2$$

$$= 2 \sum_{t=1}^{T} \sum_{i=1}^{d} \rho_{t,i} (\nabla_i f_t(\mathbf{w}_{t-1}) - \nabla_i f_{t-1}(\mathbf{w}_{t-1}))^2 + 2 \sum_{t=1}^{T} \sum_{i=1}^{d} \rho_{t,i} (\nabla_i f_t(\mathbf{w}_t) - \nabla_i f_t(\mathbf{w}_{t-1}))^2, \tag{C.2}$$

where the second step follows from changing the order of summation, based on the observation that for each fixed $s$ and $i$, there is at most one $t$ such that $i_t = i$ and $s \in (\alpha_t, t]$, and the last step is obtained by changing the order of summation over $i$ and $s$, followed by relabeling the index $s$ as $t$.

Plugging Eq. (C.2) back to Eq. (C.1), we finish the proof. ∎

## C.2. Proof of Lemma 2

**Proof** Following the proof of Lemma 1, we have

$$
\sum_{t=1}^{T} (\nabla_{i_t} f_t(\mathbf{w}_t) - \nabla_{i_t} f_{\alpha_t}(\mathbf{w}_{\alpha_t}))^2
$$

$$
\leq 2\underbrace{\sum_{t=1}^{T}\sum_{i=1}^{d} \rho_{t,i} \left(\nabla_i f_t(\mathbf{w}_{t-1}) - \nabla_i f_{t-1}(\mathbf{w}_{t-1})\right)^2}_{\text{TERM (A)}} + 2\underbrace{\sum_{t=1}^{T}\sum_{i=1}^{d} \rho_{t,i} \left(\nabla_i f_t(\mathbf{w}_t) - \nabla_i f_t(\mathbf{w}_{t-1})\right)^2}_{\text{TERM (B)}}
$$

For TERM (A), we have

$$
\mathbb{E}[\text{TERM (A)}] \leq \mathbb{E}\left[\sum_{t=1}^{T}\sum_{i=1}^{d} \left(\max_{i\in[d]} \rho_{t,i}\right) \left(\nabla_i f_t(\mathbf{w}_{t-1}) - \nabla_i f_{t-1}(\mathbf{w}_{t-1})\right)^2\right]
$$

$$
= \mathbb{E}\left[\sum_{t=1}^{T} \left(\max_{i\in[d]} \rho_{t,i}\right) \|\nabla f_t(\mathbf{w}_{t-1}) - \nabla f_{t-1}(\mathbf{w}_{t-1})\|^2\right] \leq \mathbb{E}\left[\sum_{t=1}^{T} \left(\max_{i\in[d]} \rho_{t,i}\right) \sup_{\mathbf{x}\in\mathcal{X}} \|\nabla f_t(\mathbf{x}) - \nabla f_{t-1}(\mathbf{x})\|^2\right]
$$

$$
= \sum_{t=1}^{T} \mathbb{E}\left[\max_{i\in[d]} \rho_{t,i}\right] \sup_{\mathbf{x}\in\mathcal{X}} \|\nabla f_t(\mathbf{x}) - \nabla f_{t-1}(\mathbf{x})\|^2 \leq 4d \log d V_T. \tag{C.3}
$$

where the first and third steps are by taking the maximum, the second step is obtained by summing across all coordinates $i \in [d]$, and the last step follows from the property in Eq. (B.3).

For TERM (B), we have

$$
\mathbb{E}[\text{TERM (B)}] \leq 4dL^2 \log(dT)\mathbb{E}\left[\sum_{t=1}^{T} \|\mathbf{w}_t - \mathbf{w}_{t-1}\|^2\right] + \mathcal{O}(1). \tag{C.4}
$$

The proof of Eq. (C.4) is as follows. Let $\bar{\rho} = 4d \log(dT)$ and let $Q$ denote the bad event that $\rho_{t,i} > \bar{\rho}$ for some $t \in [T]$ and $i \in [d]$. If $Q$ occurs, then for some coordinate $i$, there is a block of at least $\lfloor \bar{\rho} \rfloor$ consecutive rounds in which coordinate $i$ is not sampled. Therefore, by a union bound,

$$
\mathbb{P}(Q) \leq d(T+1)\left(1 - \frac{1}{d}\right)^{\lfloor \bar{\rho} \rfloor} \leq d(T+1)\exp\left(-\frac{\bar{\rho}-1}{d}\right) \leq \frac{4}{d^3 T^3}. \tag{C.5}
$$

Using the smoothness of $f_t$ and the deterministic bound $\rho_{t,i} \leq T+1$, we have

$$
\mathbb{E}[\text{TERM (B)}\mathbb{I}(Q)] \leq \mathbb{P}(Q) \cdot (T+1) \sup_{\mathbf{w}_0,\dots,\mathbf{w}_T\in\mathcal{X}} \sum_{t=1}^{T} \|\nabla f_t(\mathbf{w}_t) - \nabla f_t(\mathbf{w}_{t-1})\|^2
$$

$$
\leq \frac{4}{d^3 T^3} \cdot (T+1) \cdot 4L^2 R^2 T \leq \mathcal{O}(1),
$$

where $\mathbb{I}(\cdot)$ denotes the indicator function and the last step uses $\|\mathbf{w}_t - \mathbf{w}_{t-1}\| \leq 2R$.

When $\neg Q$ occurs, we have $\rho_{t,i} \leq \bar{\rho}$ for all $t \in [T]$ and $i \in [d]$. Therefore,

$$
\text{TERM (B)}\mathbb{I}(\neg Q) \leq \bar{\rho}\sum_{t=1}^{T}\sum_{i=1}^{d} \left(\nabla_i f_t(\mathbf{w}_t) - \nabla_i f_t(\mathbf{w}_{t-1})\right)^2
$$

$$\leq 4dL^2 \log(dT) \sum_{t=1}^{T} \|\mathbf{w}_t - \mathbf{w}_{t-1}\|^2.$$

Combining the above two inequalities for the events $Q$ and $\neg Q$, we obtain Eq. (C.4). Finally, combining Eq. (C.3) and Eq. (C.4) with Eq. (C.1), we complete the proof. ∎

### C.3. Proof of Theorem 1

In this part, we provide the proof of Theorem 1. To begin with, we first restate a useful lemma from Chiang et al. (2013) for self-containedness. For simplicity, we denote $\mathbf{w}^\star \in \arg\min_{\mathbf{w} \in (1-\xi)\mathcal{X}} \sum_{t=1}^{T} f_t(\mathbf{w})$.

**Lemma 9** (Lemma 2 of Chiang et al. (2013)). *Under Assumptions 1-2, for convex functions, Algorithm 1 enjoys*

$$\sum_{t=1}^{T} \frac{1}{2} \left(f_t(\mathbf{x}_t) + f_t(\mathbf{x}'_t)\right) - \min_{\mathbf{x} \in \mathcal{X}} \sum_{t=1}^{T} f_t(\mathbf{x}) \leq \sum_{t=1}^{T} f_t(\mathbf{w}_t) - \sum_{t=1}^{T} f_t(\mathbf{w}^\star) + \mathcal{O}(1).$$

**Proof** [of Theorem 1] First, using the standard analysis for OOGD (Chiang et al., 2012), we have

$$\mathbb{E}[\text{REG}_T] \leq \mathbb{E}\left[\sum_{t=1}^{T} f_t(\mathbf{w}_t) - \sum_{t=1}^{T} f_t(\mathbf{w}^\star)\right] + \mathcal{O}(1) \qquad \text{(by Lemma 9)}$$

$$\leq \mathbb{E}\left[\sum_{t=1}^{T} \langle \nabla f_t(\mathbf{w}_t), \mathbf{w}_t - \mathbf{w}^\star \rangle\right] + \mathcal{O}(1) \leq \mathbb{E}\left[\sum_{t=1}^{T} \langle \mathbf{g}_t, \mathbf{w}_t - \mathbf{w}^\star \rangle\right] + \mathcal{O}(1)$$

$$\leq \underbrace{\mathbb{E}\left[\sum_{t=1}^{T} \eta_t \|\mathbf{g}_t - \widetilde{\mathbf{g}}_t\|^2\right]}_{\text{TERM (A)}} + \underbrace{\mathbb{E}\left[\sum_{t=1}^{T} \frac{1}{2\eta_t} \left(\|\mathbf{w}^\star - \widehat{\mathbf{w}}_t\|^2 - \|\mathbf{w}^\star - \widehat{\mathbf{w}}_{t+1}\|^2\right)\right]}_{\text{TERM (B)}}$$

$$- \underbrace{\mathbb{E}\left[\sum_{t=1}^{T} \frac{1}{2\eta_t} \left(\|\widehat{\mathbf{w}}_{t+1} - \mathbf{w}_t\|^2 + \|\mathbf{w}_t - \widehat{\mathbf{w}}_t\|^2\right)\right]}_{\text{TERM (C)}} + \mathcal{O}(1), \qquad (\text{C.6})$$

where the second step is by convexity and the third step is by property *(xi)* in Lemma 6.

**Problem-Dependent Regret.** For TERM (A), we have the following bound:

$$\text{TERM (A)} \leq R\mathbb{E}\left[\sum_{t=1}^{T} \frac{\|\mathbf{g}_t - \widetilde{\mathbf{g}}_t\|^2}{\sqrt{1152 d^3 R^4 L^2 \log(dT) + \bar{V}_{t-1}}}\right]$$

$$\leq 4R\mathbb{E}\left[\sqrt{1152 d^3 R^4 L^2 \log(dT) + \sum_{t=1}^{T} \|\mathbf{g}_t - \widetilde{\mathbf{g}}_t\|^2} + \max_{t \in [T]} \frac{\|\mathbf{g}_t - \widetilde{\mathbf{g}}_t\|^2}{\sqrt{1152 d^3 R^4 L^2 \log(dT)}}\right]$$

$$\leq 4R\mathbb{E}\left[\sqrt{1152 d^3 R^4 L^2 \log(dT) + \bar{V}_T}\right] + \mathcal{O}\left(\sqrt{d}\right), \qquad \text{(by Lemma 6)}$$

where the first step is by the definition of $\eta_t$ and the second inequality is by choosing $a_s = \frac{\|\mathbf{g}_s - \widetilde{\mathbf{g}}_s\|^2}{1152 d^3 R^4 L^2 \log(dT)}$ in Lemma 22. For TERM (B), we have

$$\text{TERM (B)} \leq \mathbb{E}\left[\frac{\|\mathbf{w}^\star - \widehat{\mathbf{w}}_1\|^2}{2\eta_1} + \sum_{t=2}^{T} \left(\frac{1}{2\eta_t} - \frac{1}{2\eta_{t-1}}\right) \|\mathbf{w}^\star - \widehat{\mathbf{w}}_t\|^2\right] \leq \mathbb{E}\left[\frac{2R^2}{2\eta_1} + \sum_{t=2}^{T} \left(\frac{2R^2}{\eta_t} - \frac{2R^2}{\eta_{t-1}}\right)\right]$$

$$\leq \mathbb{E}\left[\frac{2R^2}{\eta_1} + \frac{2R^2}{\eta_T}\right] = 2R\mathbb{E}\left[\sqrt{1152 d^3 R^4 L^2 \log(dT) + \bar{V}_T}\right] + \mathcal{O}(\sqrt{d^3 \log(dT)}),$$

where the first step is by changing the order of summation and the second step is by the boundedness.

For TERM (C), we have

$$\text{TERM (C)} \geq \mathbb{E}\left[\sum_{t=2}^{T} \frac{1}{2\eta_{t-1}} \|\mathbf{w}_t - \widehat{\mathbf{w}}_t\|^2 + \|\mathbf{w}_{t-1} - \widehat{\mathbf{w}}_t\|^2\right] \geq \mathbb{E}\left[\sum_{t=2}^{T} \frac{1}{4\eta_{t-1}} \|\mathbf{w}_t - \mathbf{w}_{t-1}\|^2\right].$$

Thus, we have

$$\mathbb{E}[\text{REG}_T] \leq \mathbb{E}\left[6R\sqrt{1152d^3R^4L^2\log(dT) + \bar{V}_T} - \sum_{t=2}^{T} \frac{1}{4\eta_{t-1}} \|\mathbf{w}_t - \mathbf{w}_{t-1}\|^2\right] + \mathcal{O}\left(\sqrt{d^3\log(dT)}\right)$$

$$\leq 6R\sqrt{1152d^3R^4L^2\log(dT) + \mathbb{E}\left[\bar{V}_T\right]} - \mathbb{E}\left[\sum_{t=2}^{T} \frac{1}{4\eta_{t-1}} \|\mathbf{w}_t - \mathbf{w}_{t-1}\|^2\right] + \mathcal{O}\left(\sqrt{d^3\log(dT)}\right). \quad \text{(C.7)}$$

where the last inequality is by Jensen's inequality.

Using Lemma 2, we have

$$\sqrt{1152d^3R^4L^2\log(dT) + \mathbb{E}\left[\bar{V}_T\right]} \leq \sqrt{1152d^3R^4L^2\log(dT) + 2d^2\mathbb{E}\left[\sum_{t=1}^{T}\sum_{i=1}^{d} \rho_{t,i}\left(\nabla_i f_t(\mathbf{w}_{t-1}) - \nabla_i f_{t-1}(\mathbf{w}_{t-1})\right)^2\right]}$$

$$+ \sqrt{2d^2\mathbb{E}\left[\sum_{t=1}^{T}\sum_{i=1}^{d} \rho_{t,i}\left(\nabla_i f_t(\mathbf{w}_t) - \nabla_i f_t(\mathbf{w}_{t-1})\right)^2\right]}$$

$$\leq \sqrt{1152d^3R^4L^2\log(dT) + 8d^3V_T\log d} + \sqrt{8d^3L^2\log(dT)\mathbb{E}\left[\sum_{t=1}^{T} \|\mathbf{w}_t - \mathbf{w}_{t-1}\|^2\right]} + \mathcal{O}(1).$$

Plugging it back to (C.7), we have

$$\mathbb{E}[\text{REG}_T] \leq \mathbb{E}\left[6R\sqrt{1152d^3R^4L^2\log(dT) + \bar{V}_T} - \sum_{t=2}^{T} \frac{1}{4\eta_{t-1}} \|\mathbf{w}_t - \mathbf{w}_{t-1}\|^2\right] + \mathcal{O}\left(\sqrt{d^3\log(dT)}\right)$$

$$\leq 6R\sqrt{1152d^3\left(R^4L^2\log(dT) + 8V_T\log d\right)} + 6R\sqrt{8d^3L^2\log(dT)\mathbb{E}\left[\sum_{t=1}^{T} \|\mathbf{w}_t - \mathbf{w}_{t-1}\|^2\right]}$$

$$- \frac{RL\sqrt{1152d^3\log(dT)}}{4}\mathbb{E}\left[\sum_{t=2}^{T} \|\mathbf{w}_t - \mathbf{w}_{t-1}\|^2\right] + \mathcal{O}\left(\sqrt{d^3\log(dT)}\right)$$

$$\leq \mathcal{O}\left(\sqrt{d^3\log(dT) + d^3V_T\log d}\right),$$

where the last inequality is by AM-GM inequality.

**Problem-Independent Regret.** Using property *(x)* of Lemma 6, we have $\mathbb{E}\left[\bar{V}_T\right] \leq 8dG^2T + \mathcal{O}(1)$. Plugging it back to (C.7), we have

$$\mathbb{E}[\text{REG}_T] \leq 6R\sqrt{1152d^3R^4L^2\log(dT) + 8dG^2T} + \mathcal{O}\left(\sqrt{d}\right) = \mathcal{O}\left(\sqrt{d^3\log(dT) + dT}\right),$$

which completes the proof. ∎

## C.4. Proof of Theorem 2

**Proof** To start, by the standard OOGD analysis, e.g., Yan et al. (2023, Lemma 12), we have

$$\mathbb{E}[\text{REG}_T] \leq \mathbb{E}\left[\sum_{t=1}^{T} f_t(\mathbf{w}_t) - \sum_{t=1}^{T} f_t(\mathbf{w}^\star)\right] + \mathcal{O}(1) \leq \mathbb{E}\left[\sum_{t=1}^{T} \langle \nabla f_t(\mathbf{w}_t), \mathbf{w}_t - \mathbf{w}^\star \rangle - \frac{\lambda}{2}\|\mathbf{w}_t - \mathbf{w}^\star\|^2\right] + \mathcal{O}(1)$$

$$\leq \mathbb{E}\left[\sum_{t=1}^{T} \langle \mathbf{g}_t, \mathbf{w}_t - \mathbf{w}^\star \rangle - \frac{\lambda}{2}\|\mathbf{w}_t - \mathbf{w}^\star\|^2\right] + \mathcal{O}(1)$$

$$\leq 2\underbrace{\mathbb{E}\left[\sum_{t=1}^{T} \eta_t \|\mathbf{g}_t - \widetilde{\mathbf{g}}_t\|^2\right]}_{\text{TERM (A)}} - \underbrace{\mathbb{E}\left[\sum_{t=1}^{T} \frac{1}{2\eta_t}\left(\|\widehat{\mathbf{w}}_{t+1} - \mathbf{w}_t\|^2 + \|\mathbf{w}_t - \widehat{\mathbf{w}}_t\|^2\right)\right]}_{\text{TERM (B)}} + \mathcal{O}(1), \tag{C.8}$$

where the third inequality is by property *(xi)* in Lemma 6.

For TERM (A), we have

$$\text{TERM (A)} \leq \mathcal{O}\left(\frac{\max_{t\in[T]} \mathbb{E}\left[\|\mathbf{g}_t - \widetilde{\mathbf{g}}_t\|^2\right]}{\lambda} \log\left(\lambda \mathbb{E}\left[\sum_{t=1}^{T} \|\mathbf{g}_t - \widetilde{\mathbf{g}}_t\|^2\right]\right)\right) \tag{C.9}$$

$$\leq \mathcal{O}\left(\frac{d}{\lambda} \log\left(\lambda \mathbb{E}[\bar{V}_T]\right)\right)$$

$$\leq \mathcal{O}\left(\frac{d}{\lambda} \log\left(\lambda d^3 \log d V_T + \lambda d^3 \log(dT) \mathbb{E}\left[\sum_{t=1}^{T} \|\mathbf{w}_t - \mathbf{w}_{t-1}\|^2\right]\right)\right),$$

where the first step is by choosing $a_t = \mathbb{E}\left[\|\mathbf{g}_t - \widetilde{\mathbf{g}}_t\|^2\right]$ in Lemma 23 for any $t \in [T]$, the second step is by property *(x)* in Lemma 6, and the last step is by Lemma 2.

Consequently, we obtain the following regret bound:

$$\mathbb{E}[\text{REG}_T] \leq \mathcal{O}\left(\frac{d}{\lambda} \log\left(\lambda d^3 V_T \log d + \lambda d^3 \log(dT) \mathbb{E}\left[\sum_{t=1}^{T} \|\mathbf{w}_t - \mathbf{w}_{t-1}\|^2\right]\right)\right) - \text{TERM (B)}.$$

Following the analysis in Chiang et al. (2013, Theorem 16), we analyze two cases: *(i)* $\mathbb{E}[\sum_{t=1}^{T} \|\mathbf{w}_t - \mathbf{w}_{t-1}\|^2] = \mathcal{O}(1)$; and *(ii)* $\mathbb{E}[\sum_{t=1}^{T} \|\mathbf{w}_t - \mathbf{w}_{t-1}\|^2] = \omega(1)$. For the first case,

$$\mathbb{E}[\text{REG}_T] \leq \mathcal{O}\left(\frac{d}{\lambda} \log\left(\lambda d^3 \log d V_T + \lambda d^3 \log(dT)\right)\right) = \mathcal{O}\left(\frac{d}{\lambda} \log\left(dV_T\right)\right),$$

where we omit the $\mathcal{O}(\log \log T)$ term. For the second case, we have

$$\mathbb{E}[\text{REG}_T] \leq \mathcal{O}\left(\frac{d}{\lambda} \log\left(\lambda d^3 \left(V_T \log d + \log(dT)\right)\right)\right) + \mathcal{O}\left(\frac{d}{\lambda} \log\left(\mathbb{E}\left[\sum_{t=1}^{T} \|\mathbf{w}_t - \mathbf{w}_{t-1}\|^2\right]\right)\right) - \text{TERM (B)}$$

$$\leq \mathcal{O}\left(\frac{d}{\lambda} \log\left(dV_T\right)\right) + \mathcal{O}\left(\frac{d}{\lambda} \log\left(\mathbb{E}\left[\sum_{t=1}^{T} \|\mathbf{w}_t - \mathbf{w}_{t-1}\|^2\right]\right)\right) - \frac{\lambda}{4} \mathbb{E}\left[\sum_{t=1}^{T} \|\mathbf{w}_t - \mathbf{w}_{t-1}\|^2\right]$$

$$\leq \mathcal{O}\left(\frac{d}{\lambda} \log\left(dV_T\right)\right),$$

where the first and third inequalities use $\ln(1+x) \leq x$ for $x > 0$, and we omit the $\mathcal{O}(\log \log T)$ term. ∎

---

**Algorithm 3** Expected Gradient Descent with Two Queries

---

**Input:** Learning rates $\eta_t$, exploration parameter $\delta$ and shrinkage coefficient $\xi$
1: $\mathbf{w}_1 \leftarrow 0$
2: **for** $t = 1, \ldots, T$ **do**
3:      Pick a unit vector $\mathbf{u}_t$ uniformly from the unit sphere $\partial\mathbb{B}$
4:      Observe $f_t(\mathbf{w}_t + \delta\mathbf{u}_t)$ and $f_t(\mathbf{w}_t - \delta\mathbf{u}_t)$
5:      Set $\mathbf{g}_t \leftarrow \frac{d}{2\delta}(f_t(\mathbf{w}_t + \delta\mathbf{u}_t) - f_t(\mathbf{w}_t - \delta\mathbf{u}_t))\mathbf{u}_t$
6:      Update $\mathbf{w}_{t+1} \leftarrow \Pi_{(1-\xi)\mathcal{X}}(\mathbf{w}_t - \eta_t\mathbf{g}_t)$
7: **end for**

---

### C.5. Omitted Details in Section 3.2

To start, we first present Algorithm 3 for strongly convex functions.

**Theorem 8.** *Under Assumptions 1-2, by setting $\eta_t = \frac{1}{\lambda t}$, $\delta = \frac{1}{2d^2TR}$, and $\xi = \frac{\delta}{r}$, Algorithm 3 satisfies:*

$$\mathbb{E}[\text{REG}_T] \leq \mathcal{O}\left(\frac{d}{\lambda}\log T\right).$$

Before the proof, we first state the following technical concentration inequality.

**Lemma 10** (Theorem 5.1.4 of Vershynin (2018)). *Let $X$ be uniformly distributed on the unit sphere $\partial\mathbb{B}$, and let $g : \mathbb{R}^d \to \mathbb{R}$ be G-Lipschitz with respect to the Euclidean norm. Then for all $t \geq 0$,*

$$\mathbb{P}(|g(X) - \mathbb{E}[g(X)]| > t) \leq 2 \cdot \exp(-c'dt^2/G^2).$$

**Proof** [of Theorem 8] For non-smooth functions, by simply choosing $\widetilde{\mathbf{g}}_t = \mathbf{0}$ in Eq. (C.8), we have

$$\mathbb{E}[\text{REG}_T] \leq 2\mathbb{E}\left[\sum_{t=1}^{T}\eta_t\|\mathbf{g}_t\|^2\right] + \mathcal{O}(1).$$

In the following, we first upper bound $\mathbb{E}[\|\mathbf{g}_t\|^2]$. Following the analysis in Shamir (2017), we have

$$\mathbb{E}\left[\|\mathbf{g}_t\|^2\right] = \frac{d^2}{4\delta^2}\mathbb{E}\left[\|(f_t(\mathbf{w}_t + \delta\mathbf{u}_t) - f_t(\mathbf{w}_t - \delta\mathbf{u}_t))\mathbf{u}_t\|^2\right] \leq \frac{d^2}{\delta^2}\mathbb{E}\left[\|(f_t(\mathbf{w}_t + \delta\mathbf{u}_t) - \mathbb{E}\left[f_t(\mathbf{w}_t + \delta\mathbf{u}_t)\right])\mathbf{u}_t\|^2\right]$$

$$\leq \frac{d^2}{\delta^2}\underbrace{\sqrt{\mathbb{E}\left[(f_t(\mathbf{w}_t + \delta\mathbf{u}_t) - \mathbb{E}\left[f_t(\mathbf{w}_t + \delta\mathbf{u}_t)\right])^4\right]}}_{\text{TERM (A)}}\underbrace{\sqrt{\mathbb{E}\left[\|\mathbf{u}_t\|^4\right]}}_{\text{TERM (B)}}.$$

For TERM (A), by the Lipschitzness of $f_t$, setting $g_t(\mathbf{u}) = f_t(\mathbf{w}_t + \delta\mathbf{u})$, by Lemma 10, we have

$$\sqrt{\mathbb{E}\left[(g_t(\mathbf{u}_t) - \mathbb{E}[g_t(\mathbf{u}_t)])^4\right]} = \sqrt{\int_{t=0}^{\infty}\mathbb{P}\left((g_t(\mathbf{u}_t) - \mathbb{E}[g_t(\mathbf{u}_t)])^4 > t\right)dt}$$

$$= \sqrt{\int_{t=0}^{\infty}\mathbb{P}(|g_t(\mathbf{u}_t) - \mathbb{E}[g_t(\mathbf{u}_t)]| > \sqrt[4]{t})dt} \leq \sqrt{\int_{t=0}^{\infty}2\exp\left(-\frac{c'd\sqrt{t}}{\delta^2 G^2}\right)dt} = \sqrt{2\frac{\delta^4 G^4}{(c'd)^2}}.$$

For TERM (B), since $\|\mathbf{u}_t\|^4 = 1$, we have $\mathbb{E}\left[\|\mathbf{g}_t\|^2\right] \leq CdG^2$, where $C$ is a constant independent of $d, T$.

With this upper bound, using Lemma 23 with $a_t = \mathbb{E}[\|\mathbf{g}_t\|^2]$, we have

$$\mathbb{E}[\text{REG}_T] \leq \mathcal{O}\left(\frac{d}{\lambda}\log\left(\mathbb{E}\left[\sum_{t=1}^{T}\|\mathbf{g}_t\|^2\right]\right)\right) \leq \mathcal{O}\left(\frac{d}{\lambda}\log T\right).$$

$\blacksquare$

## C.6. Proof of Lemma 4

**Proof** To begin with, using property *(viii)* in Lemma 6, it holds that

$$\mathbb{E}\left[\bar{V}_T\right] = \mathbb{E}\left[\sum_{t=1}^{T}\|\mathbf{g}_t - \widetilde{\mathbf{g}}_t\|^2\right] \leq d^2\mathbb{E}\left[\sum_{t=1}^{T}(\nabla_{i_t}f_t(\mathbf{w}_t) - \nabla_{i_t}f_{\alpha_t}(\mathbf{w}_{\alpha_t}))^2\right] + \mathcal{O}(1)$$

$$\leq 2d^2\underbrace{\mathbb{E}\left[\sum_{t=1}^{T}\langle\nabla f_t(\mathbf{w}_t), \mathbf{e}_{i_t}\rangle^2\right]}_{\text{TERM (A)}} + 2d^2\underbrace{\mathbb{E}\left[\sum_{t=1}^{T}\langle\nabla f_{\alpha_t}(\mathbf{w}_{\alpha_t}), \mathbf{e}_{i_t}\rangle^2\right]}_{\text{TERM (B)}} + \mathcal{O}(1). \tag{C.10}$$

For TERM (A), we have

$$\text{TERM (A)} = \frac{1}{d}\mathbb{E}\left[\sum_{t=1}^{T}\|\nabla f_t(\mathbf{w}_t)\|^2\right] \leq \frac{4L}{d}\mathbb{E}\left[F_T^{\mathbf{w}}\right], \tag{C.11}$$

where $F_T^{\mathbf{w}} \triangleq \sum_{t=1}^{T}f_t(\mathbf{w}_t) - \sum_{t=1}^{T}\min_{\mathbf{x}\in\mathcal{X}^+}f_t(\mathbf{x})$ and the inequality is by the self-bounding property $\|\nabla f(\mathbf{x})\|_2^2 \leq 4L(f(\mathbf{x}) - \min_{\mathbf{x}\in\mathcal{X}^+}f(\mathbf{x}))$ for any $L$-smooth function $f : \mathcal{X}^+ \to \mathbb{R}$ and any $\mathbf{x} \in \mathcal{X}$ (Yan et al., 2024).

For TERM (B), choosing $\mathbf{c} = \mathbf{0}$ in Lemma 7, we have

$$\text{TERM (B)} \leq \sum_{t=1}^{T}\sum_{s=1}^{t-1}\frac{1}{d^2}\left(1 - \frac{1}{d}\right)^{t-s-1}\mathbb{E}_{I_{s-1}}\left[\|\nabla f_s(\mathbf{w}_s)\|^2\right]$$

$$= \sum_{s=1}^{T-1}\mathbb{E}_{I_{s-1}}\left[\|\nabla f_s(\mathbf{w}_s)\|^2\right]\sum_{t=s+1}^{T}\frac{1}{d^2}\left(1 - \frac{1}{d}\right)^{t-s-1} \leq \frac{1}{d}\sum_{s=1}^{T-1}\mathbb{E}_{I_{s-1}}\left[\|\nabla f_s(\mathbf{w}_s)\|^2\right] \leq \frac{4L}{d}\mathbb{E}\left[F_T^{\mathbf{w}}\right], \tag{C.12}$$

where the last step is also by the self-bounding property.

Plugging Eq. (C.11) and Eq. (C.12) back to (C.10), we have

$$\mathbb{E}\left[\bar{V}_T\right] \leq 16dL\mathbb{E}\left[F_T^{\mathbf{w}}\right] + \mathcal{O}(1). \tag{C.13}$$

$\blacksquare$

## C.7. Proof of Theorem 3

**Proof** We start from the regret decomposition in Eq. (C.6). For TERM (A), we have the following bound:

$$\text{TERM (A)} \leq R\mathbb{E}\left[\sum_{t=1}^{T}\frac{\|\mathbf{g}_t - \widetilde{\mathbf{g}}_t\|^2}{\sqrt{d^2 + \bar{V}_{t-1}}}\right] \leq 4R\mathbb{E}\left[\sqrt{d^2 + \sum_{t=1}^{T}\|\mathbf{g}_t - \widetilde{\mathbf{g}}_t\|^2} + \max_{t\in[T]}\frac{\|\mathbf{g}_t - \widetilde{\mathbf{g}}_t\|^2}{d}\right]$$

$$\leq 4R\mathbb{E}\left[\sqrt{d^2 + \bar{V}_T}\right] + \mathcal{O}(d),$$

where the second inequality is by choosing $a_s = \frac{\|\mathbf{g}_s - \widetilde{\mathbf{g}}_s\|^2}{d^2}$ in Lemma 22.

For TERM (B), we have

$$\text{TERM (B)} \leq \mathbb{E}\left[\frac{2R^2}{\eta_1} + \frac{2R^2}{\eta_T}\right] = 2R\mathbb{E}\left[\sqrt{d^2 + \bar{V}_T}\right] + \mathcal{O}(d).$$

Thus, we have

$$\mathbb{E}[\text{REG}_T] \leq 6R\sqrt{d^2 + \mathbb{E}\left[\bar{V}_T\right]} + \mathcal{O}(d). \tag{C.14}$$

**Linear Functions.** In the linear setting, the gradient of $f_t(\cdot)$ is a constant vector and can be denoted by $\boldsymbol{\ell}_t$ for simplicity. For the gradient-variance bound, we have the following decomposition:

$$\mathbb{E}\left[\bar{V}_T\right] = \mathbb{E}\left[\sum_{t=1}^{T}\|\mathbf{g}_t - \widetilde{\mathbf{g}}_t\|^2\right] \leq d^2\mathbb{E}\left[\sum_{t=1}^{T}(\ell_{t,i_t} - \ell_{\alpha_t,i_t})^2\right] + \mathcal{O}(1)$$

$$\leq 2d^2\underbrace{\mathbb{E}\left[\sum_{t=1}^{T}(\ell_{t,i_t} - \mu_{T,i_t})^2\right]}_{\text{TERM (C)}} + 2d^2\underbrace{\mathbb{E}\left[\sum_{t=1}^{T}(\ell_{\alpha_t,i_t} - \mu_{T,i_t})^2\right]}_{\text{TERM (D)}} + \mathcal{O}(1).$$

For TERM (C), we have

$$\sum_{t=1}^{T}\mathbb{E}\left[(\ell_{t,i_t} - \mu_{T,i_t})^2\right] = \frac{1}{d}\sum_{t=1}^{T}\|\boldsymbol{\ell}_t - \boldsymbol{\mu}_T\|^2 = \frac{W_T}{d}. \tag{C.15}$$

For TERM (D), since for linear functions, $\boldsymbol{\mu}_T$ is a constant vector. Choosing $\mathbf{c} = \boldsymbol{\mu}_T$ in Lemma 7, we have

$$\mathbb{E}\left[\sum_{t=1}^{T}(\ell_{\alpha_t,i_t} - \mu_{T,i_t})^2\right] \leq \sum_{t=1}^{T}\sum_{s=1}^{t-1}\frac{1}{d^2}\left(1-\frac{1}{d}\right)^{t-s-1}\mathbb{E}_{I_{s-1}}\left[\|\boldsymbol{\ell}_s - \boldsymbol{\mu}_T\|^2\right] + \frac{1}{d}\sum_{t=1}^{T}\left(1-\frac{1}{d}\right)^{t-1}\|\boldsymbol{\mu}_T\|^2$$

$$\leq \sum_{s=1}^{T-1}\mathbb{E}_{I_{s-1}}\left[\|\boldsymbol{\ell}_s - \boldsymbol{\mu}_T\|^2\right]\sum_{t=s+1}^{T}\frac{1}{d^2}\left(1-\frac{1}{d}\right)^{t-s-1} + \|\boldsymbol{\mu}_T\|^2 \leq \frac{1}{d}\left(\sum_{s=1}^{T-1}\|\boldsymbol{\ell}_s - \boldsymbol{\mu}_T\|^2\right) + G^2 \leq \frac{W_T}{d} + G^2, \tag{C.16}$$

where the second inequality is because for linear functions, $\boldsymbol{\ell}_t$ is deterministic for $t \in [T]$ and $\|\boldsymbol{\ell}_0 - \boldsymbol{\mu}_T\| \leq G$.

Plugging Eq. (C.15) and Eq. (C.16) back, we have

$$\mathbb{E}\left[\bar{V}_T\right] \leq 4dW_T + \mathcal{O}(d^2). \tag{C.17}$$

Plugging (C.17) back to (C.14), we have

$$\mathbb{E}[\text{REG}_T] \leq 6R\sqrt{d^2 + 4dW_T + \mathcal{O}(d^2)} + \mathcal{O}(d) \leq \mathcal{O}\left(\sqrt{dW_T} + d\right).$$

**Convex Functions.** By property *(viii)* of Lemma 6, we have the following decomposition:

$$\bar{V}_T = \sum_{t=1}^{T}\|\mathbf{g}_t - \widetilde{\mathbf{g}}_t\|^2 \leq d^2\sum_{t=1}^{T}(\nabla f_{t,i_t}(\mathbf{w}_t) - \nabla f_{\alpha_t,i_t}(\mathbf{w}_{\alpha_t}))^2 + \mathcal{O}(1)$$

$$\leq d^2\sum_{t=1}^{T}\|\nabla f_t(\mathbf{w}_t) - \nabla f_{\alpha_t}(\mathbf{w}_{\alpha_t})\|^2 + \mathcal{O}(1)$$

$$\leq 2d^2\sum_{t=1}^{T}\left(\|\nabla f_t(\mathbf{w}_t) - \boldsymbol{\mu}_T\|^2 + \|\nabla f_{\alpha_t}(\mathbf{w}_{\alpha_t}) - \boldsymbol{\mu}_T\|^2\right) + \mathcal{O}(1)$$

$$\leq 4d^2\sum_{t=1}^{T}\|\nabla f_t(\mathbf{w}_t) - \boldsymbol{\mu}_T\|^2 + d^3\|\boldsymbol{\mu}_T\|^2 + \mathcal{O}(1)$$

$$\leq 4d^2\sup_{\{\mathbf{w}_1,\ldots,\mathbf{w}_T\}\in(1-\xi)\mathcal{X}}\left\{\sum_{t=1}^{T}\|\nabla f_t(\mathbf{w}_t) - \boldsymbol{\mu}_T\|^2\right\} + \mathcal{O}(d^3) \leq 4d^2W_T + \mathcal{O}(d^3), \tag{C.18}$$

where $\boldsymbol{\mu}_T \triangleq \frac{1}{T}\sum_{t=1}^{T}\nabla f_t(\mathbf{w}_t)$. The third step is by $\langle\mathbf{x},\mathbf{e}_i\rangle^2 \leq \|\mathbf{x}\|^2$ for any $\mathbf{x}$. The fifth step follows by observing that for each $s \geq 1$, there exists at most a single index $t$ such that $\alpha_t = s$, while $\alpha_t = 0$ occurs no more than $d$ times.

Plugging (C.18) back to (C.14), we have

$$\mathbb{E}[\text{REG}_T] \leq 6R\sqrt{d^2 + 4d^2W_T + \mathcal{O}(d^3)} + \mathcal{O}(d) \leq \mathcal{O}\left(d\sqrt{W_T} + d\right).$$

**Strongly Convex Functions.** By plugging Eq. (C.18) into the bound in Eq. (C.9), we directly have

$$\mathbb{E}[\text{REG}_T] \leq \mathcal{O}\left(\frac{d}{\lambda} \log\left(dW_T\right)\right),$$

which completes the proof. ∎

### C.8. Proof of Theorem 4

**Proof**

**Convex Functions.** We start from the regret decomposition in Eq. (C.14). Plugging (C.13) back to (C.14), we have

$$\mathbb{E}[\text{REG}_T] \leq 6R\sqrt{d^2 + 16dL\mathbb{E}\left[F_T^{\mathbf{w}}\right] + \mathcal{O}(1)} + \mathcal{O}(d)$$

$$\leq \mathcal{O}\left(\sqrt{d\mathbb{E}\left[F_T^{\mathbf{w}}\right]} + d\right) \leq \mathcal{O}\left(\sqrt{dF_T} + d\right),$$

where the last step uses Lemma 24 by choosing $a = \mathcal{O}(d)$ and $b$ as a constant independent of $d, T$ and setting

$$x = \sum_{t=1}^{T} f_t(\mathbf{w}_t) - \sum_{t=1}^{T} \min_{\mathbf{x} \in \mathcal{X}^+} f_t(\mathbf{x}) \text{ and } y = \min_{\mathbf{x} \in \mathcal{X}} \sum_{t=1}^{T} f_t(\mathbf{x}) - \sum_{t=1}^{T} \min_{\mathbf{x} \in \mathcal{X}^+} f_t(\mathbf{x}).$$

**Strongly Convex Functions.** Plugging (C.13) into the bound in Eq. (C.9), we directly have

$$\mathbb{E}[\text{REG}_T] \leq \mathcal{O}\left(\frac{d}{\lambda} \log\left(d\mathbb{E}\left[F_T^{\mathbf{w}}\right]\right)\right) \leq \mathcal{O}\left(\frac{d}{\lambda} \log\left(dF_T\right)\right),$$

where the last step uses Lemma 25 by choosing $a = \mathcal{O}(d)$ and $b, c$ as constants independent of $d, T$ and setting

$$x = \sum_{t=1}^{T} f_t(\mathbf{w}_t) - \sum_{t=1}^{T} \min_{\mathbf{x} \in \mathcal{X}^+} f_t(\mathbf{x}) \text{ and } y = \min_{\mathbf{x} \in \mathcal{X}} \sum_{t=1}^{T} f_t(\mathbf{x}) - \sum_{t=1}^{T} \min_{\mathbf{x} \in \mathcal{X}^+} f_t(\mathbf{x}).$$

∎

# D. Omitted Details in Section 4

In this section, we provide the omitted details in Section 4. Specifically, we first present a practical version of Algorithm 2 in Appendix D.1, and then prove its gradient-variation regret in Appendix D.2. Without loss of generality, we assume $a_i \leq 0 < b_i$ for all $i \in [d]$.

### D.1. A Practical Version of Algorithm 2

To start, we show the relationship between $w_{t+1,i}$ and $\widetilde{g}_{t+1,i}$.

$$\eta \sum_{s=1}^{t} g_{s,i} + \eta \widetilde{g}_{t+1,i} + \frac{1}{b_i - w_{t+1,i}} - \frac{1}{w_{t+1,i} - a_i} = 0, \tag{D.1}$$

$$\widetilde{g}_{t+1,i} = \frac{1}{2} \lambda_{t+1,i}^{\frac{1}{2}} (r_i^{(+1)} - r_i^{(-1)}), \tag{D.2}$$

$$\lambda_{t+1,i} = \frac{1}{(b_i - w_{t+1,i})^2} + \frac{1}{(w_{t+1,i} - a_i)^2}. \tag{D.3}$$

Let $f_i(x) = -\log(b_i - x) - \log(x - a_i)$ be the univariate log-barrier. By combining Eq. (D.1)-Eq. (D.3), we can show that for each $i \in [d]$, $w_{t+1,i}$ satisfies the optimality condition $F_{t,i}(x) = 0$, where $F_{t,i}$ is defined as:

$$F_{t,i}(x) \triangleq \eta \left( \sum_{s=1}^t g_{s,i} + \frac{1}{2} \sqrt{f_i''(x)}(r_i^{(+1)} - r_i^{(-1)}) \right) + f_i'(x). \tag{D.4}$$

The existence and uniqueness of the solution $w_{t+1,i} \in (a_i, b_i)$ are established in Lemma 11.

**Lemma 11.** *Assume that for each dimension $i \in [d]$, the step size $\eta$ and the buffer constants $c_i = \frac{1}{2}(r_i^{(+1)} - r_i^{(-1)})$ satisfy the condition $|\eta c_i| < 1$. Then $F_{t,i}(x)$ is strictly monotone and the equation $F_{t,i}(x) = 0$ has a unique solution $w_{t+1,i}$ in the open interval $(a_i, b_i)$.*

**Proof** For simplicity, let $L(x) = x - a_i$ and $R(x) = b_i - x$. We can express the derivatives of $f_i(x)$ as:

$$f_i'(x) = \frac{1}{R(x)} - \frac{1}{L(x)}, \qquad \text{and} \qquad f_i''(x) = \frac{1}{L(x)^2} + \frac{1}{R(x)^2}.$$

Let $S_{t,i} = \sum_{s=1}^t g_{s,i}$ and $c_i = \frac{1}{2}(r_i^{(+1)} - r_i^{(-1)})$. We rewrite (D.4) as:

$$F_{t,i}(x) = \eta S_{t,i} + \eta c_i \sqrt{f_i''(x)} + f_i'(x) = \eta S_{t,i} + \frac{\eta c_i \sqrt{L(x)^2 + R(x)^2} + L(x) - R(x)}{L(x)R(x)}.$$

**Existence:** We evaluate the limits of $F_{t,i}(x)$ as $x$ approaches the boundaries. For the limit $x \to a_i^+$, we have $L(x) \to 0^+$ and $R(x) \to b_i - a_i$. The numerator of the fraction satisfies:

$$\lim_{x \to a_i^+} \left( \eta c_i \sqrt{L(x)^2 + R(x)^2} + L(x) - R(x) \right) = \eta c_i(b_i - a_i) + 0 - (b_i - a_i) = (\eta c_i - 1)(b_i - a_i).$$

Since $|\eta c_i| < 1$, it follows that $(\eta c_i - 1) < 0$. As the denominator $L(x)R(x) \to 0^+$, we have:

$$\lim_{x \to a_i^+} F_{t,i}(x) = \eta S_{t,i} + \frac{(\eta c_i - 1)(b_i - a_i)}{0^+} = -\infty.$$

Similarly, for the limit $x \to b_i^-$, we have $R(x) \to 0^+$ and $L(x) \to b_i - a_i$. The numerator satisfies:

$$\lim_{x \to b_i^-} \left( \eta c_i \sqrt{L(x)^2 + R(x)^2} + L(x) - R(x) \right) = \eta c_i(b_i - a_i) + (b_i - a_i) - 0 = (\eta c_i + 1)(b_i - a_i).$$

Since $\eta c_i + 1 > 0$, and the denominator $L(x)R(x) \to 0^+$, we have:

$$\lim_{x \to b_i^-} F_{t,i}(x) = \eta S_{t,i} + \frac{(\eta c_i + 1)(b_i - a_i)}{0^+} = +\infty.$$

By the Intermediate Value Theorem, there exists at least one $x^* \in (a_i, b_i)$ such that $F_{t,i}(x^*) = 0$.

**Uniqueness:** We show that $F_{t,i}(x)$ is strictly monotonically increasing by examining its derivative:

$$F_{t,i}'(x) = f_i''(x) + \eta c_i \frac{f_i'''(x)}{2\sqrt{f_i''(x)}} = f_i''(x) \left( 1 + \eta c_i \frac{f_i'''(x)}{2(f_i''(x))^{3/2}} \right).$$

Substituting $f_i'''(x) = 2(R(x)^{-3} - L(x)^{-3})$, and letting $\rho = R(x)/L(x) \in (0, \infty)$, the ratio term becomes:

$$\left| \frac{f_i'''(x)}{2(f_i''(x))^{3/2}} \right| = \left| \frac{R^{-3} - L^{-3}}{(L^{-2} + R^{-2})^{3/2}} \right| = \left| \frac{1 - \rho^3}{(1 + \rho^2)^{3/2}} \right|.$$

Let $h(\rho) = \frac{1 - \rho^3}{(1+\rho^2)^{3/2}}$. It can be verified that $|h(\rho)| < 1$ for all $\rho > 0$. Therefore,

$$F_{t,i}'(x) > f_i''(x)(1 - |\eta c_i|).$$

Since $|\eta c_i| < 1$ and $f_i''(x) > 0$, we have $F_{t,i}'(x) > 0$ for $x \in (a_i, b_i)$. The strict monotonicity implies the unique solution. $\blacksquare$

With the theoretical guarantee, we present a practical version for Algorithm 2 in Algorithm 4.

---

**Algorithm 4** Practical Version of Algorithm 2

---

**Input:** Step size $\eta > 0$

   Let $\mathbf{w}_1 = \mathbf{0}$, $\widetilde{\mathbf{g}}_1 = \mathbf{0}$. Let buffer vectors $\mathbf{r}^{(+1)} = \mathbf{r}^{(-1)} = \mathbf{0}$.

   **for** round $t \in [T]$ **do**

      Choose $i_t$ uniformly from $[d]$ and $\varepsilon_t$ uniformly from $\{-1, +1\}$ and fetch $z_t = r_{i_t}^{(\varepsilon_t)}$

      Play action $\mathbf{x}_t = \mathbf{w}_t + \varepsilon_t \lambda_{t,i_t}^{-1/2} \mathbf{e}_{i_t}$

      Observe partial information $v_t = \langle \boldsymbol{\ell}_t, \mathbf{x}_t \rangle$

      Compute $\mathbf{g}_t$ as in Eq. (4.2)

      Update buffer vector $r_{i_t}^{(\varepsilon_t)} = v_t$

      Construct $F_{t,i}$ as in Eq. (D.4) and solve the following system of equations for $\mathbf{w}_{t+1}$

$$F_{t,i}(w_{t+1,i}) = 0, \ \forall i \in [d]$$

      Compute $\widetilde{\mathbf{g}}_{t+1}$ as in Eq. (4.2)

   **end for**

---

### D.2. Proof of Theorem 5

Before the proof, we introduce Lemma 12 that will be used in the proof.

**Lemma 12.** *Let* $\lambda_i(x) = \frac{1}{(b_i-x)^2} + \frac{1}{(x-a_i)^2}$. *Then, for any* $x, y \in (a_i, b_i)$, *we have* $\left| \lambda_i^{-\frac{1}{2}}(x) - \lambda_i^{-\frac{1}{2}}(y) \right| \leq |x - y|$.

**Proof** [of Theorem 5] First, we denote $\|\mathbf{a}\|_{\mathbf{w}_t} = \sqrt{\mathbf{a}^\top \nabla^2 \mathcal{R}(\mathbf{w}_t)\mathbf{a}}$ and $\|\mathbf{a}\|_{\mathbf{w}_t^{-1}} = \sqrt{\mathbf{a}^\top \nabla^{-2} \mathcal{R}(\mathbf{w}_t)\mathbf{a}}$ for any $\mathbf{a} \in \mathbb{R}^d$.

To start, we observe that $\mathbb{E}_{i_t, \varepsilon_t}[\mathbf{x}_t] = \mathbf{w}_t$ and

$$\mathbb{E}_{i_t, \varepsilon_t}[\mathbf{g}_t] = \mathbb{E}_{i_t, \varepsilon_t}\left[ d\left( \langle \boldsymbol{\ell}_t, \mathbf{x}_t \rangle - \langle \boldsymbol{\ell}_{\alpha_t}, \mathbf{x}_{\alpha_t} \rangle \right) \varepsilon_t \lambda_{t,i_t}^{\frac{1}{2}} \mathbf{e}_{i_t} \right] + \widetilde{\mathbf{g}}_t \quad \text{(by definition of } \mathbf{g}_t)$$

$$= d\mathbb{E}_{i_t, \varepsilon_t}\left[ \langle \boldsymbol{\ell}_t, \mathbf{w}_t + \varepsilon_t \lambda_{t,i_t}^{-\frac{1}{2}} \mathbf{e}_{i_t} \rangle \varepsilon_t \lambda_{t,i_t}^{\frac{1}{2}} \mathbf{e}_{i_t} \right] - d\mathbb{E}_{i_t, \varepsilon_t}\left[ \varepsilon_t \lambda_{t,i_t}^{\frac{1}{2}} \langle \boldsymbol{\ell}_{\alpha_t}, \mathbf{x}_{\alpha_t} \rangle \mathbf{e}_{i_t} \right] + \widetilde{\mathbf{g}}_t$$

$$= d\mathbb{E}_{i_t, \varepsilon_t}\left[ \langle \boldsymbol{\ell}_t, \mathbf{w}_t + \varepsilon_t \lambda_{t,i_t}^{-\frac{1}{2}} \mathbf{e}_{i_t} \rangle \varepsilon_t \lambda_{t,i_t}^{\frac{1}{2}} \mathbf{e}_{i_t} \right] - \frac{d}{2} \mathbb{E}_{i_t, \varepsilon_t}\left[ \varepsilon_t \lambda_{t,i_t}^{\frac{1}{2}} \left( \langle \boldsymbol{\ell}_{\alpha_t}, \mathbf{x}_{\alpha_t} \rangle - \langle \boldsymbol{\ell}_{\beta_t}, \mathbf{x}_{\beta_t} \rangle \right) \mathbf{e}_{i_t} \right] + \widetilde{\mathbf{g}}_t$$

$$= d\mathbb{E}_{i_t, \varepsilon_t}\left[ \langle \boldsymbol{\ell}_t, \mathbf{w}_t + \varepsilon_t \lambda_{t,i_t}^{-\frac{1}{2}} \mathbf{e}_{i_t} \rangle \varepsilon_t \lambda_{t,i_t}^{\frac{1}{2}} \mathbf{e}_{i_t} \right] - d\mathbb{E}_{i_t}\left[ \widetilde{g}_{t,i_t} \mathbf{e}_{i_t} \right] + \widetilde{\mathbf{g}}_t \quad \text{(by definition of } \widetilde{\mathbf{g}}_t)$$

$$= d\mathbb{E}_{i_t, \varepsilon_t}\left[ \langle \boldsymbol{\ell}_t, \mathbf{w}_t + \varepsilon_t \lambda_{t,i_t}^{-\frac{1}{2}} \mathbf{e}_{i_t} \rangle \varepsilon_t \lambda_{t,i_t}^{\frac{1}{2}} \mathbf{e}_{i_t} \right] = d\mathbb{E}_{i_t}\left[ \ell_{t,i_t} \mathbf{e}_{i_t} \right] = \boldsymbol{\ell}_t,$$

where we define $\beta_t$ as the largest integer such that $0 \leq \beta_t < t$, $i_{\beta_t} = i_t$, and $\varepsilon_{\beta_t} = -\varepsilon_t$. The third and sixth steps are by the symmetry of $\varepsilon_t$, and both the fifth and the last step follow from $d\mathbb{E}[v_i \mathbf{e}_i] = \sum_{i=1}^d v_i \mathbf{e}_i = \mathbf{v}$, for any $\mathbf{v} \in \mathbb{R}^d$. Then, we have

$$\mathbb{E}[\langle \boldsymbol{\ell}_t, \mathbf{x}_t \rangle] = \mathbb{E}[\langle \boldsymbol{\ell}_t, \mathbb{E}_{i_t, \varepsilon_t}[\mathbf{x}_t] \rangle] = \mathbb{E}[\langle \mathbb{E}_{i_t, \varepsilon_t}[\mathbf{g}_t], \mathbf{w}_t \rangle] = \mathbb{E}[\langle \mathbf{g}_t, \mathbf{w}_t \rangle].$$

Thus, by the FTRL analysis framework, e.g., Theorem 7.47 of Orabona (2019), we have

$$\mathbb{E}[\text{REG}_T] = \mathbb{E}\left[ \sum_{t=1}^T \langle \boldsymbol{\ell}_t, \mathbf{x}_t - \mathbf{v} \rangle \right] + \mathbb{E}\left[ \sum_{t=1}^T \langle \boldsymbol{\ell}_t, \mathbf{v} - \mathbf{x}^\star \rangle \right] \leq \mathbb{E}\left[ \sum_{t=1}^T \langle \mathbf{g}_t, \mathbf{w}_t - \mathbf{v} \rangle \right] + \mathcal{O}(1)$$

$$\leq \underbrace{\mathbb{E}\left[ \eta \sum_{t=1}^T \|\mathbf{g}_t - \widetilde{\mathbf{g}}_t\|_{\mathbf{w}_t^{-1}}^2 \right]}_{\text{TERM (A)}} + \underbrace{\mathbb{E}\left[ \frac{\mathcal{R}(\mathbf{v})}{\eta} \right]}_{\text{TERM (B)}} - \underbrace{\mathbb{E}\left[ \frac{1}{\eta} \sum_{t=1}^T \|\mathbf{w}_t - \mathbf{w}_{t+1}\|_{\mathbf{w}_t}^2 \right]}_{\text{TERM (C)}} + \mathcal{O}(1), \tag{D.5}$$

where $\mathbf{x}^\star \in \arg\min_{\mathbf{x} \in \mathcal{X}} \sum_{t=1}^T \langle \boldsymbol{\ell}_t, \mathbf{x} \rangle$ and $\mathbf{v} \triangleq \frac{T-1}{T} \mathbf{x}^\star + \frac{1}{T} \mathbf{x}^\dagger$ where $\mathbf{x}^\dagger \in \mathcal{X}$ satisfies $\nabla \mathcal{R}(\mathbf{x}^\dagger) = \mathbf{0}$.

For TERM (A), we have the following bound:

$$\text{TERM (A)} = \eta d^2 \mathbb{E}\left[ \sum_{t=1}^T (v_t - z_t)^2 \lambda_{t,i_t} \mathbf{e}_{i_t}^\top \nabla^{-2} \mathcal{R}(\mathbf{w}_t) \mathbf{e}_{i_t} \right] \leq \eta d^2 \mathbb{E}\left[ \sum_{t=1}^T (v_t - z_t)^2 \right] \tag{D.6}$$

For $(v_t - z_t)^2$, we have the following decomposition:

$$
\begin{aligned}
(v_t - z_t)^2 &= (\langle \boldsymbol{\ell}_t, \mathbf{x}_t \rangle - \langle \boldsymbol{\ell}_{\alpha_t}, \mathbf{x}_{\alpha_t} \rangle)^2 \leq 2\langle \boldsymbol{\ell}_t, \mathbf{x}_t - \mathbf{x}_{\alpha_t} \rangle^2 + 2\langle \boldsymbol{\ell}_t - \boldsymbol{\ell}_{\alpha_t}, \mathbf{x}_{\alpha_t} \rangle^2 \\
&\leq 2G^2 \left\| \mathbf{w}_t + \varepsilon_t \lambda_{t,i_t}^{-\frac{1}{2}} \mathbf{e}_{i_t} - \mathbf{w}_{\alpha_t} - \varepsilon_t \lambda_{\alpha_t,i_t}^{-\frac{1}{2}} \mathbf{e}_{i_t} \right\|^2 + 2R^2 \left\| \boldsymbol{\ell}_t - \boldsymbol{\ell}_{\alpha_t} \right\|^2 && \text{(by } \|\boldsymbol{\ell}_t\| \leq G) \\
&\leq 4G^2 \|\mathbf{w}_t - \mathbf{w}_{\alpha_t}\|^2 + 4G^2 \left\| \lambda_{t,i_t}^{-\frac{1}{2}} - \lambda_{\alpha_t,i_t}^{-\frac{1}{2}} \right\|^2 + 2R^2 \|\boldsymbol{\ell}_t - \boldsymbol{\ell}_{\alpha_t}\|^2 \\
&\leq 8G^2 \|\mathbf{w}_t - \mathbf{w}_{\alpha_t}\|^2 + 2R^2 \|\boldsymbol{\ell}_t - \boldsymbol{\ell}_{\alpha_t}\|^2 \\
&\leq 2(t - \alpha_t)\left( 4G^2 \sum_{s=\alpha_t+1}^{t} \|\mathbf{w}_s - \mathbf{w}_{s-1}\|^2 + R^2 \sum_{s=\alpha_t+1}^{t} \|\boldsymbol{\ell}_s - \boldsymbol{\ell}_{s-1}\|^2 \right),
\end{aligned}
$$

where the third inequality is by Lemma 12 and the last step is by the triangle inequality and Cauchy-Schwarz inequality. Plugging the above inequality into Eq. (D.6), we have

$$
\begin{aligned}
\text{TERM (A)} &\leq \eta d^2 \mathbb{E}\left[ \sum_{t=1}^{T} \sum_{\varepsilon \in \{-1,+1\}} \sum_{i=1}^{d} \rho_{t,i}^{(\varepsilon)} \left( 8G^2 \|\mathbf{w}_t - \mathbf{w}_{t-1}\|^2 + 2R^2 \|\boldsymbol{\ell}_t - \boldsymbol{\ell}_{t-1}\|^2 \right) \right] \\
&\leq 16\eta d^4 R^2 \sum_{t=1}^{T} \|\boldsymbol{\ell}_t - \boldsymbol{\ell}_{t-1}\|^2 + 128\eta d^4 G^2 \ln(2dT) \mathbb{E}\left[ \sum_{t=2}^{T} \|\mathbf{w}_t - \mathbf{w}_{t-1}\|^2 \right] + \mathcal{O}(1), && \text{(D.7)}
\end{aligned}
$$

where in this case, we define the non-consecutive sampling gap $\rho_{t,i}^{\varepsilon}$ as

$$
\rho_{t,i}^{(\varepsilon)} \triangleq \tau_2 - \tau_1, \text{ where } \tau_1 \triangleq \max\{0 \leq \tau < t \mid i_\tau = i, \varepsilon_\tau = \varepsilon\}, \text{ and } \tau_2 \triangleq \min\{t \leq \tau \leq T+1 \mid i_\tau = i, \varepsilon_\tau = \varepsilon\}.
$$

The last inequality follows from setting $n = 2d$ in Lemma 8 and noting that $\rho_{t,i}^{(\varepsilon)} \leq 8d\ln(2dT)$ simultaneously for all $(i, \varepsilon)$ and $t$ with probability at least $1 - \mathcal{O}(1/(d^3 T^3))$. The proof for this high-probability bound is analogous to the one provided in Appendix C.3 and we omit it here.

For TERM (B), we have

$$
\text{TERM (B)} = \frac{\mathcal{R}(\mathbf{v})}{\eta} \leq \frac{2d\log T + \mathcal{O}(d)}{\eta}. \tag{D.8}
$$

For TERM (C), we have

$$
\begin{aligned}
\text{TERM (C)} &= \frac{1}{\eta} \mathbb{E}\left[ \sum_{t=2}^{T} \|\mathbf{w}_t - \mathbf{w}_{t-1}\|_{\mathbf{w}_{t-1}}^2 \right] = \frac{1}{\eta} \mathbb{E}\left[ \sum_{t=2}^{T} \sum_{i=1}^{d} \lambda_{t-1,i}(\mathbf{w}_{t,i} - \mathbf{w}_{t-1,i})^2 \right] \\
&\geq \frac{4}{\max_i (b_i - a_i)^2 \eta} \mathbb{E}\left[ \sum_{t=2}^{T} \|\mathbf{w}_t - \mathbf{w}_{t-1}\|^2 \right], \tag{D.9}
\end{aligned}
$$

where the last step is by $\lambda_{t-1,i} = \frac{1}{(w_{t-1,i} - a_i)^2} + \frac{1}{(b_i - w_{t-1,i})^2} \geq \frac{4}{(b_i - a_i)^2}$.

Plugging (D.7), (D.8), and (D.9) into (D.5), we have

$$
\begin{aligned}
\mathbb{E}[\text{REG}_T] &\leq 16\eta d^4 R^2 \sum_{t=1}^{T} \|\boldsymbol{\ell}_t - \boldsymbol{\ell}_{t-1}\|^2 + \frac{2d\log T + \mathcal{O}(d)}{\eta} + \mathcal{O}(1) \\
&\quad + \mathbb{E}\left[ \left( 128\eta d^4 G^2 \ln(2dT) - \frac{4}{\eta \max_i (b_i - a_i)^2} \right) \sum_{t=2}^{T} \|\mathbf{w}_t - \mathbf{w}_{t-1}\|^2 \right] \leq \mathcal{O}\left( d^{\frac{7}{2}} \sqrt{V_T \log^2 T \log(dT)} \right),
\end{aligned}
$$

where $V_T = \sum_{t=2}^{T} \|\boldsymbol{\ell}_t - \boldsymbol{\ell}_{t-1}\|^2$ and the last step is by choosing $\eta = \frac{1}{16RGd^2} \frac{1}{\sqrt{V_T \log(2dT)}}$, using $R = \mathcal{O}(\sqrt{d})$. ∎

Finally, for completeness, we prove that $\mathbf{x}_t \in \mathcal{X}$ by Lemma 13.

**Lemma 13.** *Under Assumptions 1-3, assume $\mathcal{X} = \prod_{i=1}^{d}[a_i, b_i]$, where $a_i \leq 0 \leq b_i$ for all $i \in [d]$. Choosing $\eta = \frac{1}{16RGd^2\sqrt{V_T \ln(2dT)}}$, for any $t$, $\mathbf{x}_t$ played by Algorithm 4 is in the interior of $\mathcal{X}$.*

**Proof** [of Lemma 13] Recall that for any $t$, $\mathbf{x}_t = \mathbf{w}_t + \varepsilon_t \lambda_{t,i_t}^{-1/2}\mathbf{e}_{i_t}$.

Since the update is coordinate-wise, we only need to verify $a_{i_t} < x_{t,i_t} < b_{i_t}$. The $i_t$-th eigenvalue of the log-barrier Hessian at $\mathbf{w}_t$ is $\lambda_{t,i_t} = (b_{i_t} - w_{t,i_t})^{-2} + (w_{t,i_t} - a_{i_t})^{-2}$. This definition implies:

$$\lambda_{t,i_t}^{-1/2} < \min(b_{i_t} - w_{t,i_t}, w_{t,i_t} - a_{i_t}).$$

Consequently, for either choice of $\varepsilon_t \in \{\pm 1\}$, if $\varepsilon_t = 1$, $x_{t,i_t} = w_{t,i_t} + \lambda_{t,i_t}^{-1/2} < w_{t,i_t} + (b_{i_t} - w_{t,i_t}) = b_{i_t}$. If $\varepsilon_t = -1$, $x_{t,i_t} = w_{t,i_t} - \lambda_{t,i_t}^{-1/2} > w_{t,i_t} - (w_{t,i_t} - a_{i_t}) = a_{i_t}$. Since $x_{t,i_t}$ is strictly within the boundaries, $\mathbf{x}_t$ remains in the interior of $\mathcal{X}$, which finishes the proof. ∎

Finally, for the sake of completeness, we provide the proof of Lemma 12.

**Proof** [of Lemma 12] Let $g(x) = \frac{1}{\sqrt{\lambda_i(x)}} = \left(\frac{1}{(x-a_i)^2} + \frac{1}{(b_i-x)^2}\right)^{-1/2}$. To prove that $g(x)$ is 1-Lipschitz, it suffices to show that $|g'(x)| \leq 1$ for all $x \in (a_i, b_i)$.

Let $u = x - a_i$ and $v = b_i - x$. Note that since $x \in (a_i, b_i)$, both $u > 0$ and $v > 0$. The function can be rewritten as:

$$g(x) = (u^{-2} + v^{-2})^{-1/2}$$

Taking the derivative with respect to $x$ (noting that $\frac{du}{dx} = 1$ and $\frac{dv}{dx} = -1$):

$$|g'(x)| = \left|\frac{u^{-3} - v^{-3}}{(u^{-2} + v^{-2})^{3/2}}\right| \leq 1,$$

where the last step follows from the fact that $u, v > 0$. ∎

# E. Omitted Proofs in Section 5

In this section, we provide the details for the results presented in Section 5, including dynamic regret and universal regret. Specifically, for both dynamic regret and universal regret, we leverage an online ensemble framework with a two-layer meta-base architecture to mitigate environmental uncertainty. In this setup, a diverse set of base learners is deployed to explore the environment, while a meta-algorithm adaptively tracks the best-performing individual.

## E.1. Omitted Details in Section 5.1

In this part, we provide the omitted details for our applications in dynamic regret. Building upon the online ensemble framework of Zhao et al. (2024), we introduce new surrogate loss functions to tackle the information limitations in the bandit setting. Specifically, we design a two-layer online ensemble structure, with the meta-algorithm employing Optimistic-Hedge (Hazan, 2016) to aggregate $N$ base learners. Each base learner is instantiated as Algorithm 1 with a distinct step size from a predefined pool and the $i$-th base learner outputs $\mathbf{w}_{t,i}$ at round $t$. In the following, we elaborate on the online scheduling strategy, base learners, and architecture of the meta-algorithm.

**Online Scheduling Strategy and Base Learners.** To ensure thorough exploration, we construct a candidate step-size pool $\mathcal{H}$, where each base learner is instantiated with a unique step size from this pool. Specifically, the set of candidate step sizes $\mathcal{H}$ is defined as

$$\mathcal{H} = \left\{\eta_i = \min\left\{\frac{1}{20L\sqrt{d^3\log(dT)}}, \sqrt{\frac{R^2}{d^3 T \log d}} \cdot 2^{i-1}\right\} \,\Big|\, i \in [N]\right\}, \tag{E.1}$$

where $N = \left\lceil \log_2\left(1 + \frac{\sqrt{T\log d}}{16L\sqrt{\log(dT)}}\right)\right\rceil + 1 = \mathcal{O}(\log T)$ is the number of base learners. For each $i \in [N]$, we instantiate the base learner $\mathcal{B}_i$ via Algorithm 1 with a fixed step size $\eta_i \in \mathcal{H}$. Specifically, Algorithm 1 can be interpreted as an

instance of Optimistic Online Gradient Descent (OOGD) (Rakhlin & Sridharan, 2013b), where the gradient $\nabla f_t(\mathbf{x}_t)$ and the optimistic hint $M_t$ are substituted with the estimators $\mathbf{g}_t$ and $\widetilde{\mathbf{g}}_t$, respectively. Recall that OOGD updates the decision as follows:

$$\mathbf{x}_t = \Pi_{\mathcal{X}}\left[\widehat{\mathbf{x}}_t - \eta_t M_t\right], \quad \widehat{\mathbf{x}}_{t+1} = \Pi_{\mathcal{X}}\left[\widehat{\mathbf{x}}_t - \eta_t \nabla f_t(\mathbf{x}_t)\right] \tag{E.2}$$

where $\eta_t > 0$ is a time-varying step size, $\widehat{\mathbf{x}}_t$ and $\widehat{\mathbf{x}}_{t+1}$ are internal decisions, and $\Pi_{\mathcal{X}}[\mathbf{x}] \triangleq \arg\min_{\mathbf{y} \in \mathcal{X}} \|\mathbf{x} - \mathbf{y}\|_2$ is the Euclidean projection onto the feasible domain $\mathcal{X}$. We then characterize the dynamic regret guarantee of OOGD below.

**Lemma 14** (Zhao et al. (2024))**.** *Under Assumptions 1-3, if the loss functions are convex, OOGD with $\eta \leq \frac{1}{4L}$, enjoys the following bound: for any $\{\mathbf{v}_t\}_{t=1}^T$*

$$\sum_{t=1}^T f_t(\mathbf{w}_t) - \sum_{t=1}^T f_t(\mathbf{v}_t) \leq \eta \sum_{t=1}^T \|\nabla f_t(\mathbf{w}_t) - M_t\|_2^2 + \frac{4}{\eta}\left(R^2 + RP_T\right) - \frac{1}{4\eta}\sum_{t=2}^T \|\mathbf{w}_t - \mathbf{w}_{t-1}\|_2^2,$$

*where $P_T \triangleq \sum_{t=2}^T \|\mathbf{v}_t - \mathbf{v}_{t-1}\|$.*

**Meta-algorithm: Optimistic-Hedge.** Formally, Optimistic-Hedge performs the following update: at round $t + 1$, given the expert losses $\boldsymbol{\ell}_t \in \mathbb{R}^N$ and an optimistic vector $\boldsymbol{m}_{t+1} \in \mathbb{R}^N$, the learner computes the weight vector $\boldsymbol{p}_{t+1} \in \Delta_N$:

$$p_{t+1,i} \propto \exp\left(-\varepsilon_t\left(\sum_{s=1}^t \ell_{s,i} + m_{t+1,i}\right)\right), \quad \forall i \in [N]. \tag{E.3}$$

Here, $\Delta_N$ represents an $N$-dimensional simplex, and $\varepsilon_t > 0$ is the learning rate of the meta-algorithm. The optimism $\boldsymbol{m}_{t+1} \in \mathbb{R}^N$ can be interpreted as an optimistic guess of the loss of round $t + 1$, and we thus incorporate it into the cumulative loss for update. Optimistic-Hedge enjoys the following regret guarantee:

**Lemma 15.** *The regret of Optimistic-Hedge (E.3) with learning rate*

$$\varepsilon_t = \sqrt{\frac{\ln N}{C_0^2 + \sum_{s=1}^t \|\boldsymbol{\ell}_s - \boldsymbol{m}_s\|_\infty^2}},$$

*to any expert $i \in [N]$ is at most*

$$\sum_{t=1}^T \langle \boldsymbol{\ell}_t, \boldsymbol{p}_t - \mathbf{e}_i \rangle \leq 5\sqrt{\ln N\left(C_0^2 + \sum_{t=1}^T \|\boldsymbol{\ell}_t - \boldsymbol{m}_t\|_\infty^2\right)} - \sum_{t=1}^T \frac{C_0}{4\sqrt{\ln N}}\|\boldsymbol{p}_t - \boldsymbol{p}_{t+1}\|_1^2 + \frac{\sqrt{\ln N}}{C_0}\max_{t \in [T]}\|\boldsymbol{\ell}_t - \boldsymbol{m}_t\|_\infty^2,$$

*where $C_0$ is a positive constant.*

To tackle the information limitations, we set the aforementioned loss $\boldsymbol{\ell}_t$ and the optimism $\boldsymbol{m}_t$ as follows: for $i \in [N]$

$$\ell_{t,i} \triangleq \langle \mathbf{g}_t, \mathbf{w}_{t,i} \rangle + \gamma\|\mathbf{w}_{t,i} - \mathbf{w}_{t-1,i}\|^2, \quad m_{t,i} \triangleq \langle \widetilde{\mathbf{g}}_t, \mathbf{w}_{t,i} \rangle + \gamma\|\mathbf{w}_{t,i} - \mathbf{w}_{t-1,i}\|^2, \tag{E.4}$$

where $\gamma > 0$ is a correction parameter, $\mathbf{g}_t$ and $\widetilde{\mathbf{g}}_t$ are defined in Eq. (2.2) and $\mathbf{w}_{t,i}$ denotes the decision of the $i$-th base learner at round $t$.

**Overall Algorithm and Dynamic Regret Analysis.** Integrating the aforementioned meta-base structure, we propose Algorithm 5 for gradient-variation dynamic regret minimization in two-point BCO, which attains the performance guarantee established in Theorem 9.

**Theorem 9.** *Under Assumptions 1-3, with*

$$\varepsilon_t = \sqrt{\frac{\ln N}{C_0^2 + \sum_{s=1}^{t-1} \|\boldsymbol{\ell}_s - \boldsymbol{m}_s\|_\infty^2}}, \quad C_0 = 16R^2 L\sqrt{d^3 \log(dT)\ln N}, \quad \gamma = 5L\sqrt{d^3 \log(dT)},$$

*for any time-varying comparators $\mathbf{v}_1, \ldots, \mathbf{v}_T \in \mathcal{X}$, Algorithm 5 achieves the following dynamic regret:*

$$\mathbb{E}\left[\text{D-Reg}_T\right] \leq \widetilde{\mathcal{O}}\left(\sqrt{d^3(1 + P_T + V_T)(1 + P_T)}\right).$$

---

**Algorithm 5** Dynamic Gradient-Variation Regret Minimization with Two-Point Feedback

1: **Input:** Base learner number $N$
2: **Initialize:** $\mathcal{A}$ — meta learner running Optimistic-Hedge (E.3);
   $\{\mathcal{B}_i\}_{i \in [N]}$ — the $i$-th base learner runs Algorithm 1 with step size in Eq. (E.1)
3: **for** $t = 1$ **to** $T$ **do**
4:     Choose $i_t$ uniformly from $[d]$.
5:     Submit $\mathbf{x}_t = \mathbf{w}_t + \delta \mathbf{e}_{i_t}$, and $\mathbf{x}'_t = \mathbf{w}_t - \delta \mathbf{e}_{i_t}$, where $\mathbf{w}_t = \sum_{i=1}^{N} p_{t,i} \mathbf{w}_{t,i}$
6:     Observe partial information $f_t(\mathbf{x}_t)$ and $f_t(\mathbf{x}'_t)$
7:     Construct the gradient estimator $\mathbf{g}_t$ and the optimism $\widetilde{\mathbf{g}}_t$ as in Eq. (2.2)
8:     $\mathcal{A}$ updates with $\ell_{t,i}$ and $m_{t,i}$ as constructed in Eq. (E.4) to get $\boldsymbol{p}_{t+1}$
9:     $\mathcal{B}_i$ updates to $\mathbf{w}_{t+1,i}$ with gradient estimator $\mathbf{g}_t$ and optimism $\widetilde{\mathbf{g}}_t$
10: **end for**

---

**Proof** [of Theorem 9] To begin with, we decompose the regret as follows:

$$\mathbb{E}\left[\text{D-REG}_T\right] = \underbrace{\mathbb{E}\left[\sum_{t=1}^{T} \langle \mathbf{g}_t, \mathbf{w}_t - \mathbf{w}_{t,i^\star} \rangle\right]}_{\text{META-REG}} + \underbrace{\mathbb{E}\left[\sum_{t=1}^{T} \langle \mathbf{g}_t, \mathbf{w}_{t,i^\star} - \mathbf{v}_t \rangle\right]}_{\text{BASE-REG}} + \mathcal{O}(1).$$

Denote by

$$S_{T,i}^{\mathbf{w}} = \sum_{t=2}^{T} \|\mathbf{w}_{t,i} - \mathbf{w}_{t-1,i}\|^2, \ S_T^{\boldsymbol{p}} = \sum_{t=2}^{T} \|\boldsymbol{p}_t - \boldsymbol{p}_{t-1}\|_1^2, S_T^{\mathbf{w}} = \sum_{t=2}^{T} \|\mathbf{w}_t - \mathbf{w}_{t-1}\|^2, \ S_T^{\text{MIX}} = \sum_{t=2}^{T} \sum_{i=1}^{N} p_{t,i} \|\mathbf{w}_{t,i} - \mathbf{w}_{t-1,i}\|^2.$$

Then, by Yan et al. (2023, Lemma 6), we have

$$S_T^{\mathbf{w}} \le 2 S_T^{\text{MIX}} + 2R^2 S_T^{\boldsymbol{p}}. \tag{E.5}$$

**Meta Regret Analysis.** For the meta regret, by Lemma 15, we have

$$\text{META-REG} = \mathbb{E}\left[\sum_{t=1}^{T} \langle \boldsymbol{\ell}_t, \boldsymbol{p}_t - \mathbf{e}_{i^\star} \rangle\right] + \gamma \mathbb{E}\left[S_{T,i^\star}^{\mathbf{w}}\right] - \gamma \mathbb{E}\left[S_T^{\text{MIX}}\right]$$

$$\le 5\sqrt{\ln N \left(C_0^2 + \mathbb{E}\left[\sum_{t=1}^{T} \|\boldsymbol{\ell}_t - \boldsymbol{m}_t\|_\infty^2\right]\right)} + \gamma \mathbb{E}\left[S_{T,i^\star}^{\mathbf{w}}\right] - \gamma \mathbb{E}\left[S_T^{\text{MIX}}\right] - \frac{C_0}{4\sqrt{\ln N}} \mathbb{E}\left[S_T^{\boldsymbol{p}}\right] + \frac{\sqrt{\ln N}}{C_0} \mathbb{E}\left[\max_{t \in [T]} \|\boldsymbol{\ell}_t - \boldsymbol{m}_t\|_\infty^2\right]$$

$$\le 5\sqrt{\ln N \left(C_0^2 + R^2 \mathbb{E}\left[\sum_{t=1}^{T} \|\mathbf{g}_t - \widetilde{\mathbf{g}}_t\|_2^2\right]\right)} + \gamma \mathbb{E}\left[S_{T,i^\star}^{\mathbf{w}}\right] - \gamma \mathbb{E}\left[S_T^{\text{MIX}}\right] - \frac{C_0}{4\sqrt{\ln N}} \mathbb{E}\left[S_T^{\boldsymbol{p}}\right] + \frac{4d^2 G^2 R^2 \sqrt{\ln N}}{C_0},$$

where the second step is by Lemma 15, and the last step follows from the definition of $\boldsymbol{\ell}_t$ and $\boldsymbol{m}_t$ in Eq. (E.4), property *(ix)* of Lemma 6, and the boundedness $\|\mathbf{x}\| \le R$.

**Base Regret Analysis.** For the base regret, substituting $\mathbf{g}_t$ and $\widetilde{\mathbf{g}}_t$ into $\nabla f_t(\mathbf{x}_t)$ and $M_t$ in Lemma 14, we have

$$\text{BASE-REG} \le \eta_{i^\star} \mathbb{E}\left[\sum_{t=1}^{T} \|\mathbf{g}_t - \widetilde{\mathbf{g}}_t\|_2^2\right] + \frac{4}{\eta_{i^\star}} \left(R^2 + R P_T\right) - \frac{1}{4\eta_{i^\star}} \mathbb{E}\left[S_{T,i^\star}^{\mathbf{w}}\right],$$

where $P_T = \sum_{t=2}^{T} \|\mathbf{v}_t - \mathbf{v}_{t-1}\|$.

**Overall Regret Analysis.** Combining the meta regret and base regret, we have

$$\mathbb{E}\left[\text{D-REG}_T\right] \le 5\sqrt{\ln N \left(C_0^2 + R^2 \mathbb{E}\left[\sum_{t=1}^{T} \|\mathbf{g}_t - \widetilde{\mathbf{g}}_t\|_2^2\right]\right)} + \eta_{i^\star} \mathbb{E}\left[\sum_{t=1}^{T} \|\mathbf{g}_t - \widetilde{\mathbf{g}}_t\|_2^2\right] + \frac{4}{\eta_{i^\star}}(R^2 + R P_T)$$

$$+ \left(\gamma - \frac{1}{4\eta_{i^\star}}\right) \mathbb{E}\left[S_{T,i^\star}^{\mathbf{w}}\right] - \gamma\mathbb{E}\left[S_T^{\mathrm{MIX}}\right] - \frac{C_0}{4\sqrt{\ln N}}\mathbb{E}\left[S_T^{\boldsymbol{p}}\right] + \mathcal{O}\left(\frac{4d^2 G^2 R^2 \sqrt{\ln N}}{C_0}\right)$$

$$\leq 5\sqrt{\ln N\left(C_0^2 + 8d^3 R^2 V_T \log d + 16d^3 L^2 R^2 \log(dT)\mathbb{E}\left[S_T^{\mathbf{w}}\right]\right)}$$
$$+ 8d^3 \eta_{i^\star} V_T \log d + 16d^3 L^2 \log(dT)\eta_{i^\star}\mathbb{E}\left[S_T^{\mathbf{w}}\right] + \frac{4}{\eta_{i^\star}}\left(R^2 + RP_T\right)$$
$$+ \left(\gamma - \frac{1}{4\eta_{i^\star}}\right) \mathbb{E}\left[S_{T,i^\star}^{\mathbf{w}}\right] - \gamma\mathbb{E}\left[S_T^{\mathrm{MIX}}\right] - \frac{C_0}{4\sqrt{\ln N}}\mathbb{E}\left[S_T^{\boldsymbol{p}}\right] + \mathcal{O}\left(\frac{4d^2 G^2 R^2 \sqrt{\ln N}}{C_0}\right)$$

$$\leq 5\sqrt{\ln N\left(C_0^2 + 8d^3 R^2 V_T \log d\right)} + 20\sqrt{\ln N}\sqrt{d^3 L^2 R^2 \log(dT)\mathbb{E}\left[S_T^{\mathbf{w}}\right]}$$
$$+ 8d^3 \eta_{i^\star} V_T \log d + 16d^3 L^2 \log(dT)\eta_{i^\star}\mathbb{E}\left[S_T^{\mathbf{w}}\right] + \frac{4}{\eta_{i^\star}}\left(R^2 + RP_T\right)$$
$$+ \left(\gamma - \frac{1}{4\eta_{i^\star}}\right) \mathbb{E}\left[S_{T,i^\star}^{\mathbf{w}}\right] - \gamma\mathbb{E}\left[S_T^{\mathrm{MIX}}\right] - \frac{C_0}{4\sqrt{\ln N}}\mathbb{E}\left[S_T^{\boldsymbol{p}}\right] + \mathcal{O}\left(\frac{4d^2 G^2 R^2 \sqrt{\ln N}}{C_0}\right)$$

$$\leq \mathcal{O}\left(\frac{4d^2 G^2 R^2 \sqrt{\ln N}}{C_0} + \sqrt{d^3 V_T \ln N \log d}\right) + 8d^3 \eta_{i^\star} V_T \log d + \frac{20R^2 \ln N}{\eta_{i^\star}}$$
$$+ \left(16d^3 L^2 \log(dT)\eta_{i^\star} + 20d^3 L^2 \log(dT)\eta_{i^\star}\right)\mathbb{E}\left[S_T^{\mathbf{w}}\right] + \frac{4}{\eta_{i^\star}}\left(R^2 + RP_T\right)$$
$$+ \left(\gamma - \frac{1}{4\eta_{i^\star}}\right) \mathbb{E}\left[S_{T,i^\star}^{\mathbf{w}}\right] - \gamma\mathbb{E}\left[S_T^{\mathrm{MIX}}\right] - \frac{C_0}{4\sqrt{\ln N}}\mathbb{E}\left[S_T^{\boldsymbol{p}}\right]$$

$$\leq \mathcal{O}\left(\frac{4d^2 G^2 R^2 \sqrt{\ln N}}{C_0} + \sqrt{d^3 \ln N\left(V_T \log d + \ln N \log(dT)\right)}\right) + 8d^3 \eta_{i^\star} V_T \log d$$
$$+ \left(32d^3 L^2 \log(dT)\eta_{i^\star} + 40d^3 L^2 \log(dT)\eta_{i^\star} - \gamma\right)\mathbb{E}\left[S_T^{\mathrm{MIX}}\right]$$
$$+ \left(32d^3 L^2 R^2 \log(dT)\eta_{i^\star} + 40d^3 L^2 R^2 \log(dT)\eta_{i^\star} - \frac{C_0}{4\sqrt{\ln N}}\right)\mathbb{E}\left[S_T^{\boldsymbol{p}}\right]$$
$$+ \left(\gamma - \frac{1}{4\eta_{i^\star}}\right) \mathbb{E}\left[S_{T,i^\star}^{\mathbf{w}}\right] + \frac{4}{\eta_{i^\star}}\left(R^2 + RP_T\right)$$

$$\leq \mathcal{O}\left(\sqrt{d^3\left(V_T \log d + \log(dT)\right)} + \frac{R^2 + RP_T}{\eta_{i^\star}} + d^3 \eta_{i^\star} V_T \log d\right),$$

where the second step is by Lemma 2, the third step is by AM-GM inequality, the fourth step is by Eq. (E.5), and the last step is by choosing

$$\gamma = 5L\sqrt{d^3 \log(dT)}, \ \eta_{i^\star} \leq \frac{1}{20L\sqrt{d^3 \log(dT)}}, \ \text{and} \ C_0 = 16R^2 L\sqrt{d^3 \log(dT)\ln N}.$$

Then, we define $\eta^\dagger = \min\left\{\frac{1}{20L\sqrt{d^3 \log(dT)}}, \sqrt{\frac{R^2 + RP_T}{d^3 V_T \log d}}\right\}$ and choose $\eta_{i^\star} \leq \eta^\dagger \leq 2\eta_{i^\star}$. Thus, we have

$$\mathbb{E}\left[\text{D-Reg}_T\right] \leq \mathcal{O}\left(\sqrt{d^3\left(V_T \log d + \log(dT)\right)} + \sqrt{d^3 V_T P_T \log d} + \sqrt{d^3 \log(dT)}\left(R^2 + RP_T\right)\right)$$
$$\leq \mathcal{O}\left(\sqrt{d^3(\log(dT) + \log(dT)P_T + V_T \log d)(1 + P_T)}\right),$$

which finishes the proof. $\blacksquare$

Finally, for the sake of completeness, we provide the proof of Lemma 15.

**Proof** [of Lemma 15] First, we introduce a useful lemma about the regret guarantee of Optimistic-Hedge.

**Lemma 16** (Theorem 7.47 of Orabona (2019))**.** *The regret of* **Optimistic-Hedge** (E.3) *with a decreasing learning rate*

$\varepsilon_t > 0$ *to any expert* $i \in [N]$ *satisfies*

$$\sum_{t=1}^{T} \langle \boldsymbol{p}_t, \boldsymbol{\ell}_t \rangle - \sum_{t=1}^{T} \ell_{t,i} \leq \max_{\boldsymbol{p} \in \Delta_N} \psi_{T+1}(\boldsymbol{p}) + \sum_{t=1}^{T} \langle \boldsymbol{\ell}_t - \boldsymbol{m}_t, \boldsymbol{p}_t - \boldsymbol{p}_{t+1} \rangle - \sum_{t=1}^{T} \frac{1}{2\varepsilon_{t-1}} \|\boldsymbol{p}_t - \boldsymbol{p}_{t+1}\|_1^2,$$

*where* $\psi_{t+1}(\boldsymbol{p}) = \frac{1}{\varepsilon_t} \left( \sum_{i=1}^{N} p_i \log p_i + \ln N \right)$.

By Lemma 16, we have

$$\sum_{t=1}^{T} \langle \boldsymbol{p}_t, \boldsymbol{\ell}_t \rangle - \sum_{t=1}^{T} \ell_{t,i} \leq \frac{\ln N}{\varepsilon_T} + \sum_{t=1}^{T} \varepsilon_{t-1} \|\boldsymbol{\ell}_t - \boldsymbol{m}_t\|_\infty^2 - \sum_{t=1}^{T} \frac{1}{4\varepsilon_{t-1}} \|\boldsymbol{p}_t - \boldsymbol{p}_{t+1}\|_1^2$$

$$\leq 5 \sqrt{\ln N \left( C_0^2 + \sum_{t=1}^{T} \|\boldsymbol{\ell}_t - \boldsymbol{m}_t\|_\infty^2 \right)} - \sum_{t=1}^{T} \frac{C_0}{4\sqrt{\ln N}} \|\boldsymbol{p}_t - \boldsymbol{p}_{t+1}\|_1^2 + \frac{\sqrt{\ln N}}{C_0} \max_{t \in [T]} \|\boldsymbol{\ell}_t - \boldsymbol{m}_t\|_\infty^2.$$

where the last step is by Lemma 22. ∎

### E.2. Omitted Details in Section 5.2

In this part, we provide the omitted details for our applications in universal regret. Following the online ensemble framework in Yan et al. (2024), we design new surrogate loss functions to accommodate different types of functions, including strongly convex, convex, and linear functions. Specifically, we instantiate multiple base learners for strongly convex functions with varying strong convexity parameters from a predefined pool, along with one base learner each for convex and linear functions. The meta-algorithm aggregates the decisions of all base learners using the Optimistic-Adapt-ML-Prod algorithm (Wei et al., 2016).

We first provide a decomposition of the non-consecutive gradient variation $\bar{V}_T$ (2.1) for linear functions, analogous to that in Lemma 5.

**Lemma 17** (Lemmas 7,9 of Chiang et al. (2013)). *Under Assumptions 1-3, for linear functions, Algorithm 1 satisfies:*

$$\mathbb{E}\left[\bar{V}_T\right] \leq 2d^3 V_T + \mathcal{O}(1).$$

Next, similar to the dynamic regret analysis in Appendix E.1, we elaborate on the online scheduling strategy, the base learners, and the meta-algorithm in the following.

**Online Scheduling Strategy and Base Learners.**  To tackle different types of functions, we design multiple base learners as instances of Algorithm 1 with distinct step size schedules. To address the unknown strong convexity $\lambda$, we employ a set of learners with candidate parameters from a predefined pool, in addition to single base learners for linear and convex function classes. Specifically, for the nondegenerate case of $\lambda \in [1/T, 1]$, we can discretize the unknown $\lambda$ into a candidate pool $\mathcal{H}^{\mathrm{sc}}$ using an exponential grid, defined as

$$\mathcal{H}^{\mathrm{sc}} \triangleq \left\{ \frac{1}{T}, \frac{2}{T}, \frac{2^2}{T}, \cdots, \frac{2^{N_{\mathrm{sc}}-1}}{T} \right\}, \tag{E.6}$$

where $N_{\mathrm{sc}} = \lceil \log_2 T \rceil + 1 = \mathcal{O}(\log T)$ is the number of candidates. It can be proved that the discretized candidate pool $\mathcal{H}^{\mathrm{sc}}$ can approximate the continuous value of $\lambda$ with only constant errors. In total, there are $N \triangleq 2 + |\mathcal{H}^{\mathrm{sc}}| = 2 + N_{\mathrm{sc}} = \mathcal{O}(\log T)$ base learners, where the first $N_{\mathrm{sc}}$ base learners correspond to strongly convex functions with different guessed strong convexity coefficients from $\mathcal{H}^{\mathrm{sc}}$, and the last two base learners correspond to convex and linear functions, respectively. The best base learner is the one with the right guess of the curvature type and the closest guess of the curvature coefficient. For example, suppose the online functions are $\lambda$-strongly convex (while this is unknown to the online learner), then the right guessed coefficient of the best base learner (indexed by $i^\star$) satisfies $\lambda_{i^\star} \leq \lambda \leq 2\lambda_{i^\star}$.

To handle limited information, we devise surrogate loss functions for the various base learner categories. This allows us to treat each base learner as an OOGD (E.2) instance operating on the constructed surrogate feedback at each round $t$. Specifically, the configurations of base learners (surrogate loss functions and OOGD step sizes) are summarized in Table 2.

*Table 2.* Summary of Base Learner Configurations.

| | Strongly Convex ($\lambda_i$) | Convex | Linear |
|---|---|---|---|
| **Surrogate Loss** | $h_{t,i}^{\mathrm{sc}}(\mathbf{x}) \triangleq \langle \mathbf{g}_t, \mathbf{x} \rangle + \frac{\lambda_i}{4}\|\mathbf{x} - \mathbf{w}_t\|^2$ | $h_t^{\mathrm{cvx}}(\mathbf{x}) \triangleq \langle \mathbf{g}_t, \mathbf{x} \rangle$ | $h_t^{\mathrm{lin}}(\mathbf{x}) \triangleq \langle \mathbf{g}_t, \mathbf{x} \rangle$ |
| **OOGD Step Size** | $\eta_t = \frac{4}{\lambda_i t}$ | $\eta_t = \frac{2R}{\sqrt{d^2 + \sum_{s=1}^{t-1}\|\mathbf{g}_s - \widetilde{\mathbf{g}}_s\|^2}}$ | $\eta_t = \frac{2R}{\sqrt{d^2 + \sum_{s=1}^{t-1}\|\mathbf{g}_s - \widetilde{\mathbf{g}}_s\|^2}}$ |

Here, we present the regret guarantees of OOGD for convex and strongly convex functions, which will be utilized in our analysis of universal regret.

**Lemma 18** (Lemma 10 of Yan et al. (2023)). *Under Assumptions 1-3, if the loss functions are convex,* OOGD (E.2) *with* $\eta_t = 2R/\sqrt{\kappa^2 + A_{t-1}}$, *where $\kappa$ is a parameter to be specified, enjoys the following regret guarantee: for any $\mathbf{v} \in \mathcal{X}$*

$$\sum_{t=1}^{T} f_t(\mathbf{x}_t) - \sum_{t=1}^{T} f_t(\mathbf{v}) \le 10R\sqrt{\kappa^2 + A_T} + \kappa R - \frac{\kappa}{4}\sum_{t=2}^{T}\|\mathbf{x}_t - \mathbf{x}_{t-1}\|^2 + \frac{1}{\kappa}\mathcal{O}\left(\max_{t \in [T]}\|\nabla f_t(\mathbf{x}_t) - M_t\|^2\right),$$

*where $A_t \triangleq \sum_{s=1}^{t}\|\nabla f_s(\mathbf{x}_s) - M_s\|^2$ for any $t \in [T]$.*

**Lemma 19** (Lemma 12 of Yan et al. (2023)). *Under Assumptions 1-3, if the loss functions are $\lambda$-strongly convex,* OOGD (E.2) *with $\eta_t = 2/(\lambda t)$ enjoys the following regret guarantee: for any $\mathbf{v} \in \mathcal{X}$*

$$\sum_{t=1}^{T} f_t(\mathbf{x}_t) - \sum_{t=1}^{T} f_t(\mathbf{v}) \le 2\sum_{t=1}^{T}\eta_t\|\nabla f_t(\mathbf{x}_t) - M_t\|^2 + \mathcal{O}(1).$$

**Meta-algorithm: Optimistic-Adapt-ML-Prod.** We introduce the Optimistic-Adapt-ML-Prod algorithm as our meta algorithm to combine the decisions of all base learners.

Specifically, the weight vector $\boldsymbol{p}_{t+1} \in \Delta_N$ is updated as follows: $\forall i \in [N]$,

$$\forall t \ge 1, W_{t,i} = \left(W_{t-1,i} \cdot \exp\left(\varepsilon_{t-1,i}r_{t,i} - \varepsilon_{t-1,i}^2(r_{t,i} - m_{t,i})^2\right)\right)^{\frac{\varepsilon_{t,i}}{\varepsilon_{t-1,i}}}, \text{ and } W_{0,i} = \frac{1}{N}, \tag{E.7}$$
$$p_{t+1,i} \propto \varepsilon_{t,i} \cdot \exp(\varepsilon_{t,i}m_{t+1,i}) \cdot W_{t,i},$$

where $W_{t,i}$ and $\varepsilon_{t,i}$ denote the potential variable and learning rate for the $i$-th base learner, respectively. The feedback loss vector $\boldsymbol{\ell}_t \in \mathbb{R}^N$ is configured as $\ell_{t,i} \triangleq \frac{1}{2\sqrt{10}dGR}\langle \mathbf{g}_t, \mathbf{w}_{t,i} \rangle + \frac{1}{2} \in [0, 1]$, where by property *(vi)* of Lemma 6, $\|\mathbf{g}_t\| \le \sqrt{10}dG$. $r_{t,i} = \langle \boldsymbol{p}_t, \boldsymbol{\ell}_t \rangle - \ell_{t,i}$ measures the instantaneous regret. The optimistic vector $\boldsymbol{m}_t \in \mathbb{R}^N$ is designed as[1]

$$m_{t,i} = 0 \text{ for } i \in [N_{\mathrm{sc}}], \text{ and } m_{t,i} = \frac{1}{2\sqrt{10}dGR}\langle \widetilde{\mathbf{g}}_t, \mathbf{w}_t - \mathbf{w}_{t,i} \rangle \text{ for } i \in \{N-1, N\}, \tag{E.8}$$

where the last two entries correspond to the convex and linear base learners, respectively.

For any $i \in [N]$, the learning rate $\varepsilon_{t,i}$ is set as

$$\varepsilon_{t,i} = \min\left\{\frac{1}{8}, \sqrt{\frac{\log N}{\sum_{s=1}^{t}(r_{s,i} - m_{s,i})^2}}\right\}. \tag{E.9}$$

Here, we present the guarantee of Optimistic-Adapt-ML-Prod, which will be utilized in our analysis of universal regret.

**Lemma 20** (Theorem 3.4 of Wei et al. (2016)). *Denote by $\boldsymbol{p}_t \in \Delta_N$ the algorithm's weights, $\boldsymbol{\ell}_t \in [0, 1]^N$ the loss vector, and $m_{t,i}$ the optimism. With the learning rate in (E.9), the regret of Optimistic-Adapt-ML-Prod (E.7) with respect to any expert $i \in [N]$ satisfies*

$$\sum_{t=1}^{T}\langle \boldsymbol{\ell}_t, \boldsymbol{p}_t - \mathbf{e}_i \rangle \le C_1\sqrt{1 + \sum_{t=1}^{T}(r_{t,i} - m_{t,i})^2} + C_2,$$

---

[1]While $\mathbf{w}_t$ depends on $\boldsymbol{m}_{t,i}$, the update only requires the scalar $\langle \widetilde{\mathbf{g}}_t, \mathbf{w}_t \rangle$, which can be obtained by solving the one-dimensional fixed-point problem $\langle \widetilde{\mathbf{g}}_t, \mathbf{w}_t(z) \rangle = z$. Here, the function $\mathbf{w}_t(z)$ is induced by the dependency chain $\mathbf{w}_t(p_{t,i}(\boldsymbol{m}_{t,i}(z)))$. For a comprehensive derivation, see Wei et al. (2016)

---

**Algorithm 6** Universal Algorithm for Gradient-Variation Regret in Two-Point BCO

---

**Input:** Base learner configurations $\{\mathcal{B}_i\}_{i \in [N]} \triangleq \{\mathcal{B}_i^{sc}\}_{i \in [N_{sc}]} \cup \mathcal{B}^{cvx} \cup \mathcal{B}^{lin}$

1: **Initialize**: $\mathcal{M}$ — meta learner Optimistic-Adapt-ML-Prod with $W_{0,i} = \frac{1}{N}$ for all $i \in [N]$
2:     $\{\mathcal{B}_i\}_{i \in [N]}$ — base learners as specified in Appendix E.2
3: **for** $t = 1$ **to** $T$ **do**
4:     Choose $i_t$ uniformly from $[d]$.
5:     Submit $\mathbf{x}_t = \mathbf{w}_t + \delta \mathbf{e}_{i_t}$, and $\mathbf{x}_t' = \mathbf{w}_t - \delta \mathbf{e}_{i_t}$, where $\mathbf{w}_t = \sum_{i=1}^{N} p_{t,i} \mathbf{w}_{t,i}$
6:     Observe partial information $f_t(\mathbf{x}_t)$ and $f_t(\mathbf{x}_t')$
7:     Construct the gradient estimator $\mathbf{g}_t$ and the optimism $\widetilde{\mathbf{g}}_t$ as in (2.2)
8:     $\{\mathcal{B}_i^{sc}\}_{i \in [N_{sc}]}$, $\mathcal{B}^{cvx}$, and $\mathcal{B}^{lin}$ update their own decisions to $\{\mathbf{w}_{t+1,i}\}_{i=1}^{N}$ using surrogate losses of $\{h_{t,i}^{sc}(\cdot)\}_{\lambda_i \in \mathcal{H}^{sc}}$,
    $h_t^{cvx}(\cdot)$, and $h_t^{lin}(\cdot)$ in Table 2
9:     Calculate $\boldsymbol{m}_{t+1}$ (E.8) and $\mathbf{r}_t$ using $\{\mathbf{w}_{t,i}\}_{i=1}^{N}$, $\mathbf{w}_t$, $\mathbf{g}_t$, $\widetilde{\mathbf{g}}_t$, and $\{\mathbf{w}_{t+1,i}\}_{i=1}^{N}$, send them to $\mathcal{M}$, and obtain $\boldsymbol{p}_{t+1}$
10: **end for**

---

where $\mathbf{e}_i$ is the $i$-th standard basis vector, $C_1 = \sqrt{\log N} + \log(1 + \frac{N}{e}(1 + \log(T+1)))/\sqrt{\log N}$, and $C_2 = \frac{1}{4}(\log N + \log(1 + \frac{N}{e}(1 + \log(T+1)))) + 2\sqrt{\log N} + 16 \log N$.

Since the number of base learners is $N = \mathcal{O}(\log T)$, the constants $C_1$ and $C_2$ are of order $\mathcal{O}(\log \log T)$ and can be ignored.

**Overall Algorithm and Universal Regret Analysis.** Integrating the above components, we propose Algorithm 6 for universal regret minimization in two-point BCO, which attains the performance guarantee established in Theorem 10.

**Theorem 10.** *Under Assumptions 1, 2, 4, Algorithm 6 achieves the following universal regret:*

$$\mathbb{E}[\text{REG}_T] \leq \begin{cases} \mathcal{O}\left(\frac{d}{\lambda} \log(dV_T)\right), & (\lambda\text{-strongly convex case}), \\ \widetilde{\mathcal{O}}\left(\sqrt{d^3 V_T} + d^3\right), & (\text{convex case}), \\ \mathcal{O}\left(\sqrt{d^3 V_T}\right), & (\text{linear case}). \end{cases}$$

**Remark 4.** In contrast to the full-information setting, where universal guarantees are commonly studied for convex, exp-concave, and strongly convex functions, we consider linear, convex, and strongly convex functions under two-point bandit feedback. On the one hand, linear functions constitute a fundamental function class that has been extensively studied in the bandit literature. On the other hand, exp-concave functions are substantially more challenging under bandit feedback, as exploiting their curvature structure typically requires gradient information. Indeed, obtaining worst-case regret guarantees for exp-concave functions with two-point feedback remains an open problem.

We first present a key lemma that is crucial for analyzing the meta-regret of Optimistic-Adapt-ML-Prod.

**Lemma 21.** *Under Assumptions 1-3, with the same exploration parameter as in Lemma 6, for the best strongly convex base learner indexed by $i^\star$ in Algorithm 6, we have*

$$\mathbb{E}\left[\langle \mathbf{g}_t, \mathbf{w}_t - \mathbf{w}_{t,i^\star}\rangle^2\right] \leq 10 d G^2 \mathbb{E}\left[\|\mathbf{w}_t - \mathbf{w}_{t,i^\star}\|^2\right].$$

**Proof** We first give different regret decompositions, then analyze the meta regret, and finally provide the proofs for different kinds of loss functions.

**Regret Decomposition.** For $\lambda$-*strongly convex* functions, we have the following decomposition:

$$\mathbb{E}[\text{REG}_T] \leq \underbrace{\mathbb{E}\left[\sum_{t=1}^{T}\langle \mathbf{g}_t, \mathbf{w}_t - \mathbf{w}_{t,i^\star}\rangle - \frac{\lambda_{i^\star}}{4}\sum_{t=1}^{T}\|\mathbf{w}_t - \mathbf{w}_{t,i^\star}\|^2\right]}_{\text{META-REG}} \qquad (\text{by } \lambda_{i^\star} \leq \lambda \leq 2\lambda_{i^\star})$$

$$+ \underbrace{\mathbb{E}\left[\sum_{t=1}^{T} h_{t,i^\star}^{sc}(\mathbf{w}_{t,i^\star}) - \sum_{t=1}^{T} h_{t,i^\star}^{sc}(\mathbf{w}^\star)\right]}_{\text{BASE-REG}} - \frac{1}{2}\mathbb{E}\left[\sum_{t=1}^{T} \mathcal{D}_{f_t}(\mathbf{w}^\star, \mathbf{w}_t)\right] + \mathcal{O}(1), \qquad (\text{E.10})$$

due to the definition of the surrogate $h_{t,i}^{\mathrm{sc}}(\mathbf{x}) \triangleq \langle \mathbf{g}_t, \mathbf{x} \rangle + \frac{\lambda_i}{4} \|\mathbf{x} - \mathbf{w}_t\|^2$, where $\lambda_i \in \mathcal{H}^{\mathrm{sc}}$ in (E.6). For *convex* functions, we decompose the regret as

$$
\mathbb{E}[\mathrm{REG}_T] \leq \mathbb{E}\left[\sum_{t=1}^T \langle \mathbf{g}_t, \mathbf{w}_t - \mathbf{w}^\star \rangle - \sum_{t=1}^T \mathcal{D}_{f_t}(\mathbf{w}^\star, \mathbf{w}_t)\right] + \mathcal{O}(1)
$$

$$
= \underbrace{\mathbb{E}\left[\sum_{t=1}^T \langle \mathbf{g}_t, \mathbf{w}_t - \mathbf{w}_{t,i^\star} \rangle\right]}_{\text{META-REG}} + \underbrace{\mathbb{E}\left[\sum_{t=1}^T h_{t,i^\star}^{\mathrm{cvx}}(\mathbf{w}_{t,i^\star}) - \sum_{t=1}^T h_{t,i^\star}^{\mathrm{cvx}}(\mathbf{w}^\star)\right]}_{\text{BASE-REG}} - \mathbb{E}\left[\sum_{t=1}^T \mathcal{D}_{f_t}(\mathbf{w}^\star, \mathbf{w}_t)\right] + \mathcal{O}(1), \qquad \text{(E.11)}
$$

where $h_{t,i}^{\mathrm{cvx}}(\mathbf{x}) \triangleq \langle \mathbf{g}_t, \mathbf{x} \rangle$.

For *linear* functions, since $\mathcal{D}_{f_t}(\mathbf{w}^\star, \mathbf{w}_t) = 0$ for $t \in [T]$, we decompose the regret as

$$
\mathbb{E}[\mathrm{REG}_T] \leq \underbrace{\mathbb{E}\left[\sum_{t=1}^T \langle \mathbf{g}_t, \mathbf{w}_t - \mathbf{w}_{t,i^\star} \rangle\right]}_{\text{META-REG}} + \underbrace{\mathbb{E}\left[\sum_{t=1}^T h_{t,i^\star}^{\mathrm{lin}}(\mathbf{w}_{t,i^\star}) - \sum_{t=1}^T h_{t,i^\star}^{\mathrm{lin}}(\mathbf{w}^\star)\right]}_{\text{BASE-REG}} + \mathcal{O}(1), \qquad \text{(E.12)}
$$

where $h_{t,i}^{\mathrm{lin}}(\mathbf{x}) \triangleq \langle \mathbf{g}_t, \mathbf{x} \rangle$.

**Meta Regret Analysis.** For $\lambda$-*strongly convex* functions, we have

$$
\mathrm{META\text{-}REG} = \mathbb{E}\left[4\sqrt{10}dGR \sum_{t=1}^T \langle \boldsymbol{\ell}_t, \boldsymbol{p}_t - \mathbf{e}_{i^\star} \rangle - \frac{\lambda_{i^\star}}{4} \sum_{t=1}^T \|\mathbf{w}_t - \mathbf{w}_{t,i^\star}\|^2\right]
$$

$$
\leq C_1 \mathbb{E}\left[\sqrt{160 d^2 G^2 R^2 + \sum_{t=1}^T \langle \mathbf{g}_t, \mathbf{w}_t - \mathbf{w}_{t,i^\star} \rangle^2}\right] - \frac{\lambda_{i^\star}}{4} \mathbb{E}\left[\sum_{t=1}^T \|\mathbf{w}_t - \mathbf{w}_{t,i^\star}\|^2\right] + 2\sqrt{10}dGRC_2
$$

$$
\leq \mathcal{O}(d) + C_1 \sqrt{\sum_{t=1}^T 10 dG^2 \mathbb{E}\left[\|\mathbf{w}_t - \mathbf{w}_{t,i^\star}\|^2\right]} - \frac{\lambda_{i^\star}}{4} \mathbb{E}\left[\sum_{t=1}^T \|\mathbf{w}_t - \mathbf{w}_{t,i^\star}\|^2\right]
$$

$$
\leq \mathcal{O}(d) + \left(\frac{10 C_1 G^2}{C_3} - \frac{\lambda_{i^\star}}{4}\right) \mathbb{E}\left[\sum_{t=1}^T \|\mathbf{w}_t - \mathbf{w}_{t,i^\star}\|^2\right], \qquad \text{(E.13)}
$$

where the first step is by definition, the second step is by Lemma 20, the third step is by Lemma 21, and the last step is due to AM-GM inequality. $C_3$ is a *d-independent* constant to be specified.

For *convex* functions, we bound the meta regret by

$$
\mathrm{META\text{-}REG} \leq C_1 \sqrt{160 d^2 G^2 R^2 + \mathbb{E}\left[\sum_{t=1}^T \langle \mathbf{g}_t - \widetilde{\mathbf{g}}_t, \mathbf{w}_t - \mathbf{w}_{t,i^\star} \rangle^2\right]} + 2\sqrt{10}dGRC_2
$$

$$
\leq C_1 \sqrt{160 d^2 G^2 R^2 + 48 d^3 R^2 V_T \log d + 192 d^3 L R^2 \log(dT) \mathbb{E}\left[\sum_{t=1}^T \mathcal{D}_{f_t}(\mathbf{w}^\star, \mathbf{w}_t)\right]} + 2\sqrt{10}dGRC_2
$$

$$
\leq \mathcal{O}(\sqrt{d^3 V_T \log d}) + C_1 \sqrt{192 d^3 L R^2 \log(dT) \mathbb{E}\left[\sum_{t=1}^T \mathcal{D}_{f_t}(\mathbf{w}^\star, \mathbf{w}_t)\right]}
$$

$$
\leq \mathcal{O}(\sqrt{d^3 V_T \log d}) + \mathcal{O}(C_4 d^3 \log(dT)) + \frac{C_1}{2C_4} \mathbb{E}\left[\sum_{t=1}^T \mathcal{D}_{f_t}(\mathbf{w}^\star, \mathbf{w}_t)\right], \qquad \text{(E.14)}
$$

where the first step is by Lemma 20 and Jensen's inequality, the second step is by Lemma 5 and $\|\mathbf{w}_t - \mathbf{w}_{t,i^\star}\| \leq 2R$, the third step uses $\sqrt{a+b} \leq \sqrt{a} + \sqrt{b}$ for any $a, b \geq 0$, the last step uses AM-GM inequality: $\sqrt{ab} \leq \frac{ax}{2} + \frac{b}{2x}$ for any

$a, b, x > 0$. Note that the $d, T$-independent constant $C_4$ is used to ensure that the Bregman divergence term is canceled and will be specified in the end.

For *linear* functions, we bound the meta regret by

$$\text{META-REG} \leq C_1 \sqrt{160d^2G^2R^2 + \mathbb{E}\left[\sum_{t=1}^T \langle \mathbf{g}_t - \widetilde{\mathbf{g}}_t, \mathbf{w}_t - \mathbf{w}_{t,i^\star}\rangle^2\right]} + 2\sqrt{10}dGRC_2$$

$$\leq C_1 \sqrt{160d^2G^2R^2 + 8d^3R^2V_T} + 2\sqrt{10}dGRC_2 \leq \mathcal{O}(\sqrt{d^3V_T}), \quad (\text{E.15})$$

where the second step is by Lemma 17.

**Base Regret Analysis.** In this part, we first provide different decompositions of the non-consecutive gradient variation defined on surrogates for strongly convex and convex functions, respectively, and then analyze the base regret in the corresponding cases.

For $\lambda$-*strongly convex* functions, we apply Lemma 19 to surrogate loss function sequence $\{h_{t,i^\star}^{\text{sc}}\}_{t=1}^T$. By choosing the optimism $M_t$ as $\widetilde{\mathbf{g}}_t + \frac{\lambda_{i^\star}}{2}(\mathbf{w}_{t-1,i^\star} - \mathbf{w}_{t-1})$, we bound the non-consecutive gradient variation on surrogates, i.e., $\bar{V}_{T,i^\star}^{\text{sc}} \triangleq \sum_{t=2}^T \|\nabla h_{t,i^\star}^{\text{sc}}(\mathbf{w}_{t,i^\star}) - M_t\|^2$, by

$$\mathbb{E}[\bar{V}_{T,i^\star}^{\text{sc}}] = \mathbb{E}\left[\sum_{t=2}^T \left\|\left(\mathbf{g}_t + \frac{\lambda_{i^\star}}{2}(\mathbf{w}_{t,i^\star} - \mathbf{w}_t)\right) - \left(\widetilde{\mathbf{g}}_t + \frac{\lambda_{i^\star}}{2}(\mathbf{w}_{t-1,i^\star} - \mathbf{w}_{t-1})\right)\right\|^2\right]$$

$$\leq \mathbb{E}\left[3\sum_{t=2}^T \|\mathbf{g}_t - \widetilde{\mathbf{g}}_t\|^2 + 3\sum_{t=2}^T \left\|\frac{\lambda_{i^\star}}{2}(\mathbf{w}_{t,i^\star} - \mathbf{w}_t)\right\|^2 + 3\sum_{t=2}^T \left\|\frac{\lambda_{i^\star}}{2}(\mathbf{w}_{t-1,i^\star} - \mathbf{w}_{t-1})\right\|^2\right]$$

$$\leq 36d^3V_T \log d + 144d^3L\log(dT)\mathbb{E}\left[\sum_{t=1}^T \mathcal{D}_{f_t}(\mathbf{w}^\star, \mathbf{w}_t)\right] + 2\lambda_{i^\star}^2\mathbb{E}\left[\sum_{t=2}^T \|\mathbf{w}_{t,i^\star} - \mathbf{w}_t\|^2\right],$$

where the first step uses the definition of $\nabla h_{t,i}^{\text{sc}}(\mathbf{x}) = \mathbf{g}_t + \frac{\lambda_i}{2}(\mathbf{x} - \mathbf{w}_t)$ and the last step is due to Lemma 5.

For *convex* and *linear* functions, the non-consecutive gradient variation $\bar{V}_{T,i^\star}^{\text{cvx}} \triangleq \sum_{t=2}^T \|\mathbf{g}_t - \widetilde{\mathbf{g}}_t\|^2$ and $\bar{V}_{T,i^\star}^{\text{lin}} \triangleq \sum_{t=2}^T \|\mathbf{g}_t - \widetilde{\mathbf{g}}_t\|^2$ can be bounded via Lemma 5 and Lemma 17, respectively.

To conclude, for different curvature types, we provide correspondingly different analyses of the non-consecutive gradient variation on surrogates:

$$\mathbb{E}[\bar{V}_{T,i^\star}^{\{\text{sc,cvx,lin}\}}] \leq \begin{cases} 36d^3V_T\log d + 144d^3L\log(dT)\mathbb{E}\left[\sum_{t=1}^T \mathcal{D}_{f_t}(\mathbf{w}^\star, \mathbf{w}_t)\right] + 2\lambda_{i^\star}^2\mathbb{E}\left[\sum_{t=2}^T \|\mathbf{w}_{t,i^\star} - \mathbf{w}_t\|^2\right], & (\lambda\text{-strongly convex}), \\[3mm] 12d^3V_T\log d + 48d^3L\log(dT)\mathbb{E}\left[\sum_{t=1}^T \mathcal{D}_{f_t}(\mathbf{w}^\star, \mathbf{w}_t)\right], & (\text{convex}), \\[3mm] 2d^3V_T, & (\text{linear}). \end{cases}$$

In the following, we analyze the base regret for different curvature types.

For $\lambda$-*strongly convex* functions, we have for any $t \geq 1$,

$$\mathbb{E}\left[\|\nabla h_t^{\text{sc}}(\mathbf{w}_{t,i^\star}) - M_t\|^2\right] = \mathbb{E}\left[\left\|\left(\mathbf{g}_t + \frac{\lambda_{i^\star}}{2}(\mathbf{w}_{t,i^\star} - \mathbf{w}_t)\right) - \left(\widetilde{\mathbf{g}}_t + \frac{\lambda_{i^\star}}{2}(\mathbf{w}_{t-1,i^\star} - \mathbf{w}_{t-1})\right)\right\|^2\right]$$

$$\leq 3\mathbb{E}\left[\|\mathbf{g}_t - \widetilde{\mathbf{g}}_t\|^2\right] + 3\mathbb{E}\left[\left\|\frac{\lambda_{i^\star}}{2}(\mathbf{w}_{t,i^\star} - \mathbf{w}_t)\right\|^2\right] + 3\mathbb{E}\left[\left\|\frac{\lambda_{i^\star}}{2}(\mathbf{w}_{t-1,i^\star} - \mathbf{w}_{t-1})\right\|^2\right]$$

$$\leq 24dG^2 + 8R^2\lambda_{i^\star}^2 + \mathcal{O}\left(\frac{1}{T}\right),$$

where the last step is by Lemma 6 and the boundedness of $\mathcal{X}$. We denote $C_5 = 24dG^2 + 8R^2\lambda_{i^\star}^2 + \mathcal{O}\left(\frac{1}{T}\right)$ for simplicity. Thus, by Lemma 19, the $i^\star$-th base learner guarantees:

$$
\text{BASE-REG} \leq 2\mathbb{E}\left[\sum_{t=1}^{T}\eta_t\|\nabla h_t^{\text{sc}}(\mathbf{w}_{t,i^\star}) - M_t\|^2\right] + \mathcal{O}(1) \leq \frac{C_5}{\lambda_{i^\star}}\log\left(1 + \lambda_{i^\star}\mathbb{E}\left[\bar{V}_{T,i^\star}^{\text{sc}}\right]\right)
$$

$$
\leq \frac{C_5}{\lambda_{i^\star}}\log\left(1 + 36d^3\lambda_{i^\star}V_T\log d + 144d^3L\lambda_{i^\star}\log(dT)\mathbb{E}\left[\sum_{t=1}^{T}\mathcal{D}_{f_t}(\mathbf{w}^\star,\mathbf{w}_t)\right] + 2\lambda_{i^\star}^3\mathbb{E}\left[\sum_{t=2}^{T}\|\mathbf{w}_{t,i^\star} - \mathbf{w}_t\|^2\right]\right)
$$

$$
\leq \frac{C_5}{\lambda_{i^\star}}\log\left(C_6\left(1 + 36d^3\lambda_{i^\star}V_T\log d + 144d^3L\lambda_{i^\star}\log(dT)\mathbb{E}\left[\sum_{t=1}^{T}\mathcal{D}_{f_t}(\mathbf{w}^\star,\mathbf{w}_t)\right]\right)\right) + \frac{\lambda_{i^\star}}{8}\mathbb{E}\left[\sum_{t=2}^{T}\|\mathbf{w}_{t,i^\star} - \mathbf{w}_t\|^2\right]
$$

$$
\leq \mathcal{O}\left(\frac{d}{\lambda_{i^\star}}\log\left(V_T + \log(dT)\right)\right) + \frac{1}{2}\mathbb{E}\left[\sum_{t=1}^{T}\mathcal{D}_{f_t}(\mathbf{w}^\star,\mathbf{w}_t)\right] + \frac{\lambda_{i^\star}}{8}\mathbb{E}\left[\sum_{t=1}^{T}\|\mathbf{w}_{t,i^\star} - \mathbf{w}_t\|^2\right], \tag{E.16}
$$

where the first inequality is by Lemma 23, the fourth step is by $C_6 = 16C_5$, $\ln(1+x) \leq x$ for $x \geq 0$, and $\lambda_{i^\star} \leq \lambda \leq 1$. The last step is by the following analysis. If $\mathbb{E}\left[\sum_{t=1}^{T}\mathcal{D}_{f_t}(\mathbf{w}^\star,\mathbf{w}_t)\right] \leq 1$, then we obtain

$$
\log\left(C_6\left(1 + 36d^3\lambda_{i^\star}V_T\log d + 144d^3L\lambda_{i^\star}\log(dT)\mathbb{E}\left[\sum_{t=1}^{T}\mathcal{D}_{f_t}(\mathbf{w}^\star,\mathbf{w}_t)\right]\right)\right) \leq \mathcal{O}(\log\left(dV_T + d\log(dT)\right)).
$$

Otherwise, we have

$$
\log\left(C_6\left(1 + 36d^3\lambda_{i^\star}V_T\log d + 144d^3L\lambda_{i^\star}\log(dT)\mathbb{E}\left[\sum_{t=1}^{T}\mathcal{D}_{f_t}(\mathbf{w}^\star,\mathbf{w}_t)\right]\right)\right)
$$

$$
\leq \log C_6 + \log\left(1 + \left(36d^3V_T\log d + 144d^3L\log(dT)\lambda_{i^\star}\right)\mathbb{E}\left[\sum_{t=1}^{T}\mathcal{D}_{f_t}(\mathbf{w}^\star,\mathbf{w}_t)\right]\right)
$$

$$
\leq \mathcal{O}(\log\left(dV_T + d\log(dT)\right)) + \frac{1}{2}\mathbb{E}\left[\sum_{t=1}^{T}\mathcal{D}_{f_t}(\mathbf{w}^\star,\mathbf{w}_t)\right],
$$

where the first inequality is due to $\mathbb{E}\left[\sum_{t=1}^{T}\mathcal{D}_{f_t}(\mathbf{w}^\star,\mathbf{w}_t)\right] > 1$ and the last inequality is by $\ln(1+x) \leq x$ for $x \geq 0$.

For *convex* functions, we apply Lemma 18 to surrogate loss function sequence $\{h_{t,i^\star}^{\text{cvx}}\}_{t=1}^{T}$. Choosing $M_t = \tilde{\mathbf{g}}_t$ and $\kappa = d$,

$$
\text{BASE-REG} \leq 10R\sqrt{d^2 + \mathbb{E}[\bar{V}_{T,i^\star}^{\text{cvx}}]} + \mathcal{O}(dR) + \frac{1}{d}\mathcal{O}\left(\max_{t\in[T]}\|\mathbf{g}_t - \tilde{\mathbf{g}}_t\|^2\right) \leq 10R\sqrt{d^2 + \mathbb{E}[\bar{V}_{T,i^\star}^{\text{cvx}}]} + \mathcal{O}(d)
$$

$$
\leq \mathcal{O}(\sqrt{d^2 + d^3V_T\log d}) + 10R\sqrt{48d^3L\log(dT)\mathbb{E}\left[\sum_{t=1}^{T}\mathcal{D}_{f_t}(\mathbf{w}^\star,\mathbf{w}_t)\right]} + \mathcal{O}(d) \qquad \text{(by Lemma 5)}
$$

$$
\leq \mathcal{O}(\sqrt{d^3V_T\log d}) + \mathcal{O}(d^3\log(dT)) + \frac{10RL}{2C_7}\mathbb{E}\left[\sum_{t=1}^{T}\mathcal{D}_{f_t}(\mathbf{w}^\star,\mathbf{w}_t)\right], \tag{E.17}
$$

where the first step is by Jensen's inequality, the second step is by Lemma 6, and the last step uses AM-GM inequality. $C_7$ is a $d$-independent constant to be specified.

For *linear* functions, we apply Lemma 18 to surrogate loss function sequence $\{h_{t,i^\star}^{\text{lin}}\}_{t=1}^{T}$. Choosing $M_t = \tilde{\mathbf{g}}_t$ and $\kappa = d$,

$$
\text{BASE-REG} \leq 10R\sqrt{d^2 + \mathbb{E}[\bar{V}_{T,i^\star}^{\text{lin}}]} + \mathcal{O}(dR) + \frac{1}{d}\mathcal{O}\left(\max_{t\in[T]}\|\mathbf{g}_t - \tilde{\mathbf{g}}_t\|^2\right)
$$

$$
\leq 10R\sqrt{d^2 + \mathbb{E}[\bar{V}_{T,i^\star}^{\text{lin}}]} + \mathcal{O}(d) \leq \mathcal{O}\left(\sqrt{d^3V_T}\right), \tag{E.18}
$$

where the first step is by Jensen's inequality.

**Overall Regret Analysis.** For $\lambda$-*strongly convex* functions, plugging Eq. (E.13) and Eq. (E.16) into Eq. (E.10) and choosing $C_3 = 160C_1G^2/\lambda$, we obtain

$$\mathbb{E}[\text{REG}_T] \leq \mathcal{O}\left(\frac{d}{\lambda}\log\left(V_T + \log(dT)\right)\right) + \mathcal{O}(d) + \left(\frac{1}{2} - \frac{1}{2}\right)\mathbb{E}\left[\sum_{t=1}^{T}\mathcal{D}_{f_t}(\mathbf{w}^\star, \mathbf{w}_t)\right]$$

$$+ \left(\frac{10C_1G^2}{C_3} + \frac{\lambda_{i^\star}}{8} - \frac{\lambda_{i^\star}}{4}\right)\mathbb{E}\left[\sum_{t=1}^{T}\|\mathbf{w}_t - \mathbf{w}_{t,i^\star}\|^2\right] \leq \mathcal{O}\left(\frac{d}{\lambda}\log\left(V_T + \log(dT)\right)\right).$$

For *convex* functions, plugging Eq. (E.14) and Eq. (E.17) into Eq. (E.11), we obtain

$$\mathbb{E}[\text{REG}_T] \leq \mathcal{O}(\sqrt{d^3 V_T \log d}) + \mathcal{O}(d^3\log(dT))$$

$$+ \left(\frac{C_1}{2C_4} + \frac{10RL}{2C_7} - 1\right)\mathbb{E}\left[\sum_{t=1}^{T}\mathcal{D}_{f_t}(\mathbf{w}^\star, \mathbf{w}_t)\right] \leq \mathcal{O}\left(\sqrt{d^3 V_T \log d} + d^3\log(dT)\right),$$

by choosing $C_4 = C_1$ and $C_7 = 10RL$.

For *linear* functions, plugging Eq. (E.15) and Eq. (E.18) into Eq. (E.12), we obtain

$$\mathbb{E}[\text{REG}_T] \leq \mathcal{O}(\sqrt{d^3 V_T}).$$

Note that the constants $C_3, C_4, C_6, C_7$ only exist in analysis and thus can be chosen arbitrarily, which finishes the proof. ∎

Finally, we present the proofs of Lemmas Lemma 5 and Lemma 21 for completeness.

**Proof** [of Lemma 5] By Lemma 6, we have

$$\bar{V}_T = \sum_{t=1}^{T}\|\mathbf{g}_t - \widetilde{\mathbf{g}}_t\|^2 \leq d^2\sum_{t=1}^{T}\sum_{i=1}^{d}\rho_{t,i}(\nabla_i f_t(\mathbf{w}_t) - \nabla_i f_{t-1}(\mathbf{w}_{t-1}))^2 + \mathcal{O}(1)$$

$$\leq 3d^2\sum_{t=1}^{T}\sum_{i=1}^{d}\rho_{t,i}(\nabla_i f_t(\mathbf{w}_t) - \nabla_i f_t(\mathbf{w}^\star))^2 + 3d^2\sum_{t=1}^{T}\sum_{i=1}^{d}\rho_{t,i}(\nabla_i f_t(\mathbf{w}^\star) - \nabla_i f_{t-1}(\mathbf{w}^\star))^2$$

$$+ 3d^2\sum_{t=1}^{T}\sum_{i=1}^{d}\rho_{t,i}(\nabla_i f_{t-1}(\mathbf{w}^\star) - \nabla_i f_{t-1}(\mathbf{w}_{t-1}))^2 + \mathcal{O}(1), \tag{E.19}$$

where $\rho_{t,i}$ is defined in Eq. (B.1) and the second step is due to Cauchy-Schwarz inequality. Taking expectation over the randomness of $\{\mathbf{e}_{i_t}\}_{t=1}^{T}$, we bound the second term in Eq. (E.19) by

$$\mathbb{E}\left[\sum_{t=1}^{T}\sum_{i=1}^{d}\rho_{t,i}(\nabla_i f_t(\mathbf{w}^\star) - \nabla_i f_{t-1}(\mathbf{w}^\star))^2\right] \leq \mathbb{E}\left[\sum_{t=1}^{T}\max_{i\in[d]}[\rho_{t,i}]\|\nabla f_t(\mathbf{w}^\star) - \nabla f_{t-1}(\mathbf{w}^\star)\|^2\right] \leq 4d\log d V_T, \tag{E.20}$$

where the second step is by Eq. (B.3).

The analyses for the first and third terms in Eq. (E.19) are similar to the counterpart in the proof of Lemma 2. Let $Q$ be the same bad event as in Eq. (C.5), with $\bar{\rho} = 4d\log(dT)$. We can upper-bound the expectation of the first term in Eq. (E.19) by

$$\mathbb{E}\left[\sum_{t=1}^{T}\sum_{i=1}^{d}\rho_{t,i}(\nabla_i f_t(\mathbf{w}_t) - \nabla_i f_t(\mathbf{w}^\star))^2\right]$$

$$\leq \mathbb{E}\left[\sum_{t=1}^{T}\sum_{i=1}^{d}\rho_{t,i}(\nabla_i f_t(\mathbf{w}_t) - \nabla_i f_t(\mathbf{w}^\star))^2\mathbb{I}(Q)\right] + \mathbb{E}\left[\sum_{t=1}^{T}\sum_{i=1}^{d}\rho_{t,i}(\nabla_i f_t(\mathbf{w}_t) - \nabla_i f_t(\mathbf{w}^\star))^2\mathbb{I}(\neg Q)\right]$$

$$\leq \mathcal{O}(1) + 4d\log(dT)\mathbb{E}\left[\sum_{t=1}^{T}\|\nabla f_t(\mathbf{w}_t) - \nabla f_t(\mathbf{w}^\star)\|^2\right] \leq \mathcal{O}(1) + 8dL\log(dT)\mathbb{E}\left[\sum_{t=1}^{T}\mathcal{D}_{f_t}(\mathbf{w}^\star, \mathbf{w}_t)\right], \tag{E.21}$$

where the third step uses $\rho_{t,i} \leq T + 1$ on $Q$, $\rho_{t,i} \leq \bar{\rho}$ on $\neg Q$, Eq. (C.5), and the boundedness of $\mathcal{X}$, and the last step is by $\|\nabla f_t(\mathbf{x}) - \nabla f_t(\mathbf{y})\|^2 \leq 2L\mathcal{D}_{f_t}(\mathbf{y}, \mathbf{x})$ for any $\mathbf{x}, \mathbf{y} \in \mathcal{X}^+$ (Yan et al., 2024). Similarly, we can upper-bound the expectation of the third term of Eq. (E.19) by

$$\mathbb{E}\left[\sum_{t=1}^{T}\sum_{i=1}^{d}\rho_{t,i}(\nabla_i f_{t-1}(\mathbf{w}_{t-1}) - \nabla_i f_{t-1}(\mathbf{w}^\star))^2\right] \leq \mathcal{O}(1) + 8dL\log(dT)\mathbb{E}\left[\sum_{t=1}^{T}\mathcal{D}_{f_t}(\mathbf{w}^\star, \mathbf{w}_t)\right]. \tag{E.22}$$

Plugging Eq. (E.20), Eq. (E.21), and Eq. (E.22) back into Eq. (E.19), we finish the proof. ∎

**Proof** [of Lemma 21] Let $\mathbb{E}_t[\cdot]$ denote the conditional expectation given the history before sampling $i_t$ at round $t$. Since $\mathbf{w}_t$, $\mathbf{w}_{t,i^\star}$, and $\widetilde{\mathbf{g}}_t$ are determined before sampling $i_t$, they are fixed under $\mathbb{E}_t[\cdot]$.

For each $j \in [d]$, define the directional difference

$$q_{t,j} \triangleq \frac{1}{2\delta}\left(f_t(\mathbf{w}_t + \delta\mathbf{e}_j) - f_t(\mathbf{w}_t - \delta\mathbf{e}_j)\right).$$

Thus, the observed quantity in Eq. (2.2) satisfies $v_t = q_{t,i_t}$. By $G$-Lipschitzness, $|q_{t,j}| \leq G$ for every $j \in [d]$. Moreover, since each coordinate of $\widetilde{\mathbf{g}}_t$ stores a previously observed directional difference, property *(ii)* of Lemma 6 implies $|\widetilde{g}_{t,j}| \leq G$ for every $j \in [d]$.

By Eq. (2.2), we have

$$\mathbf{g}_t = \widetilde{\mathbf{g}}_t + d\left(q_{t,i_t} - \widetilde{g}_{t,i_t}\right)\mathbf{e}_{i_t}.$$

Therefore,

$$\langle\mathbf{g}_t, \mathbf{w}_t - \mathbf{w}_{t,i^\star}\rangle = \langle\widetilde{\mathbf{g}}_t, \mathbf{w}_t - \mathbf{w}_{t,i^\star}\rangle + d\left(q_{t,i_t} - \widetilde{g}_{t,i_t}\right)\langle\mathbf{e}_{i_t}, \mathbf{w}_t - \mathbf{w}_{t,i^\star}\rangle.$$

Using $(a + b)^2 \leq 2a^2 + 2b^2$, we obtain

$$\langle\mathbf{g}_t, \mathbf{w}_t - \mathbf{w}_{t,i^\star}\rangle^2 \leq 2\langle\widetilde{\mathbf{g}}_t, \mathbf{w}_t - \mathbf{w}_{t,i^\star}\rangle^2 + 2d^2\left(q_{t,i_t} - \widetilde{g}_{t,i_t}\right)^2\langle\mathbf{e}_{i_t}, \mathbf{w}_t - \mathbf{w}_{t,i^\star}\rangle^2.$$

By property *(v)* of Lemma 6, the first term satisfies

$$2\langle\widetilde{\mathbf{g}}_t, \mathbf{w}_t - \mathbf{w}_{t,i^\star}\rangle^2 \leq 2\|\widetilde{\mathbf{g}}_t\|^2\|\mathbf{w}_t - \mathbf{w}_{t,i^\star}\|^2 \leq 2dG^2\|\mathbf{w}_t - \mathbf{w}_{t,i^\star}\|^2.$$

For the second term, since $i_t$ is uniformly sampled from $[d]$ conditional on the history before round $t$, we have

$$\mathbb{E}_t\left[2d^2\left(q_{t,i_t} - \widetilde{g}_{t,i_t}\right)^2\langle\mathbf{e}_{i_t}, \mathbf{w}_t - \mathbf{w}_{t,i^\star}\rangle^2\right] = 2d\sum_{j=1}^{d}\left(q_{t,j} - \widetilde{g}_{t,j}\right)^2\langle\mathbf{e}_j, \mathbf{w}_t - \mathbf{w}_{t,i^\star}\rangle^2$$

$$\leq 8dG^2\sum_{j=1}^{d}\langle\mathbf{e}_j, \mathbf{w}_t - \mathbf{w}_{t,i^\star}\rangle^2 = 8dG^2\|\mathbf{w}_t - \mathbf{w}_{t,i^\star}\|^2.$$

Combining the above bounds yields

$$\mathbb{E}_t\left[\langle\mathbf{g}_t, \mathbf{w}_t - \mathbf{w}_{t,i^\star}\rangle^2\right] \leq 10dG^2\|\mathbf{w}_t - \mathbf{w}_{t,i^\star}\|^2.$$

Taking expectation over the history completes the proof. ∎

**Worst-Case Universal Regret.** As a byproduct, we show that the online ensemble framework also achieves worst-case universal regret guarantees using a single gradient estimator shared across all base learners, with a simpler analysis.

Following Yan et al. (2023, Proposition 1), we use Adapt-ML-Prod (Gaillard et al., 2014) as the meta-algorithm and Bandit Gradient Descent (Agarwal et al., 2010) as the base algorithm. As in the gradient-variation universal setting, we instantiate a collection of strongly convex base learners with candidate curvature parameters from the predefined pool (E.6), together with

one additional base learner for the linear and convex cases. Therefore, the ensemble contains $N \triangleq 1 + |\mathcal{H}^{\text{sc}}| = \mathcal{O}(\log T)$ base learners in total.

At each round $t$, we construct the two-point gradient estimator

$$\mathbf{g}_t = \frac{d}{2\delta}\big(f_t(\mathbf{w}_t + \delta \mathbf{u}_t) - f_t(\mathbf{w}_t - \delta \mathbf{u}_t)\big)\mathbf{u}_t,$$

and feed Adapt-ML-Prod the normalized linearized loss vector $\boldsymbol{\ell}_t = (\ell_{t,1}, \dots, \ell_{t,N})$, where $\ell_{t,i} \triangleq \frac{1}{2dGR}\langle \mathbf{g}_t, \mathbf{w}_{t,i}\rangle + \frac{1}{2} \in [0,1]$ and $\mathbf{w}_{t,i}$ denotes the decision of the $i$-th base learner. Here, the range condition follows from $\|\mathbf{g}_t\| \leq dG$ and $\|\mathbf{w}_{t,i}\| \leq R$. For each candidate curvature parameter $\lambda_i \in \mathcal{H}^{\text{sc}}$, the corresponding strongly convex base learner runs Algorithm 3 with the associated configuration. The additional base learner handles the linear and convex cases using the configuration of Theorem 1 of Shamir (2017).

**Theorem 11.** *Under Assumptions 1-2, the above ensemble algorithm achieves the following universal regret:*

$$\mathbb{E}[\text{REG}_T] \leq \begin{cases} \mathcal{O}\left(\frac{d}{\lambda}\log T\right), & (\lambda\text{-strongly convex case}), \\ \mathcal{O}\left(\sqrt{dT} + d\right), & (\text{convex/linear case}) \end{cases}$$

**Proof** For $\lambda$-*strongly convex* functions, let $i^\star$ index a base learner whose candidate parameter satisfies $\lambda_{i^\star} \leq \lambda \leq 2\lambda_{i^\star}$. We have the decomposition

$$\mathbb{E}[\text{REG}_T] \leq \underbrace{\mathbb{E}\left[\sum_{t=1}^{T}\langle \mathbf{g}_t, \mathbf{w}_t - \mathbf{w}_{t,i^\star}\rangle\right]}_{\text{META-REG}} + \underbrace{\mathbb{E}\left[\sum_{t=1}^{T}h_{t,i^\star}^{\text{sc}}(\mathbf{w}_{t,i^\star}) - \sum_{t=1}^{T}h_{t,i^\star}^{\text{sc}}(\mathbf{w}^\star)\right]}_{\text{BASE-REG}} - \frac{\lambda_{i^\star}}{2}\mathbb{E}\left[\sum_{t=1}^{T}\|\mathbf{w}_t - \mathbf{w}_{t,i^\star}\|^2\right] + \mathcal{O}(1),$$

where $h_{t,i}^{\text{sc}}(\mathbf{x}) \triangleq \langle \mathbf{g}_t, \mathbf{x}\rangle + \frac{\lambda_i}{2}\|\mathbf{x} - \mathbf{w}_t\|^2$ is the surrogate loss corresponding to $\lambda_i \in \mathcal{H}^{\text{sc}}$ in (E.6).

For the meta regret, applying Corollary 4 of Gaillard et al. (2014) to the normalized losses above and rescaling back to the original linearized losses, there exist constants $C_8$ and $C_9$ such that

$$\text{META-REG} \leq \mathcal{O}(d) + C_8\mathbb{E}\left[\sqrt{\ln\ln T \cdot \sum_{t=1}^{T}\langle \mathbf{g}_t, \mathbf{w}_t - \mathbf{w}_{t,i^\star}\rangle^2}\right]$$

$$\leq \mathcal{O}(d) + C_8\sqrt{\ln\ln T \cdot \mathbb{E}\left[\sum_{t=1}^{T}\|\mathbf{g}_t\|^2\|\mathbf{w}_t - \mathbf{w}_{t,i^\star}\|^2\right]} \leq \mathcal{O}(d) + C_8\sqrt{C_9 dG^2 \ln\ln T \cdot \mathbb{E}\left[\sum_{t=1}^{T}\|\mathbf{w}_t - \mathbf{w}_{t,i^\star}\|^2\right]}$$

$$\leq \mathcal{O}(d) + \frac{C_8^2 C_9 dG^2 \ln\ln T}{2\lambda_{i^\star}} + \frac{\lambda_{i^\star}}{2}\mathbb{E}\left[\sum_{t=1}^{T}\|\mathbf{w}_t - \mathbf{w}_{t,i^\star}\|^2\right] \leq \mathcal{O}(d) + \frac{C_8^2 C_9 dG^2 \ln\ln T}{\lambda} + \frac{\lambda_{i^\star}}{2}\mathbb{E}\left[\sum_{t=1}^{T}\|\mathbf{w}_t - \mathbf{w}_{t,i^\star}\|^2\right].$$

Here, the second inequality follows from Jensen's inequality and Cauchy-Schwarz, the third inequality follows from Lemma 5 of Shamir (2017), the fourth inequality follows from the AM-GM inequality, and the last inequality uses $\lambda_{i^\star} \leq \lambda \leq 2\lambda_{i^\star}$. For the base regret, Theorem 8 implies BASE-REG $\leq \mathcal{O}(\frac{d}{\lambda_{i^\star}}\log T) = \mathcal{O}(\frac{d}{\lambda}\log T)$. Combining the meta and base regret bounds with the negative quadratic term in the decomposition gives $\mathbb{E}[\text{REG}_T] \leq \mathcal{O}(\frac{d}{\lambda}\log T)$, where the $\log\log T$ factor is omitted according to our convention.

For *convex* and *linear* functions, we have the decomposition

$$\mathbb{E}[\text{REG}_T] \leq \underbrace{\mathbb{E}\left[\sum_{t=1}^{T}\langle \mathbf{g}_t, \mathbf{w}_t - \mathbf{w}_{t,i^\star}\rangle\right]}_{\text{META-REG}} + \underbrace{\mathbb{E}\left[\sum_{t=1}^{T}\langle \mathbf{g}_t, \mathbf{w}_{t,i^\star} - \mathbf{w}^\star\rangle\right]}_{\text{BASE-REG}} + \mathcal{O}(1),$$

For the meta regret, applying Corollary 4 of Gaillard et al. (2014) to the normalized losses and rescaling, there exists a constant $C_8$ such that

$$\text{META-REG} \leq \mathcal{O}(d) + C_8\mathbb{E}\left[\sqrt{\ln\ln T \cdot \sum_{t=1}^{T}\langle \mathbf{g}_t, \mathbf{w}_t - \mathbf{w}_{t,i^\star}\rangle^2}\right]$$

$$\leq \mathcal{O}(d) + 2RC_8 \sqrt{\ln \ln T \cdot \sum_{t=1}^{T} \mathbb{E}\left[\|\mathbf{g}_t\|^2\right]} \leq \mathcal{O}(\sqrt{dT} + d).$$

Here, the second inequality follows from Jensen's inequality and $\|\mathbf{w}_t - \mathbf{w}_{t,i^\star}\| \leq 2R$, while the last inequality follows from Lemma 5 of Shamir (2017). By Theorem 1 of Shamir (2017), the base regret satisfies $\textsc{Base-Reg} \leq \mathcal{O}(\sqrt{dT})$. Hence, combining the meta and base regret bounds yields $\mathbb{E}[\textsc{Reg}_T] \leq \mathcal{O}(\sqrt{dT} + d)$, where the $\log \log T$ factor is omitted according to our convention.

∎

# F. Technical Lemmas

**Lemma 22** (Lemma 4.8 of Pogodin & Lattimore (2019)). *Let $a_1, a_2, \ldots, a_T$ be non-negative real numbers. Then*

$$\sum_{t=1}^{T} \frac{a_t}{\sqrt{1 + \sum_{s=1}^{t-1} a_s}} \leq 4\sqrt{1 + \sum_{t=1}^{T} a_t} + \max_{t \in [T]} a_t.$$

**Lemma 23** (Lemma 9 of Yan et al. (2023)). *For a sequence of $\{a_t\}_{t=1}^{T}$ and $b$, where $a_t, b > 0$ for any $t \in [T]$, denoting by $a_{\max} \triangleq \max_t a_t$ and $A \triangleq \lceil b \sum_{t=1}^{T} a_t \rceil$, we have*

$$\sum_{t=1}^{T} \frac{a_t}{bt} \leq \frac{a_{\max}}{b}(1 + \log A) + \frac{1}{b^2}.$$

**Lemma 24** (Lemma 9 of Zhao et al. (2024)). *For any $x, y, a, b > 0$ satisfying $x - y \leq \sqrt{ax} + b$, it holds that*

$$x - y \leq \sqrt{ay + ab} + a + b.$$

**Lemma 25** (Lemma 16 of Orabona et al. (2012)). *If $a, b, c, x, y > 0$ satisfy $x - y \leq a \log(bx + c)$, then it holds that*

$$x - y \leq a \log\left(2ab \log \frac{2ab}{e} + 2by + 2c\right).$$

