# OpenReview forum: "Improved Dimension Dependence for Bandit Convex Optimization with Gradient Variation"
_ICML.cc/2026/Conference — ICML 2026 spotlight_

### Official Review · Reviewer_HCsa · 2026-03-02

**Soundness:** 3
**Presentation:** 2
**Significance:** 3
**Originality:** 3
**Overall Recommendation:** 4
**Confidence:** 3

**Summary:**

The paper studies the gradient variation of Bandit Convex Optimization, mainly under the two-points feedback setup.
The main contributions include a new analysis for an existing algorithm, which yields better regret in the convex and strongly convex setup, as well as results related to gradient variance, small loss, one-point BLO, and applications to other fields.

**Compliance With Llm Reviewing Policy:**

Affirmed.

**Final Justification:**

I think the comparison with previous works should be more clear, but I am sure the authors can address it in the final version.
I maintain my score.

**Key Questions For Authors:**

I have a few questions, and the authors may find it useful to fix some typos.

Q1: Theorem 8, how can it be that the dependency on $m$ is different from $n$?

Q2: In theorem 5, can you choose $\eta$ that depends on $V_T$, while $V_T$ depends on $\eta$?

Typos:

The definition of gradient variance $W_T$ (Eq 3.3) is not clear. What is the argument of the supremum?

In Eq. 4.1, do you need to remove the comma?

Theorems 3,4 I think the word “regret” is missing there. “enjoys $O(\dots)$” regret.

**Limitations:**

yes

**Strengths And Weaknesses:**

My overall impression from the paper is positive, but the paper lacks clarity in certain parts.

On the positive side, the paper shows a clear improvement in the dimensionality dependency.
The paper addresses many problems, such as gradient variance, small loss and more.
The paper analyses an old algorithm, which in my taste is a positive thing. I prefer to exhaust the current algorithms with new analysis, and come up with new algorithms only when it is unavoidable, rather than inventing unnecessarily new algorithms.

Nevertheless, there are several issues I would like the authors to cosider.
First, it is hard to understand if the authors assume smoothness or not. As I understand it, they assumed it.
If smoothness is important, why not stating this in the main contributions paragraph?

From Chiang et al. 2013 and until now, there was no improvement in this field?

I do not understand the paragraph that starts at 293L. It is not clear from the text if Shamir 2017 proved it already for the more general case, or if you imply a new analysis to Shamir 2017.

To summarize my review, my main concern is comparison with previous works, with emphasis on smoothness. If the authors can address these issues I am willing to consider raising my score.

---

> ### Author Rebuttal · Authors · 2026-03-31
>
> We sincerely appreciate your thoughtful feedback. Below, we respond to the weaknesses you identified and provide detailed answers to your questions.
>
> ---
> **W1:** It is hard to understand whether the authors assume smoothness. As I understand it, they do. If smoothness is important, why not state this in the main contributions paragraph?
>
> **A1:** Thanks for the comment. The smoothness assumption is essential for establishing gradient-variation regret [1] and is stated as Assumption 3 in Section 2.1. In the revised manuscript, we have clarified its necessity in the "Main Contributions" section.
>
> [1] Online optimization with gradual variations, COLT 2012
>
> ---
> **W2:** From Chiang et al. 2013 and until now, there was no improvement in this field?
>
> **A2:** Thank you for this question. While the gradient-variation bound in two-point BCO has been studied since Chiang et al. (2013), our work provides **the first improvement in the dimension** dependence of the regret bound. Specifically, our key technical contribution lies in explicitly formalizing the "non-consecutivity" structure within the bandit setting and providing a fine-grained analysis of it.
>
> Furthermore, our contribution is not restricted to the two-point BCO setup; it is deeply rooted in the study of gradient estimators, which are fundamental to bandits. Both the estimators in Eq. (2.2) and Eq. (4.2) are variants of *variance-reduced* gradient estimators, a class of estimators widely studied in bandits [1, 2]. We believe our work may be useful across broader bandit-feedback settings.
>
> [1] More adaptive algorithms for adversarial bandits, COLT 2018
>
> [2] Taking a hint: How to leverage loss predictors in contextual bandits, COLT 2020
>
> ---
> **W3:** I do not understand the paragraph that starts at 293L. It is not clear from the text whether Shamir (2017) already proved it for the more general case, or whether you imply a new analysis beyond Shamir (2017).
>
> **A3:** Thanks for this observation. While Shamir (2017) mentioned the potential for extending the techniques to the strongly convex case, the author did **not** prove the same regret guarantee as in our paper.
>
> Our contribution in Appendix C.4 is a non-trivial extension that formally adapts the concentration-based analysis to the strongly convex setting. To our knowledge, this is **the first time** this result has been established. We will revise the final version to more explicitly distinguish our formal derivation from the intuitive suggestion in Shamir (2017).
>
> ---
> **Q1:** Theorem 8, how can it be that the dependency on $m$ is different from $n$?
>
> **A4:** Thanks for the sharp observation! There was indeed a typo in the statement of Theorem 8 regarding the $m, n$ dependency for the $y$-player. Specifically, the bounds mentioned are individual worst-case regret guarantees, which hold for each player regardless of the opponent's strategy. The correct bounds for the $x$ and $y$ players are $\tilde{O}(\sqrt{m^7n^2T})$ and $\tilde{O}(\sqrt{n^7m^2T})$, respectively. We have corrected this typo in the revised manuscript. Thanks again for pointing this out.
>
> ---
> **Q2:** In Theorem 5, can you choose $\eta$ that depends on $V_T$, while $V_T$ depends on $\eta$?
>
> **A5:** Thanks for the question. We clarify that there is no circular dependency between them. As defined in Eq. (3.3), $V_T \triangleq \sum_{t=2}^T \\|\boldsymbol{\ell}\_t-\boldsymbol{\ell}_{t-1}\\|^2$ is an environment-dependent quantity determined solely by the loss sequence $\{\boldsymbol{\ell}_t\}$. It is defined independently of the algorithm’s iterates $\mathbf{x}_t$ or the choice of step size $\eta$.
>
> ---
> **Q3:** The definition of gradient variance $W_T$ (Eq. (3.3)) is not clear. What is the argument of the supremum?
>
> **A6:** Thanks for the observation. $W_T$ is a problem-dependent quantity that captures the worst-case cumulative variance of the gradients. In Eq. (3.3), the supremum is taken over **all possible** sequences $\{\mathbf{x}\_1, \dots, \mathbf{x}\_T\}$ within the feasible set $\mathcal{X}$. This sequence is different from the actual algorithmic trajectory. In the revised manuscript, we have updated the definition using $\xi_t$ to clearly distinguish it from the algorithm's decisions:$$W_T \triangleq \sup_{\{\xi_1, \dots, \xi_T\} \in \mathcal{X}} \left\\{ \sum_{t=1}^T \\|\nabla f_t(\xi_t) - \mu_T\\|^2 \right\\},$$ where $\mu_T \triangleq \frac{1}{T} \sum_{t=1}^T \nabla f_t(\xi_t)$ is the mean gradient.
>
> ---
> **Q4:** In Eq. 4.1, do you need to remove the comma?
>
> **A7:** Thanks for the careful reading. This typo has been corrected in the updated version.
>
> ---
> **Q5:** In Theorems 3 and 4, I think the word "regret" is missing there: "enjoys" $O(\cdot)$ regret.
>
> **A8:** Thanks for pointing it out. We have corrected this typo in the revised version.
>
> ---
> We hope the above replies address your concerns and would appreciate a reevaluation of our paper's score. We are happy to provide further details if needed.

---

> > ### Author Rebuttal · Reviewer_HCsa · 2026-04-03
> >
> > Thank you for the response.
> > The paper presents a clear contribution, but I still lack a comparison with other works regarding smoothness. Did all previous works with similar forms of bounds assume smoothness?

---

> > > ### Author Response · Authors · 2026-04-03
> > >
> > > **[Updated ! ! !]** Thank you for your follow-up comments. We appreciate the opportunity to further clarify the role of the smoothness assumption.
> > >
> > > In the online learning literature, the smoothness assumption is standard for achieving various problem-dependent bounds, such as gradient variation, small-loss, and gradient variance. This requirement holds across diverse feedback settings:
> > >
> > > **Full-Information Setting:**
> > >
> > > - **Gradient variation:** [1] proposed the first gradient-variation bound under smoothness.
> > >
> > > - **Small loss:** [2] first studied small-loss bounds in online optimization under smoothness. Following this, [3] investigated small-loss bounds for other types of functions, also under the smoothness assumption.
> > >
> > > - **Gradient variance:** [4] studied these bounds in Prediction from Expert Advice (PEA) and online linear optimization, both of which are settings where smoothness is satisfied.
> > >
> > > **Bandit Settings:** For both one-point bandit [5] and two-point bandit [6] feedback, smoothness remains a key technical assumption for deriving gradient-variance or gradient-variation bounds.
> > >
> > > In summary, the smoothness assumption is a fundamental requirement for achieving **problem-dependent regret** guarantees. We hope these clarifications can address your concerns and we are happy to provide any further details you may require.
> > >
> > > [1] Online optimization with gradual variations, COLT 2012
> > >
> > > [2] Smoothness, low noise and fast rates, NIPS 2010
> > >
> > > [3] Beyond logarithmic bounds in online learning, AISTATS 2012
> > >
> > > [4] Extracting certainty from uncertainty: Regret bounded by variation in costs, COLT 2008
> > >
> > > [5] Better algorithms for benign bandits, JMLR 2011
> > >
> > > [6] Beating bandits in gradually evolving worlds, COLT 2013
> > >
> > > **[New]:** As the rebuttal period is coming to an end, we are wondering if our responses have answered your question about the requirement of the smoothness assumption in previous works. If not, we are happy to answer your follow-up questions. And if our answer has resolved your question, we would appreciate if you could reevaluate the score of our paper. Thanks again for your support!

---

### Official Review · Reviewer_tEEe · 2026-03-12

**Soundness:** 3
**Presentation:** 3
**Significance:** 3
**Originality:** 3
**Overall Recommendation:** 4
**Confidence:** 3

**Summary:**

This paper studies gradient-variation regret bounds in Bandit Convex Optimization (BCO) with two-point feedback. The main contribution is a refined analysis of the non-consecutive gradient variation, which arises because the learner samples only one direction per round. The authors improve dimension dependence for convex functions and strongly convex functions. They also derive the first gradient-variance and small-loss regret bounds for two-point BCO, extend to one-point bandit linear optimization over hyper-rectangular domains, and apply their techniques to dynamic regret, universal regret, and bandit games.

**Compliance With Llm Reviewing Policy:**

Affirmed.

**Final Justification:**

The rebuttal addresses my concerns and I will maintain my current evaluation.

**Key Questions For Authors:**

1. For the one-point BLO result, what is the fundamental obstacle to extending beyond hyper-rectangular domains? Have you ever considered other self-concordant barriers?
2. Are there any lower bound results?

**Limitations:**

Yes. The authors acknowledge the restriction to hyper-rectangular domains for one-point BLO and leave the general case to future work.

**Strengths And Weaknesses:**

Strengths:

**Soundness.** The technical approach is clean and the core insight is elegant.

**Significance.** The improvements over dimension dependence are meaningful.

**Presentation.** The paper is generally well-organized. Table 1 is clear.

**Originality.** The proof techniques are novel and can be extended to many settings.

---

Weakness:

**One-point BLO result is weak.** The $O(d^{7/2} \sqrt{V_{T}})$ bound has a very large dimension dependence.

**Missing discussion of lower bounds.** This paper does not discuss whether the achieved dimension dependences are tight.

**No experiments.** The paper is purely theoretical, and simple synthetic experiments would strengthen the contribution.

---

> ### Author Rebuttal · Authors · 2026-03-31
>
> We thank the reviewer for the constructive feedback. Below, we respond to the weaknesses and answer your questions.
>
> ---
> **W1: One-point BLO result has a large dimension dependence.**
>
> **A1:** Thanks for the observation. The $d^{7/2}$ dependence is worse than in the two-point setting, reflecting the difficulty of adaptivity in one-point BLO. It comes from three sources:
> - **Adaptivity**: gradient-variation adaptivity contributes $d^2$.
> - **Self-Concordance**: instead of the usual $d^{1/2}$, our proof incurs $d$ because, to obtain the $V_T$ bound, $\eta$ must ensure the third term on the RHS of Eq. (D.10) is non-positive, preventing the standard balancing argument.
> - **Diameter**: $R=O(\sqrt{d})$ contributes $d^{1/2}$.
>
> While limited in dimension dependence and domain generality, this is still **the first gradient-variation bound for one-point BLO**. Improving the $d$-dependence and extending beyond these domains are important future directions.
>
> ---
> **W2: Missing discussion of lower bounds.**
>
> **A2:** Thanks for the comment. We justify the tightness of our results below:
> - Convex Setting: The standard lower bound for convex BCO is $\Omega(\sqrt{dT})$ [1]. Since gradient variation $V_T$, small loss $F_T$, and gradient variance $W_T$ can all be $O(T)$ in the worst case, the corresponding lower bounds are $\Omega(\sqrt{d V_T})$, $\Omega(\sqrt{d F_T})$, and $\Omega(\sqrt{d W_T})$.
>   - $W_T$: Our bound for linear functions is optimal up to an *additive* $O(d)$ term.
>   - $F_T$: Our result for convex functions is optimal up to an *additive* $O(d)$ term.
>   - $V_T$: We have narrowed the gap from the previous $O(d^2 \sqrt{V_T})$ to $\tilde{O}(d^{3/2} \sqrt{V_T})$. It remains open whether $\Omega(\sqrt{d V_T})$ is the tightest possible lower bound.
> - Strongly Convex Setting: Even the worst-case regret bound remains open.
>
> We will clarify tightness and these open problems in the final version. Thanks for the suggestion.
>
> [1] Optimal rates for zero-order convex optimization: The power of two function evaluations, TIT 2015
>
> ---
> **W3: No experiments.**
>
> **A3:** Thanks for the suggestion. We add three experiments below; full figures are available at [link](https://anonymous.4open.science/r/ICML2026_GV2pointBCO-E44E/).
>
> **Experiment 1 (convex):** We compare Algorithm 1 with Agarwal et al. (2010) and [1] on two convex tasks. The first is a quadratic loss with small variation $V_T=O(1)$:
> $$
> f_t(x)=\\|x-\mu_t\\|_{Q_t}^2,
> $$
> where $\mu_t$ varies slowly and $Q_t$ is PSD and rank-deficient.
>
> The second is online logistic regression with large variation $V_T=O(T)$:
> $$
> f_t(x)=\log(1+\exp(-y_t\langle a_t,x\rangle)),
> $$
> with $(a_t,y_t)$ from the **a9a** dataset.
>
> **Conclusion 1:** As shown in Figure 1, our method performs best on both tasks, showing the benefit of variation adaptivity.
>
> **Experiment 2 (strongly convex):** We compare Algorithm 1 with Chiang et al. (2013) and Agarwal et al. (2010) on two strongly convex tasks. Task 1 uses a loss with $V_T=O(1)$:
> $$
> f_t(x)=\frac{\lambda}{2}\\|x-\mu_t\\|^2+\frac{c}{4}\\|x-\mu_t\\|_4^4,
> $$
> where $\mu_t$ varies slowly.
>
> Task 2 is online logistic regression on **a9a** with $V_T=O(T)$:
> $$
> f_t(x)=\log(1+\exp(-y_t\langle a_t,x\rangle))+\frac{\lambda}{2}\\|x\\|^2.
> $$
>
> **Conclusion 2:** As shown in Figure 2, our method attains the smallest regret, benefiting from gradient-variation adaptivity and better dimension dependence.
>
> **Experiment 3 (dimension scaling):** We study how regret scales with dimension by comparing Algorithm 1 with Chiang et al. (2013) on three strongly convex losses. The first is
> $$
> f_t^{(1)}(x)=\frac{\lambda}{2}\\|x-\mu_t\\|^2+\frac{c}{4}\\|x-\mu_t\\|_4^4,
> $$
>
> The second, $f_t^{(2)}(x)$, is the same as the loss in Experiment 2.
>
> The third is
> $$
> f_t^{(3)}(x)=\frac{\lambda}{2}\\|x-\mu_t\\|^2+w(\exp(z)-z-1),
> $$
> where $z=s^\top(x-\mu_t)$, $\mu_t$ varies slowly and $w,s$ are constants.
>
> **Conclusion 3:** As shown in Figure 3, for all losses, the ratio increases with $d$, confirming better dimension dependence than Chiang et al. (2013).
>
> Overall, these experiments support three claims: stronger performance in the convex and strongly convex settings, and better dimension scaling consistent with theory.
>
> [1] A gradient estimator via L1-randomization for online zero-order optimization with two point feedback, NeurIPS 2022
>
> ---
> **Q1: Main obstacle to extending one-point BLO beyond hyper-rectangular domains.**
>
> **A4:** Thanks for the insightful question! For general self-concordant barriers, the main difficulty is that the eigenvectors of $\nabla^2 R(w)$ vary with $w$, making historical information hard to reuse. For hyper-rectangular domains, by contrast, the eigenvectors are fixed and aligned with the standard basis, which makes the historical information tractable and yields our regret bound.
>
> ---
> We will revise the paper according to your comments to improve readability, and we would appreciate a reevaluation of our paper's score. We are happy to provide further details if needed.

---

> > ### Author Rebuttal · Reviewer_tEEe · 2026-04-04
> >
> > I thank the authors for the detailed responses and I decide to keep my current evaluation.

---

> > > ### Author Response · Authors · 2026-04-04
> > >
> > > We are grateful for the reviewer’s constructive feedback and recognition of our work. We will carefully integrate your suggestions into the revised manuscript to enhance its quality.

---

### Official Review · Reviewer_JHLa · 2026-03-13

**Soundness:** 3
**Presentation:** 3
**Significance:** 3
**Originality:** 3
**Overall Recommendation:** 5
**Confidence:** 3

**Summary:**

The paper proposes a new analysis for reducing the dimension-dependence multiplicative factor from $d^{3/2}$ and $d^2$ to $d^{1/2}$ and $d$ in the gradient-variation regret bounds of a number of bandit convex optimization (BCO) setups for linear, convex and strongly convex functions. In many cases, the reduced dependence closes the gap between existing lower and upper bounds. The technique is extended from two-point BCO to one-point BLO, as well as dynamic and universal regrets.

**Compliance With Llm Reviewing Policy:**

Affirmed.

**Final Justification:**

The rebuttal fully addressed all of my questions.

**Key Questions For Authors:**

1. Regarding the geometric distributions + coupon collector arguments: is there a tight lower bound for $E[T_{all}]$? I think $O(nlogn)$ must be tight.

2. Correct me if I'm wrong, but in Algorithm 2 and Theorem 5, you don't actually need to know the segments $[a_i, b_i]$ of the hyper-rectangle, only that they exist? I'm trying to determine what exactly are the mathematical properties of the hyper-rectangle that allow Theorem 5 to hold, that don't exist in the more standard assumption of a compact set.

3. Why is the dimension-dependence factor $d^{7/2}$ so large in Theorem 5? The intuition is that $d$ corresponds to the number of arms $K$ in standard multi-armed bandits (MABs), and I'm not aware of any data-dependent bounds in MABs have that large dependency on $K$.

**Limitations:**

Yes

**Strengths And Weaknesses:**

## Soundness
The paper is technically sound. I got interested in the technique behind the dimension reduction technique in the paper, and so I verified the proof of appendix B "Analysis of Non-Consecutive Sampling Gap". The proof is correct.

## Presentation
The paper is well-written

## Significance:
The results appear to be novel and of significant interest, as the dependency on $d$ is probably tight. The proposed analyses do not introduce new assumptions nor require any trade-off with other quantities in the regret bounds in order to obtain improved dependency on $d$.

More importantly, I think the new analysis techniques can be applied in a broader range of problems, thus the paper potentially has a high impact on future research.

## Originality
I understand that the paper's originality is not limited to the observation that the sum $\sum_{i=1}^d \rho_{t,i}$ can be replaced by their max, but also several key new steps in analyzing different optimization and games setup, so I think the paper's novelty and impact are good.

---

> ### Author Rebuttal · Authors · 2026-03-31
>
> Thank you for your valuable feedback and for the careful review of our paper. Below, we address your concerns regarding the weaknesses and provide detailed answers to your questions.
>
> ---
> **Q1:** Regarding the geometric distributions + coupon collector arguments: is there a tight lower bound for $\mathbb{E}[T_{all}]$? I think $O(n\log n)$ must be tight.
>
> **A1:** Thanks for this insightful comment. The reviewer is correct: the $O(n \log n)$ bound is indeed tight. Specifically, for the standard Coupon Collector's Problem, the expected time is exactly $\mathbb{E}[T_{all}] = n H_n$, where $H_n$ is the $n$-th harmonic number. Since $H_n = \ln n + \gamma + O(1/n)$, it follows that $\mathbb{E}[T_{all}] = \Theta(n \log n)$, which matches the lower bound.
>
> ---
> **Q2:** Correct me if I'm wrong, but in Algorithm 2 and Theorem 5, you don't actually need to know the segments $[a_i,b_i]$ of the hyper-rectangle, only that they exist? I'm trying to determine exactly what mathematical properties of the hyper-rectangle allow Theorem 5 to hold and do not exist under the more standard assumption of a compact set.
>
> **A2:** Thanks for the question. To clarify, explicit knowledge of the segments $[a_i, b_i]$ is indeed required to construct the **log-barrier function** $\mathcal{R}(\mathbf{w}) = - \sum_{i=1}^d (\log(w_i - a_i) + \log(b_i - w_i))$, as stated in Line 381 in the left column, which is essential for the algorithm update.
>
> ---
> **Q3:** Why is the dimension-dependence factor $d^{7/2}$ so large in Theorem 5? The intuition is that $d$ corresponds to the number of arms $K$ in standard multi-armed bandits (MABs), and I'm not aware of any data-dependent bounds in MABs that have such a large dependency on $K$.
>
> **A3:** Thank you for this insightful question. The $d^{7/2}$ dependence, while seemingly large compared to discrete MABs, stems from the compounding technical challenges of achieving adaptivity in one-point BLO. This factor is the cumulative result of several components:
> - **Adaptivity**: The adaptivity to gradient variation contributes a factor of $d^2$.
> - **Self-Concordant Geometry**: While self-concordant analysis typically introduces a $d^{1/2}$ factor, our derivation incurs a factor of $d$. Specifically, to establish the $V_T$ regret bound, the step size $\eta$ must be constrained to ensure the non-positivity of the third term on the RHS of Eq. (D.10). This technical requirement prevents us from performing the standard balancing of terms that would otherwise yield the optimal $d^{1/2}$ dependence.
> - **Domain Diameter**: The diameter $R$ contributes $d^{1/2}$ (since $R = O(\sqrt{d})$).
>
> While we acknowledge the large dimension dependence and the focus on specific domains, we emphasize that our work provides the first gradient-variation bound in one-point BLO. Establishing this feasibility is a non-trivial first step; we consider optimizing the dimension dependence and generalizing the domain constraints to be compelling directions for our future research.
>
> ---
>
>
> We will revise the paper according to your comments to help readers understand this work better. Thanks again for your insightful review and appreciation of our work!

---

> > ### Author Rebuttal · Reviewer_JHLa · 2026-04-01
> >
> > The authors' answers addressed all of my questions. I've went through the other questions by other reviewers as well. I'll keep my score of 5 (Accept).

---

> > > ### Author Response · Authors · 2026-04-01
> > >
> > > Thank you very much for your thoughtful and encouraging response!

---

### Official Review · Reviewer_YaLQ · 2026-03-13

**Soundness:** 3
**Presentation:** 2
**Significance:** 2
**Originality:** 2
**Overall Recommendation:** 5
**Confidence:** 3

**Summary:**

The paper studies gradient variation in Bandit Convex Optimization (BCO) with two-point feedback. By proposing a refined analysis on the non-consecutive gradient variation, they improve the dimension dependence for both convex and strongly convex functions compared with the best known results (Chiang et al., 2013).

**Compliance With Llm Reviewing Policy:**

Affirmed.

**Final Justification:**

The questions are addressed in the rebuttal.

**Key Questions For Authors:**

See the questions above.

**Limitations:**

See questions above.

**Strengths And Weaknesses:**

The paper is technically strong and establishes regret guarantees under different setups and different metrics. The structure of the paper is relatively clear and the paper is thoughtfully written.

My major questions and comments can be found below.

1. Technical contribution. I found it difficult to evaluate the contribution of the paper compared to Chiang et al. (2013).

1.1. Initially, I thought Section 3.1 contained the main new technical ideas of the paper. However, after reading more carefully and checking prior work, it seems that a substantial portion of the derivation in Section 3.1 may already appear in Chiang et al. (2013). For example, the argument in Lines 203–210 (right column) seems similar to Chiang et al. (2013), and the derivation in Lines 220–244 (left column) also appears closely related to prior analysis. The main new ingredient I could identify is the sharper bound in Lines 250–256 (left column). Because prior ideas and new arguments are mixed in the presentation, it is difficult to clearly evaluate the novelty of the analysis. It would help if the authors could separat which parts are adapted from prior work and which parts are genuinely new.

1.2. Related to this point, in Section 4, the authors mention that "In this section, beyond the two-point setup, we demonstrate the versatility of our technique in the one-point Bandit Linear Optimization (BLO) setting." What is "our technique" specifically? Does it refer to a specific equation presented before, a new lemma, or something else?


2. It is also not very clear to me why the paper studies two-point BCO. What are the practically relevant applications for two-point BCO? Or what is the theoretical motivations to study this problem?


My minor comments are as follows.

1. Line 121, right column: Here, the OOGD is defined using $x_t$ as the updating variable, while later in Algorithm 1, $w_t$ becomes the update variable. It would be helpful to clarify the difference between $x_t$ and $w_t$.

2. Line 251, left column: It seems that a square is missing in the first line of the equation.

---

> ### Author Rebuttal · Authors · 2026-03-31
>
> Thank you for the insightful feedback. Below, we clarify our contributions and how our approach differs from prior work.
>
> ---
> **Q1: Difficulty distinguishing new contributions from prior analysis in Section 3.1. Please separate parts adapted from Chiang et al. (2013) from genuinely new technical ideas.**
>
> **A1:** Thanks for the suggestion! The core technical novelty of our work in the convex setting is **establishing a novel connection between the non-consecutive sampling gap $\rho_{t,i}$ and the Coupon Collector’s Problem (CCP),** which improves the result from $O(d^2\sqrt{V_T})$ to $\tilde{O}(d^{3/2}\sqrt{V_T})$.
>
> To better distinguish our own contributions from prior analysis, we clarify the structure of Sec 3.1 as follows:
>
> **Lines 199–219 (Prior Decomposition):** We adopt the decomposition from Chiang et al. (2013) to introduce $\rho_{t,i}$.
>
> **Lines 220–245 (Limitations of Prior Approach):** We show that in the convex case, a naive application of earlier techniques leads to an extra factor of $d$ compared to the linear case due to the coupling between variables. Note that the analysis in this part is **not** used in our proof.
>
> **Lines 246–263 (Novel Technical Contribution):** We propose a **new decomposition for Eq. (3.2)** that decouples the aforementioned interdependence. By further establishing a novel connection between $\max_{i \in [d]} \rho_{t,i}$ and CCP (detailed in App B), we improve the additional factor from $d$ to $\log d$.
>
> To improve readability, in the revised version, we will:
>
> 1. Condense the preliminary analysis (Lines 219L–219R) into a single lemma.
>
> 2. Add a detailed comparison between our analysis and that of Chiang et al. (2013).
>
> 3. Highlight our technical novelty: Connection between $\rho_{t,i}$ and CCP.
>
> ---
> **Q2: What specifically is "our technique" mentioned in Sec 4 for one-point BLO?**
>
> **A2:** Thanks for the question. Our technique refers to the **integration of our gradient estimator (Eq. 2.2) and the analysis of non-consecutive sampling gaps.** Specifically, we extend it to one-point BLO in two key aspects:
>
> - **Estimator Design:** We design a new gradient estimator for one-point BLO (Eq. 4.2) that mirrors the optimistic structure of Eq. (2.2). In both estimators, the squared difference between the gradient estimator $\mathbf{g}_t$ and the optimism $\tilde{\mathbf{g}}_t$ is related to the non-consecutive gradient variation $\bar{V}_T$.
>
> - **Theoretical Framework:** By adapting our non-consecutive gap analysis, we obtain $O(d^{7/2} \sqrt{V_T \log^3 T})$. This is the first gradient-variation bound for one-point BLO (under hyper-rectangular domains).
>
> We will revise the manuscript to clarify "our technique" as above. Thanks again for your comment.
>
> ---
> **Q3: Motivations for two-point BCO: practical applications or theoretical grounds?**
>
> **A3:** Thanks for the question. Two-point BCO is a foundational framework due to its inherent ties to zeroth-order optimization [1, 2] and its applicability in black-box adversarial attacks [3] and evolutionary strategies [4].
>
> Beyond its utility, two-point BCO serves as a **critical bridge** between the full-information setup and one-point BCO. That is, the techniques developed for two-point BCO might also benefit advances in one-point BCO. As demonstrated in Sec 4, the techniques developed for the two-point case directly benefit the more challenging one-point BLO setting, yielding **the first** gradient-variation regret bound for one-point BLO under hyper-rectangular domains.
>
> [1] Random gradient-free minimization of convex functions, FoCM 2015
>
> [2] Optimal rates for zero-order convex optimization: The power of two function evaluations, TIT 2015
>
> [3] ZOO: Zeroth-order optimization based black-box attacks to deep neural networks without training substitute models, AISec 2017
>
> [4] Evolution strategies as a scalable alternative to reinforcement learning, OpenAI 2017
>
> ---
> **Q4: Line 121R: Clarify the difference between $\mathbf{x}_t$ and $\mathbf{w}_t$.**
>
> **A4:** Thanks for this observation. The distinction stems from differences between the two settings:
>
> - **$\mathbf{x}_t$ (Full-info):** Used in Line 121 to define the OOGD framework. It is not a variable in our bandit algorithm.
> - **$\mathbf{w}_t$ (Bandit):** The actual update variable in Algorithm 1. The query points $\mathbf{x}\_{t,1}, \mathbf{x}_{t,2}$ are sampled as perturbations around $\mathbf{w}_t$ to obtain bandit feedback.
>
> ---
> **Q5: Line 251L: Missing square in the first line of the equation.**
>
> **A5:** Thank you for pointing it out. We have corrected this typo in the revised version.
>
> ---
> To conclude, we improve dimension dependence in two-point BCO by introducing the non-consecutivity structure. As showcased in Sec 4, this structure is also useful for other problems beyond two-point BCO.
>
> We hope our responses have addressed your concerns and would appreciate a reevaluation of our paper's score. We are happy to provide further details if needed.

---

> > ### Author Rebuttal · Reviewer_YaLQ · 2026-04-03
> >
> > My questions have been addressed and I thank the authors for their response.

---

> > > ### Author Response · Authors · 2026-04-03
> > >
> > > We thank the reviewer for the positive feedback and recognition. We will incorporate your valuable suggestions into the revised manuscript.

---

### Decision · Program_Chairs · 2026-04-30

**Decision:**

Accept (spotlight)

**Comment:**

This paper proposes a novel method that improves the dimension dependence on gradient variations bounds in Bandit Convex Optimization with two-point feedback and in linear Bandits with one point feedback.
Thereby they close the theoretical lower and upper bounds in several applications.

All reviewers agree that the results are sound and that the contributions are significant.